# The alarmin interleukin-1α triggers secondary degeneration through reactive astrocytes and endothelium after spinal cord injury

Floriane Bretheau[1], Adrian Castellanos-Molina[1], Dominic Bélanger[1], Maxime Kusik [1], Benoit Mailhot[1], Ana Boisvert[1], Nicolas Vallières[1], Martine Lessard[1], Matthias Gunzer [2,3], Xiaoyu Liu[4], Éric Boilard [5], Ning Quan[4] & Steve Lacroix [1] ✉

Spinal cord injury (SCI) triggers neuroinflammation, and subsequently secondary degeneration and oligodendrocyte (OL) death. We report that the alarmin interleukin (IL)−1α is produced by damaged microglia after SCI. Intracisterna magna injection of IL-1α in mice rapidly induces neutrophil infiltration and OL death throughout the spinal cord, mimicking the injury cascade seen in SCI sites. These effects are abolished through co-treatment with the IL-1R1 antagonist anakinra, as well as in IL-1R1-knockout mice which demonstrate enhanced locomotor recovery after SCI. Conditional restoration of IL-1R1 expression in astrocytes or endothelial cells (ECs), but not in OLs or microglia, restores IL-1α-induced effects, while astrocyte- or EC-specific *Il1r1* deletion reduces OL loss. Conditioned medium derived from IL-1α-stimulated astrocytes results in toxicity for OLs; further, IL-1α-stimulated astrocytes generate reactive oxygen species (ROS), and blocking ROS production in IL-1α-treated or SCI mice prevented OL loss. Thus, after SCI, microglia release IL-1α, inducing astrocyte- and EC-mediated OL degeneration.

The pathophysiology of spinal cord injury (SCI) is divided into two distinct phases: the primary damage and the secondary damage[1,2]. The initial mechanical trauma caused by SCI, characterized by the initial impact and the possible subsequent compression, leads to reduced blood flow in the spinal cord and ischemia, an increased permeability of the blood-spinal cord barrier (BSCB), and the development of hemorrhages at the lesion site. All of these conditions initiate cell death within the first few min to the first few h following the injury. These processes rapidly lead to a series of biochemical changes, ultimately culminating in a progressive wave of secondary damage[3]. Recent work in proteomics has predicted that the transition from primary to secondary damage after SCI and the accompanying neuroinflammatory responses are triggered by the release of danger-associated molecular patterns (DAMPs, also referred to as alarmins) from disrupted cells[4]. The most prevalent DAMPs in the injured CNS are nucleic acids and nucleotide derivatives, including ATP, high-mobility group box (HMGB) proteins, S100 class of proteins, interleukin (IL)−33 and IL-1α[5–10]. However, blocking these

[1]Axe neurosciences du Centre de recherche du Centre hospitalier universitaire (CHU) de Québec–Université Laval et Département de médecine moléculaire de l'Université Laval, Québec G1V 4G2 QC, Canada. [2]Institute for Experimental Immunology and Imaging, University Hospital Essen, University of Duisburg-Essen, Essen D-45141, Germany. [3]Leibniz-Institut für Analytische Wissenschaften-ISAS-e.V., Dortmund, Germany. [4]Charles E. Schmidt College of Medicine, Florida Atlantic University, Jupiter 33458 FL, USA. [5]Axe maladies infectieuses et immunitaires du Centre de recherche du CHU de Québec–Université Laval et Centre de recherche en arthrite de l'Université Laval, Québec G1V 4G2 QC, Canada. ✉e-mail: Steve.Lacroix@crchul.ulaval.ca

individual DAMPs or their receptors has only shown modest improvements in protecting CNS cells from death and improving functional recovery after injury[10]. In some cases, these interventions can worsen neurological outcomes[9,11], with many conflicting studies reporting either beneficial or detrimental effects in mutant mouse lines with SCI[12–14]. The only exceptions to this ambiguity are IL-1α and ATP and their receptors, whose neutralization in vivo consistently reduced neuroinflammation and secondary damage and improved neurological outcome after CNS injury[7,8,15,16]. The relationship between these processes remains poorly understood and requires further investigation.

We recently established that microglia and/or macrophages at sites of SCI rapidly produce the alarmin IL-1α, in turn triggering neuroinflammation and locomotor deficits[8]. The presence of DAMPs at sites of SCI has also been associated with the production and release of inflammatory mediators, such as the proinflammatory cytokines IL-1β and tumor necrosis factor (TNF). Of particular interest is the fact that CNS endothelial cells (ECs) abundantly express both IL-1R1 and TNF receptor 1 (TNFR1), and are readily activated by inflammatory environments[17,18]. Activation of endothelial IL-1R1 and TNFR1 results in various responses, including cytokine and chemokine release[19,20], expression of cell adhesion molecules[21,22], disruption of vascular permeability[23,24], and leukocyte trafficking[22]. Astrocytes, which are major contributors to glial scar formation after SCI[25], react quickly to these inflammatory mediators as well. Using rodent primary CNS cell cultures and post-mortem brain tissue from patients with various neurodegenerative diseases, Liddelow and colleagues recently found that microglia-derived cytokines and growth factors determine whether astrocytes will have neurotoxic or pro-survival effects[26], although it is generally acknowledged that reactive astrocytes may adopt multiple states[27]. A subtype of reactive astrocytes termed A1 astrocytes, activated by microglia-derived IL-1α (but not IL-1β), TNF and complement component 1q (C1q), may play a role in the mechanisms leading to the death of OLs and neurons in diseases such as Alzheimer's, Huntington's and Parkinson's disease, amyotrophic lateral sclerosis and multiple sclerosis. However, not all microglia and astrocyte responses have a detrimental impact on functional recovery in the context of SCI, and depletion of these cells may prompt negative outcomes such as disruption of scar formation, enhanced parenchymal immune infiltrates, and reduced OL and neuronal survival[28,29]. The extent to which signals from microglia, astrocytes, ECs and blood-derived immune cells are driving secondary degeneration after SCI therefore remains an open question.

The mechanisms underlying secondary tissue damage after SCI generate an environment that is particularly toxic for OLs[30,31]. In mice, the number of OLs rapidly decreases during the first 24 h post-SCI, with peak loss occurring at 1 week post-injury[32]. Apoptotic death of OLs has been detected in the degenerating white matter tracts of rodents, monkeys and humans until at minimum 2–3 weeks post-SCI[33–35]. Whether this OL loss is directly responsible for the demyelination and conduction failure observed in axons that survived the initial trauma[36,37], or the behavioral deficits detected after SCI, remains however uncertain.

Following up on our recent discovery that deletion of the *Il1a* gene protects OLs and reduces tissue damage after SCI, we investigated the cellular and molecular mechanisms that underlie IL-1α-dependent effects in the normal and injured mouse spinal cord. We found that microglia-derived IL-1α induces OL death through the astrocytic and endothelial IL-1R1. Mechanistically, stimulation of astrocytic IL-1R1 by IL-1α converted these cells into a reactive phenotype characterized by enhanced production and release of reactive oxygen species (ROS)

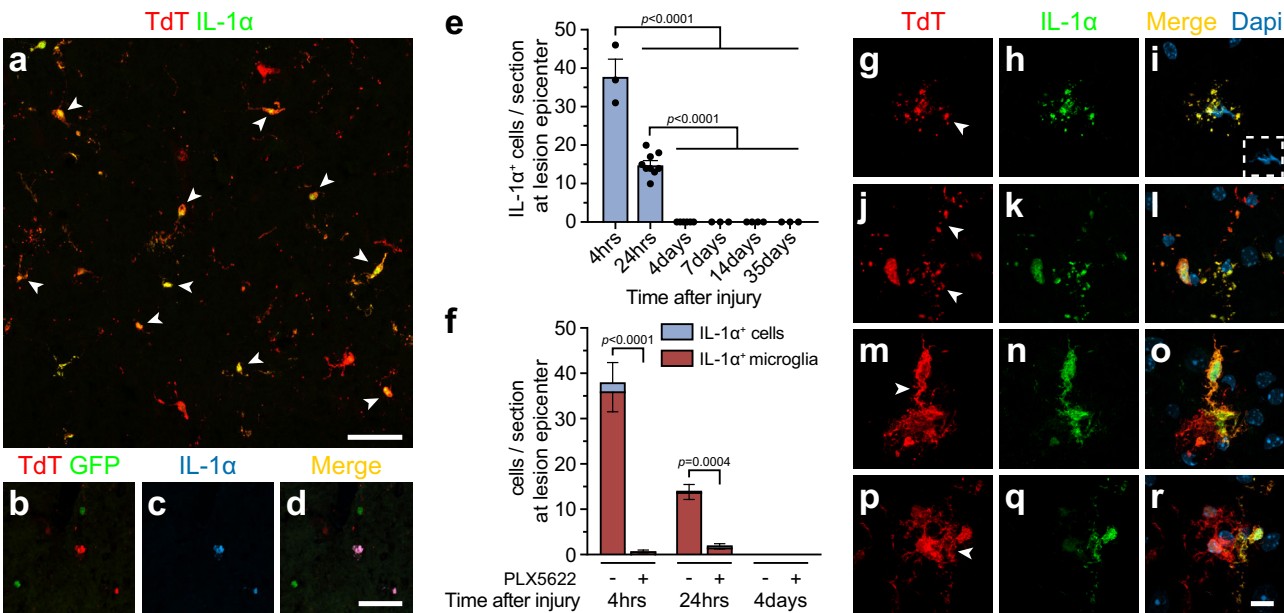

**Fig. 1 | Damaged microglia rapidly produce IL-1α at the site of spinal cord contusion in mice. a** Representative confocal image showing IL-1α immunostaining (green) in the spinal cord of an injured *Cx3cr1*Cre::*Rosa26*TdT transgenic mouse, in which microglia express the fluorescent reporter Td-Tomato (TdT, red), at 4 h post-SCI. White arrowheads point to specific double-labeled cells. **b–d** Confocal immunofluorescence images showing expression of IL-1α (blue) in microglia (TdT, red), but not macrophages (LysM, green), in the spinal cord of an injured *LysM-eGFP*::*Cx3cr1*CreER::*R26-TdT* mouse at 4 h post-SCI. **e, f** Quantification of IL-1α-positive (+) cells and IL-1α⁺TdT⁺ microglia at the lesion epicenter in untreated *Cx3cr1*Cre::*Rosa26*TdT mice (**e** $n = 3$ at 4 h, $n = 9$ at 24 h, $n = 6$ at 4 days, $n = 3$ at 7 days, $n = 4$ at 14 days, $n = 3$ at 35 days) and fluorescent reporter mice treated with PLX5622 or the control diet (**f** $n = 3$ control diet 4 h, $n = 3$ PLX diet 4 h, $n = 4$ control diet 24 h, $n = 4$ PLX diet 24 h, $n = 3$ control diet 4 days, $n = 3$ PLX diet 4 days) and killed at various time points post-SCI. **g–r** High magnification confocal images of IL-1α⁺ TdT⁺ microglia revealed that these cells often exhibit damaged cell bodies and processes (**g–l**) or have retracted, swollen processes, indicative of an activated status (**m–r**). White arrowheads point to relevant cell morphologies reminiscent of damaged (**g–l**) or activated (**m–r**) cells. Nuclear staining (DAPI) is shown in blue (**i, l, o, r**) in the merged images. Data are presented as mean values +/− SEM and statistical significance was determined by a one-way (**e**) or two-way (**f**) ANOVA with Bonferroni's post-hoc test. Pairwise comparisons and p-values are indicated in the graphs. Scale bars: (**a**) 50 µm, (**b–d**, in **d**) 50 µm, (**g–r**, in **r**) 10 µm.

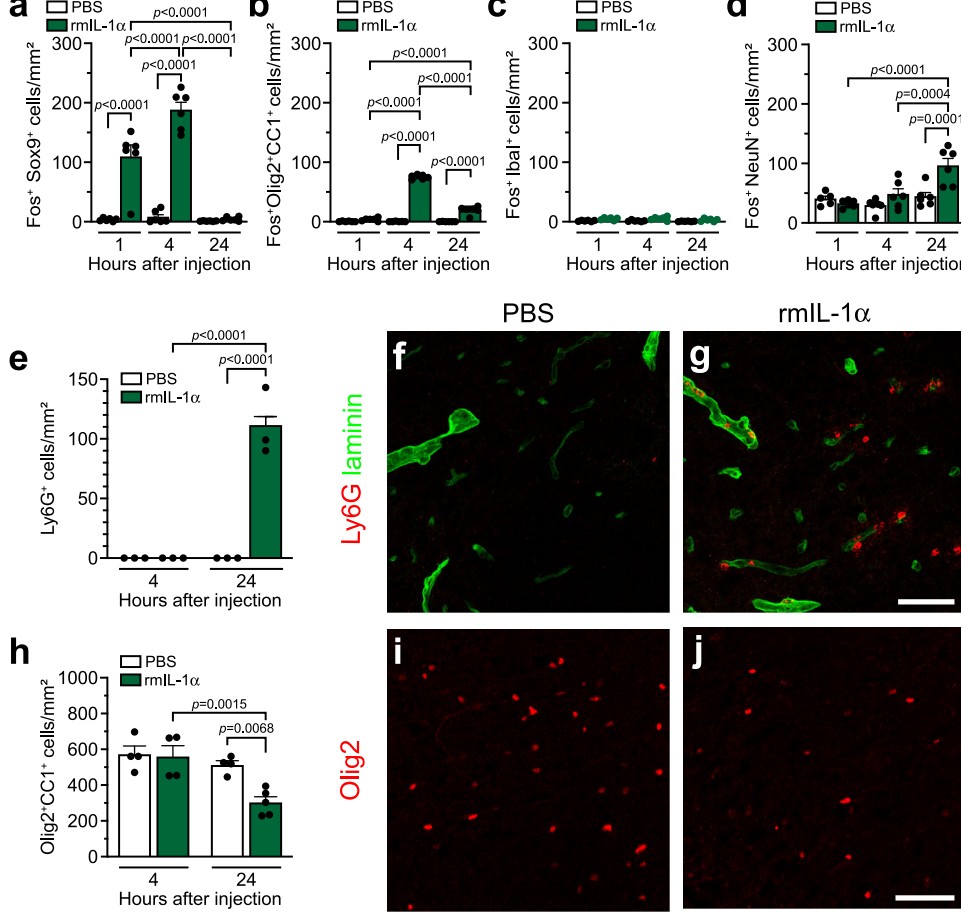

**Fig. 2 | Injection of IL-1α into the CNS of mice induces rapid activation of glial cells, neutrophil infiltration, and loss of mature oligodendrocytes in the spinal cord. a–d** Quantification of the activation marker Fos in astrocytes (Fos⁺Sox9⁺ cells), oligodendrocytes (Fos⁺Olig2⁺CC1⁺), microglia (Fos⁺Iba1⁺), and neurons (Fos⁺NeuN⁺) in the spinal cord of C57BL/6 mice injected with either PBS or rmIL-1α i.c.m. and killed at 1, 4 or 24 h post-injection ($n = 6$ mice/group). **e** Quantification of the total number of Ly6G⁺ neutrophils that infiltrated the spinal cord at 4 and 24 h post-injection of PBS or rmIL-1α ($n = 3$ mice/group). **f, g** Representative confocal images showing Ly6G (a marker of neutrophils, red) and laminin (a marker of blood vessel basement membranes, green) immunostainings in the spinal cord of C57BL/6 mice injected with either PBS (**f**) or rmIL-1α (**g**) at 24 h post-injection. **h** Quantification of the total number of Olig2⁺CC1⁺ mature oligodendrocytes in the spinal cord white matter of C57BL/6 mice at 4 and 24 h post-i.c.m. injection of either PBS or rmIL-1α ($n = 4$–5 mice/group: $n = 4$ PBS 4 h, $n = 4$ rmIL-1α 4 h, $n = 4$ PBS 24 h, $n = 5$ rmIL-1α 24 h). **i, j** Representative confocal images showing immunostaining for the oligodendrocyte transcription factor 2 (Olig2, red cells), a nuclear marker of oligodendrocyte lineage cells, in the spinal cord of C57BL/6 mice at 24 h post-injection of either PBS (**i**) or rmIL-1α (**j**). Data are presented as mean values +/− SEM and statistical significance was determined by a two-way ANOVA followed by a Bonferroni post-hoc test (**a–e**, **h**). Pairwise comparisons and *p*-values are indicated in the graphs. Scale bars: (**f, g**, in **g**) 50 μm, (**i, j**, in **j**) 50 μm.

inciting toxicity in OLs. Moreover, blocking ROS production in IL-1α-injected mice and SCI mice prevented OL death. Thus, blocking IL-1R1 signaling or ROS production may be a promising therapeutic avenue to prevent OL loss following a CNS insult.

## Results

### IL-1α is produced by injured microglia during the early acute phase of SCI

We previously demonstrated that microglia and/or macrophages are the main producers of IL-1α after SCI[8]. To clarify which of these two cell types primarily produces IL-1α, we performed immunofluorescence staining against IL-1α on spinal cord tissue sections from *Cx3cr1*^CreER::*R26-TdT* and *LysM-eGFP::Cx3cr1*^CreER::*R26-TdT* mice killed at 4 h post-SCI (Fig. 1a–d & Supplementary Fig. 1a–d), which corresponds to the peak of IL-1α protein expression (Fig. 1e). Both *Cx3cr1*^CreER::*R26-TdT* and *LysM-eGFP::Cx3cr1*^CreER::*R26-TdT* mice received tamoxifen treatment 1 month prior to SCI to drive TdT expression specifically in microglia, and not in peripheral myeloid cells, as previously established[29]. Quantification revealed that nearly all (95 ± 5%) IL-1α-positive (+) cells also expressed TdT, suggesting

that they are microglia (Fig. 1f). To confirm this finding, microglia were depleted in vivo using PLX5622, a CSF1R inhibitor from Plexxikon. As shown in Fig. 1f and Supplementary Fig. 1e–j, very few IL-1α⁺ cells were detected in the injured spinal cord of *Cx3cr1*^CreER::*R26-TdT* mice fed PLX5622 chow (0.7 ± 0.3% IL-1α⁺ cells/mm²) compared to those fed the control diet (38.0 ± 4.4 IL-1α⁺ cells/mm²). Thus, our results show that microglia located at the lesion epicenter and surrounding damaged areas are the main cellular source of IL-1α production after SCI.

Consistent with its role as an alarmin, IL-1α is typically synthesized as a precursor protein and constitutively stored in the nucleus and cytoplasm of cells, and we next sought to determine the health of IL-1α-producing microglia in the injured mouse spinal cord. Most IL-1α⁺ microglia had both damaged cell bodies/processes or exhibited retracted, swollen processes, indicating an activated status (Fig. 1g–r). Confocal microscopy images revealed a homogenous distribution of IL-1α protein within the whole microglia (nucleus and cytoplasm). Together, these results show that damaged microglia produce the alarmin IL-1α during the early acute phase of SCI.

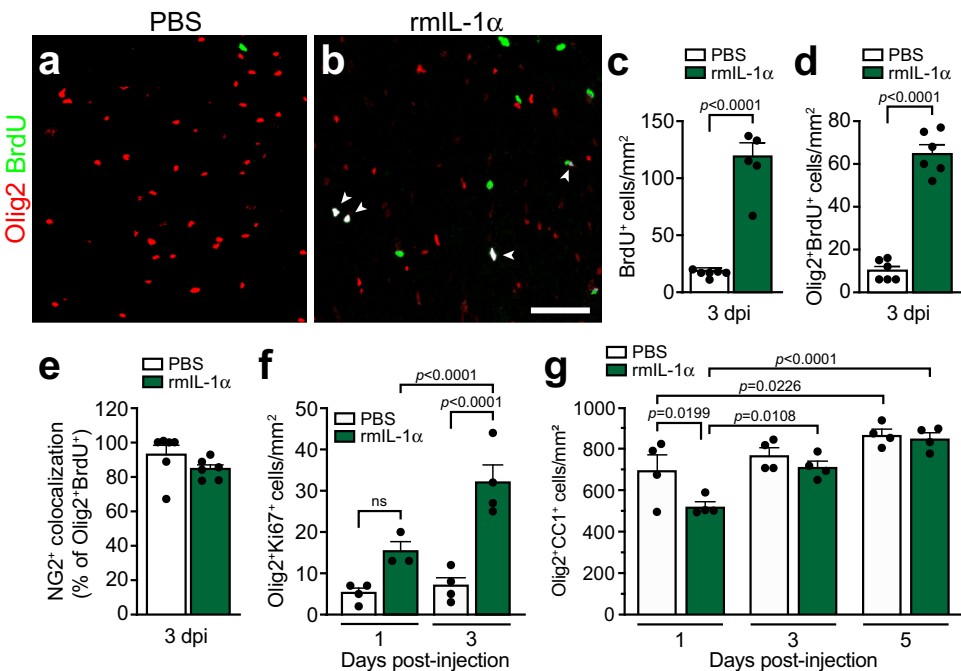

**Fig. 3 | Proliferating oligodendrocyte precursor cells rapidly restore the number of mature oligodendrocytes in mice injected centrally with IL-1α.**
**a**, **b** Representative confocal images showing Olig2 (red) and BrdU (a marker of cell proliferation, green) immunostainings in the spinal cord white matter of C57BL/6 mice injected with either PBS (**a**) or rmIL-1α (**b**) at 3 days post-injection. White arrowheads indicate certain proliferating oligodendrocyte lineage cells.
**c**, **d** Quantification of the total number of BrdU⁺ cells (**c**) and Olig2⁺BrdU⁺ double-positive cells (**d**) in the spinal cord white matter at 3 days post-injection of PBS or rmIL-1α intra-cisterna magna (i.c.m.) ($n = 6$ mice/group). **e** Percentage of Olig2⁺ BrdU⁺ cells coexpressing the NG2 marker, an indicator of OPCs ($n = 6$ mice/group).

**f** Quantification of the total number of Olig2⁺Ki67⁺ double-positive cells in the spinal cord white matter of C57BL/6 mice at 1 and 3 days post-i.c.m. injection of either PBS or rmIL-1α ($n = 3–4$ mice/group: $n = 4$ PBS Day 1, $n = 3$ rmIL-1α Day 1, $n = 4$ PBS Day 3, $n = 4$ rmIL-1α Day 3). **g** Quantification of the total number of Olig2⁺ CC1⁺ mature oligodendrocytes in the spinal cord white matter at 1, 3 and 5 days post-i.c.m. treatment with PBS or rmIL-1α ($n = 4$ mice/group). Data are presented as mean values +/− SEM and statistical significance was determined by either a one (**c–e**) or two-way (**f–g**) ANOVA followed by a Bonferroni post-hoc test. Pairwise comparisons and p-values are indicated in the graphs. Scale bars: (**a**, **b**, in **b**) 50 μm.

## Acute central delivery of IL-1α in mice induces activation of glial cells throughout the spinal cord

Trauma-induced necrotic cell death, as it occurs after SCI, is likely to induce the release of several different alarmins at roughly the same time, such as IL-1α, HMGB1, IL-33, and ATP[38,39]. In order to study the effects of IL-1α in a more straightforward in vivo model, we injected murine recombinant IL-1α (rmIL-1α) intra-cisterna magna (i.c.m.) to C57BL/6 mice and measured systemic plasma levels of 32 total cytokines and chemokines at 3 different time points (1, 4 and 24 h) post-injection (Supplementary Fig. 2). When compared to traumatic SCI, central delivery of IL-1α induced a similar profile of cytokines/chemokines in the plasma, with the time course being slightly accelerated in mice directly injected with the alarmin. We then assessed the activation of neurons and various glial cells in the spinal cord of IL-1α-injected mice at identical time points. Expression of the transcription factor c-Fos (Fos) was used as marker of increased transcriptional activity, which we define as cell activation[40]. Immunostaining for Fos in spinal cord tissue sections from mice injected with IL-1α revealed that Sox9⁺ astrocytes are activated as early as 1 h post-injection (Fig. 2a), while activation of Olig2⁺ CC1⁺ oligodendrocytes (OLs) was slightly delayed, occurring at 4 h (Fig. 2b). The peak of activation for both glial cell types was observed at 4 h, with a total of 187.7 ± 13.3 Fos⁺ Sox9⁺ astrocytes/mm² and 75.2 ± 1.5 Fos⁺ Olig2⁺ CC1⁺ OLs/mm² in the thoracic spinal cord white matter (Fig. 2a, b & Supplementary Fig. 3a–g). The number of activated astrocytes and OLs decreased afterward to 4.4 ± 1.4 cells/mm² and 19.3 ± 2.8 cells/mm², respectively, at 24 h post-injection. In contrast, we found no evidence of microglia activation after IL-1α injection, as demonstrated by the absence of colocalization of Fos with Iba1 immunostaining at 1, 4 and 24 h after IL-1α injection (Fig. 2c & Supplementary Fig. 3h–j). For all time points examined,

nearly zero Fos⁺ cells were detected in the spinal cord white matter of PBS-injected mice. Unlike glial cells, neuronal activation peaked at 24 h post IL-1α injection, as determined by the upregulation of the Fos protein in NeuN⁺ cells of the spinal cord gray matter (Fig. 2d & Supplementary Fig. 3k–m). However, this was the final time point we analyzed, and peak neuronal activation may well have occurred later. Thus, it appears that cell type activation in response to IL-1α injection follows a precise timeline, from astrocytes to OLs and then to neurons.

## Central delivery of IL-1α induces neutrophil infiltration into the neurovascular space and parenchyma of the spinal cord

With the knowledge that IL-1α initiates sterile inflammation by recruiting neutrophils in Matrigel plugs supplemented with necrotic cell products[41], we examined whether IL-1α injection resulted in the recruitment of Ly6G⁺ neutrophils along the rostro-caudal spinal cord axis. At 4 h post-IL-1α injection, no Ly6G⁺ cells were observed in the spinal cord (Fig. 2e). However, the situation drastically changed at 24 h, wherein a massive influx of neutrophils in both male and female mice was detected (Supplementary Fig. 4a). As shown in Fig. 2f, g and Supplementary Fig. 4b, neutrophils were found both in the neurovascular space and tissue parenchyma throughout the entire rostro-caudal axis of the spinal cord after i.c.m. injection of IL-1α, but not PBS. Infiltrating innate immune cells such as neutrophils are thus implicated in the cascade of cellular events elicited by IL-1α, succeeding the activation of glial cells.

## IL-1α induces the death of mature oligodendrocytes in the mouse spinal cord

Neuroinflammation is suspected to be a major cause of the secondary degeneration occurring after SCI. We thus evaluated whether the

presence of IL-1α in the spinal cord affects cell health and homeostasis, placing our focus on neurons and cells of OL lineage, as these cell populations were found to be particularly vulnerable to secondary damage after SCI[42,43]. Mature OLs were identified as cells immunopositive for both Olig2 and CC1 in the spinal cord white matter, while neurons were identified as NeuN⁺ cells in the spinal cord gray matter. At 4 h post-injection, no differences in numbers of Olig2⁺ CC1⁺ mature OLs were seen between mice injected with either PBS or IL-1α (Fig. 2h). However, after 24 h, the total number of mature OLs was reduced by approximately 40% in mice that received IL-1α i.c.m. compared to PBS-injected control mice (Fig. 2h–j), an effect observed in both males and females (Supplementary Fig. 4c). We counted on average $512.3 \pm 24.2$ Olig2⁺ CC1⁺ cells/mm² in the spinal cord white matter of mice injected with PBS compared to $302.6 \pm 32.6$ Olig2⁺ CC1⁺ cells/mm² in mice injected with IL-1α. Given the recent discovery by Floriddia and colleagues that distinct OL populations have spatial preferences in the spinal cord and exhibit different responses to injury[44], we next investigated whether mature OLs vulnerable to IL-1α were localized at a specific spinal level or white matter tract. IL-1α-mediated OL loss was observed throughout the entire rostro-caudal axis, independent of the white matter tract analyzed (Supplementary Fig. 4d–g). However, oligodendrocyte death slightly diminished as the distance from the injection site increased. No signs of neurodegeneration were observed during the first 24 h of injection of IL-1α (Supplementary Fig. 3n). However, the increase in neutrophil infiltration and decrease in the number of mature OLs at 24 h post-IL-1α injection was amplified given higher concentrations of the alarmin (Supplementary Fig. 4h, i), thus suggesting a dose-response effect. In contrast, the concentration of IL-1β had to be increased 2.5-fold that of IL-1α to mimic its effect on OL loss. Still, IL-1α and IL-1β injected at 20 ng/μl induced similar neutrophil recruitment, suggesting separate mechanisms regulating infiltration of neutrophils and OL cell death (Pearson's correlation, $p = 0.1813$; Supplementary Fig. 4j).

### Mature oligodendrocytes are rapidly replaced following death by increased proliferation of oligodendrocyte lineage cells

Mature OLs are a highly diverse cell population all originating from a common oligodendrocyte precursor cell (OPC), whose proliferation and differentiation are regulated by extrinsic signals present in the cell microenvironment[44]. Considering mature OLs are decimated during the early acute phase of SCI[8], an effect that was replicated via acute central delivery of IL-1α in the present study, we next investigated whether the OL cell population is restored following death by performing in vivo 5'-bromodeoxyuridine (BrdU, a marker of cell proliferation) experiments. As shown in Fig. 3a–d, the number of Olig2⁺ BrdU⁺ cells was increased by ~6.4 fold in the spinal cord white matter of IL-1α-injected mice compared to the control group at 3 days post-injection. Colocalization studies indicated that 85% or more of proliferating Olig2⁺ BrdU⁺ cells are NG2⁺, thus confirming their identity as OPCs (Fig. 3e). Data from the BrdU pulse experiments were confirmed by immunofluorescence staining, showing the proliferative marker Ki67 to be augmented by ~4.6-fold in the Olig2 cell population in response to IL-1α (Fig. 3f). Importantly, the number of mature OLs in the IL-1α group returned to baseline values after 3–5 days (Fig. 3g). Altogether, these results suggest that the death of mature OLs induced by IL-1α is compensated for by the rapid proliferation of OL lineage cells.

### IL-1α induces neuroinflammation and OL cell death through IL-1R1 signaling

Evidence suggests that IL-1R1 is the main signaling receptor for IL-1α, and we postulated that IL-1R1 blockade using anakinra, a recombinant IL-1 receptor antagonist with greater affinity for IL-1R1 than IL-1α itself[45], would reduce the central effects of the cytokine. When anakinra was administered concomitantly with IL-1α, infiltration of neutrophils in the mouse spinal cord was completely blocked (Fig. 4a–c). Moreover, treatment with anakinra protected against IL-1α-induced OL cell death (Fig. 4d–f). Thus, the blockade of IL-1R1 signaling is sufficient to abolish the potential pathophysiological effects of IL-1α in the spinal cord.

### Recovery of locomotor function is improved in mice lacking IL-1R1 after SCI

Based on the previous findings, we postulated that mice deficient in IL-1R1 would exhibit reduced signs of secondary degeneration, resulting in improved functional recovery after SCI. Therefore, we investigated whether the absence of the *Il1r1* gene positively affects locomotor recovery after SCI, as previously reported for *Il1a⁻/⁻* mice[8]. Naive *Il1r1⁻/⁻*, *Il1a⁻/⁻* and WT mice all received perfect scores on the 9-point Basso Mouse Scale for locomotion (BMS) and the 11-point BMS subscore. The situation differed, however, after groups received a moderate (50 kdyn) traumatic SCI. As shown in Fig. 4g–j, *Il1r1⁻/⁻* mice displayed significantly better locomotor recovery than WT mice after SCI, while also mimicking the neurological improvements observed in *Il1a⁻/⁻* mice. Together, these results suggest that global deficiency in IL-1R1 signaling contributes to better functional recovery after SCI, likely by preventing early manifestations of secondary degeneration.

### IL-1R1 is expressed by both glial and endothelial cells in the mouse spinal cord

Next, we assessed IL-1R1 protein expression in the main cell types that reside in the spinal cord. For this, we opted for an immunoblotting approach using membrane protein extracts derived from either specific primary cell types isolated from the normal CNS, or cell lines. Immunoblot analysis revealed that IL-1R1 is expressed by CNS ECs, OPCs, astrocytes and microglia (Fig. 4k). Intriguingly, the protein detected using the anti-IL-1R1 polyclonal antibody in OPCs and astrocytes was of a slightly lower molecular weight than the predicted ~67-kDa IL-1R1 protein, suggesting the possibility that cells of the glial-restricted lineage could express a truncated splice variant of the mouse IL-1R1 receptor, referred hereafter to as τIL-1R1. Along this line, we point out that the Quan laboratory has previously shown that τIL-1R1 is identical to IL-1R1 at the C terminus, but with a shortened extracellular domain and therefore lower molecular weight[46].

### Oligodendrocyte cell death is not directly mediated by IL-1α

To investigate the mechanisms by which IL-1α mediates its effects, we took advantage of the IL-1R1-restored mouse line developed by ref. [47]. IL-1R1-restored mice, designed hereafter to as *Il1r1⁻ʳ/ʳ* mice, exhibit an IL-1R1 (and τIL-1R1) knockout phenotype that can be reversed in a cell-specific manner by Cre-mediated recombination. First, we asked whether OL cell death could be mediated by a direct effect of IL-1α on OLs. To address this question in vivo, we crossed *Il1r1ʳ/ʳ* mice with mice expressing an inducible Cre recombinase under the control of the *Pdgfra* promoter, thus restoring *Il1r1* gene expression specifically in OPCs and their cell derivatives. To measure the efficiency of Cre-mediated IL-1R1 restoration in OL lineage cells, qPCR was used to amplify the floxed *Neo* cassette, causing disruption of the *Il1r1* gene in PDGFRα⁺ OPCs and O4⁺ pro-OLs immunopanned from the adult mouse spinal cord. We estimated the restoration of *Il1r1* gene expression in *Pdgfra*^CreER^::*Il1r1ʳ/ʳ* mice to be around 20% and 25% of normal levels in PDGFRα⁺ OPCs and O4⁺ pro-OLs, respectively (Fig. 5a). As expected, no restoration of the wild-type *Il1r1* allele was observed in OL lineage cells of *Il1r1ʳ/ʳ* mice. Despite partial restoration of *Il1r1* gene expression in OLs of *Pdgfra*^CreER^::*Il1r1ʳ/ʳ* mice, we failed to restore the infiltration of neutrophils and death of mature OLs in response to i.c.m. delivery of IL-1α (Fig. 5b, c). The total number of mature OLs in the spinal cord white matter of *Pdgfra*^CreER^::*Il1r1ʳ/ʳ* mice injected with IL-1α was similar to that of *Il1r1ʳ/ʳ* mice and animals injected with PBS (Fig. 5c).

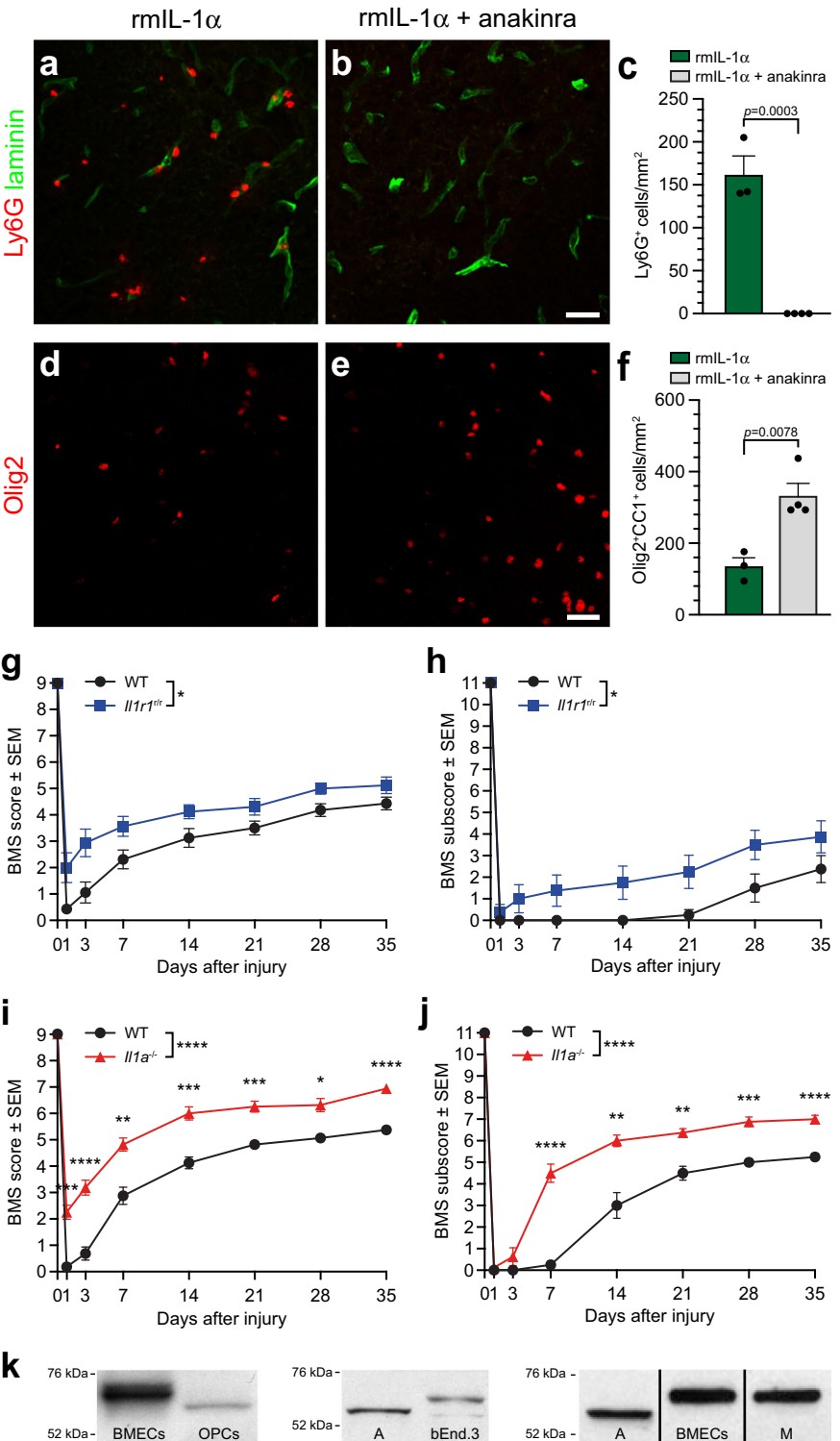

Altogether, these results suggest that IL-1α does not seem to cause OL cell death by a direct mechanism of action.

**IL-1α mediates OL cell death independently of microglial IL-1R1**

We previously established that IL-1α is released by microglia in the first few h post-SCI and that these cells express IL-1R1. Our next aim was to investigate whether IL-1α could have autocrine effects on microglia, promoting in return neuroinflammation and the death of mature OLs.

To test this, we crossed *Cx3cr1*^CreER and *Il1r1*^r/r mice to restore IL-1R1 expression specifically in microglia. The efficiency of IL-1R1 restoration was estimated at approximately 65% in microglia isolated from the normal spinal cord of adult *Cx3cr1*^CreER::*Il1r1*^r/r mice (Fig. 6a). We next conducted an experiment in which a single dose of rmIL-1α was injected i.c.m. in either WT, *Il1r1*^r/r or *Cx3cr1*^CreER::*Il1r1*^r/r mice. As shown in Fig. 6b, neutrophils infiltrated the spinal cord of WT mice within 24 h of injection, but not *Il1r1*^r/r or *Cx3cr1*^CreER::*Il1r1*^r/r mice. Similarly, only WT

**Fig. 4 | IL-1α mediates its effects in both the inflamed and injured mouse spinal cord through IL-1R1. a, b** Representative confocal images showing the presence (or absence) of Ly6G⁺ neutrophils (red cells) in the spinal cord of C57BL/6 mice injected i.c.m. with either rmIL-1α alone (**a**) or rmIL-1α + anakinra, a recombinant human IL-1R antagonist (**b**). All mice were killed at 24 h post-injection. An anti-pan-laminin antibody was used to stain blood vessel basement membranes (green staining). **c** Quantification of the total number of Ly6G⁺ neutrophils that infiltrated the spinal cord of mice (*n* = 3–4 mice/group: *n* = 3 rmIL-1α, *n* = 4 rmIL-1α + anakinra). **d, e** Representative images showing immunostaining for the Olig2 transcription factor in the spinal cord of C57BL/6 mice injected with either rmIL-1α (**d**) or rmIL-1α + anakinra (**e**) and killed at 24 h. **f** Quantification of the total number of Olig2⁺ CC1⁺ mature oligodendrocytes in the spinal cord white matter at 24 h post-i.c.m. treatment (*n* = 3–4 mice/group: *n* = 3 rmIL-1α, *n* = 4 rmIL-1α + anakinra).

**g–j** Locomotor function was assessed using the BMS score (**g, i**) and BMS subscore (**h, j**) over a 35-day period post-SCI in wild-type (WT), *Il1r1*^r/r^ (*Il1r1*^–/–^) or *Il1a*^–/–^ mice (*n* = 8 mice per group). **k** Detection by immunoblotting of IL-1R1 in various murine primary and immortalized cells including: Lane 1 = primary brain microvascular endothelial cells (BMECs), Lane 2 = primary oligodendrocyte progenitor cells (OPCs), Lane 3 = primary astrocytes (A), Lane 4 = immortalized bEnd.3 ECs (bEnd.3), Lane 5 = primary astrocytes (A), Lane 6 = primary BMECs, and Lane 7 = primary microglia (M). Data are presented as mean values +/− SEM and statistical significance was determined by either a two-tailed Student's *t*-test (**c, f**) or a two-way repeated measures ANOVA followed by a Bonferroni post-hoc test (**g–j**). When not directly indicated in the graphs, *p*-values are as follows: ****$p < 0.0001$, ***$p < 0.001$, **$p < 0.01$, *$p < 0.05$, compared to the WT group. Scale bars: (**a, b**, in **b**) 25 μm, (**d, e**, in **e**) 25 μm.

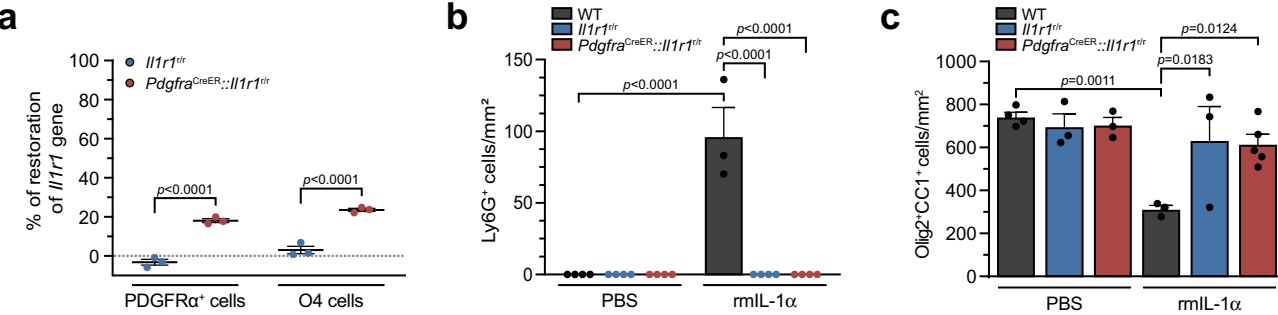

**Fig. 5 | Restoration of the *Il1r1* gene in oligodendrocyte lineage cells does not lead to IL-1α-mediated neuroinflammation and oligodendrocyte loss.**
**a** Quantification of restoration of *Il1r1* gene expression in primary PDGFRα⁺ and O4⁺ oligodendrocyte lineage cells isolated by immunopanning from the uninjured spinal cord of adult *Il1r1*^r/r^ mice, who express an IL-1R1-knockout phenotype, and *Pdgfra*^CreER^::*Il1r1*^r/r^ mice at 50 days post-tamoxifen treatment (*n* = 3 per group and each sample has a pool of 4 mice). **b** Quantification of the number of Ly6G⁺ neutrophils that infiltrated the spinal cord of WT, *Il1r1*^r/r^ and *Pdgfra*^CreER^::*Il1r1*^r/r^ mice at 24 h post-injection of either PBS or rmIL-1α (*n* = 3–4 mice/group: *n* = 4 WT + PBS,

*n* = 4 *Il1r1*^r/r^ + PBS, *n* = 4 *Pdgfra*^CreER^::*Il1r1*^r/r^ + PBS, *n* = 3 WT + rmIL-1α, *n* = 4 *Il1r1*^r/r^ + rmIL-1α, *n* = 4 *Pdgfra*^CreER^::*Il1r1*^r/r^ + rmIL-1α). **c** Quantification of the total number of Olig2⁺ CC1⁺ mature oligodendrocytes in the spinal cord white matter of WT, *Il1r1*^r/r^ and *Pdgfra*^CreER^::*Il1r1*^r/r^ mice at 24 h post-injection of either PBS or rmIL-1α (*n* = 3–5 mice/group: *n* = 4 WT + PBS, *n* = 3 *Il1r1*^r/r^ + PBS, *n* = 3 *Pdgfra*^CreER^::*Il1r1*^r/r^ + PBS, *n* = 3 WT + rmIL-1α, *n* = 3 *Il1r1*^r/r^ + rmIL-1α, *n* = 5 *Pdgfra*^CreER^::*Il1r1*^r/r^ + rmIL-1α). Data are presented as mean values +/− SEM and statistical significance was determined by a two-way ANOVA followed by a Bonferroni post-hoc test (**a–c**). Pairwise comparisons and p-values are indicated in the graphs.

mice demonstrated a decreased OL cell number in response to IL-1α treatment (Fig. 6c), suggesting that microglial IL-1R1 signaling does not seem to mediate the central effects of IL-1α.

However, when microglia in mice were depleted using PLX5622 chow, the number of neutrophils that infiltrated the spinal cord in response to acute i.c.m. delivery of IL-1α nearly doubled, passing from 165.0 ± 22.3 to 301.0 ± 39.5 Ly6G⁺ cells/mm² (Fig. 6d–j). In parallel, the number of mature OLs decreased by approximately 50% in mice that received the combination of IL-1α and PLX5622, compared to those who were injected with IL-1α and fed the control diet (Fig. 6k). This indicates that the central effects of IL-1α are amplified in the absence of microglia, with microglia-depleted mice having nearly two times more infiltrating neutrophils and half the number of mature OLs in the spinal cord. Considering the existence of a decoy IL-1 receptor, IL-1R2, we next hypothesized that the protection conferred by microglia could have been provided by an upregulation of this receptor. To test this possibility, we took advantage of tamoxifen-inducible *Cx3cr1*^CreER^::*R26-TdT* mice to specifically isolate microglia with minimal contamination by blood-derived immune cells[29]. We then extracted total RNA from microglia 24 h after injection with PBS or rmIL-1α, and measured *Il1r2* mRNA levels using quantitative real-time PCR. Data showed that IL-1R2 is undetectable at the mRNA level in microglia of PBS-injected mice, but that its expression is dramatically increased in microglia following i.c.m. injection of IL-1α (Fig. 6l). Together, these results suggest that the mechanism by which IL-1α induces OL cell death in the spinal cord occurs independently of microglial IL-1R1 signaling. Instead, microglia appear to protect OLs by sequestering IL-1 cytokines within the extracellular compartment through the decoy IL-1R2.

### Activation of IL-1R1 signaling in CNS endothelial cells causes BSCB breakdown and neutrophil entry and mediates part of the OL cell death response

The expression of IL-1R1 by CNS ECs is well described in the literature. In previous studies, our group as well as others have reported the importance of endothelial IL-1R1 signaling in the neuroinflammatory processes taking place at the blood-brain barrier (BBB) and BSCB in the experimental autoimmune encephalomyelitis (EAE) mouse model of multiple sclerosis[20–22,48]. To investigate the functional role of the endothelial IL-1R1 in exhibiting the central effects of IL-1α, *Cdh5*^CreER^ mice were crossed with *Il1r1*^r/r^ or *Il1r1*^fl/fl^ mice to restore or delete, respectively, IL-1R1 expression in ECs (Fig. 7). We determined that *Il1r1* gene expression was restored to 90% of normal levels in CNS ECs of adult *Cdh5*^CreER^::*Il1r1*^r/r^ mice at 1 month post-tamoxifen treatment (Fig. 7b). This restoration of endothelial IL-1R1 expression was sufficient to partially restore neutrophil infiltration and associated BSCB leakage observed after i.c.m. injection of IL-1α (Fig. 7c & Supplementary Figs. 5–6). As shown in Fig. 7d and Supplementary Fig. 5c, d, the average number of mature OLs in the spinal cord of *Cdh5*^CreER^::*Il1r1*^r/r^ mice was statistically higher than that observed for the WT group after central administration of IL-1α, but still reduced compared to the globally IL-1R1-deficient (*Il1r1*^r/r^) mice. Restoration of IL-1R1 expression in CNS ECs was, however, insufficient in mimicking the extent of locomotor deficits observed in SCI WT mice, as assessed using the BMS score (Fig. 7e, f). These results suggest that endothelial IL-1R1 signaling only partially contributes to the global effects of IL-1α in SCI pathogenesis.

This prompted us to investigate the potential contribution of endothelial IL-1R1 in another transgenic mouse model, namely

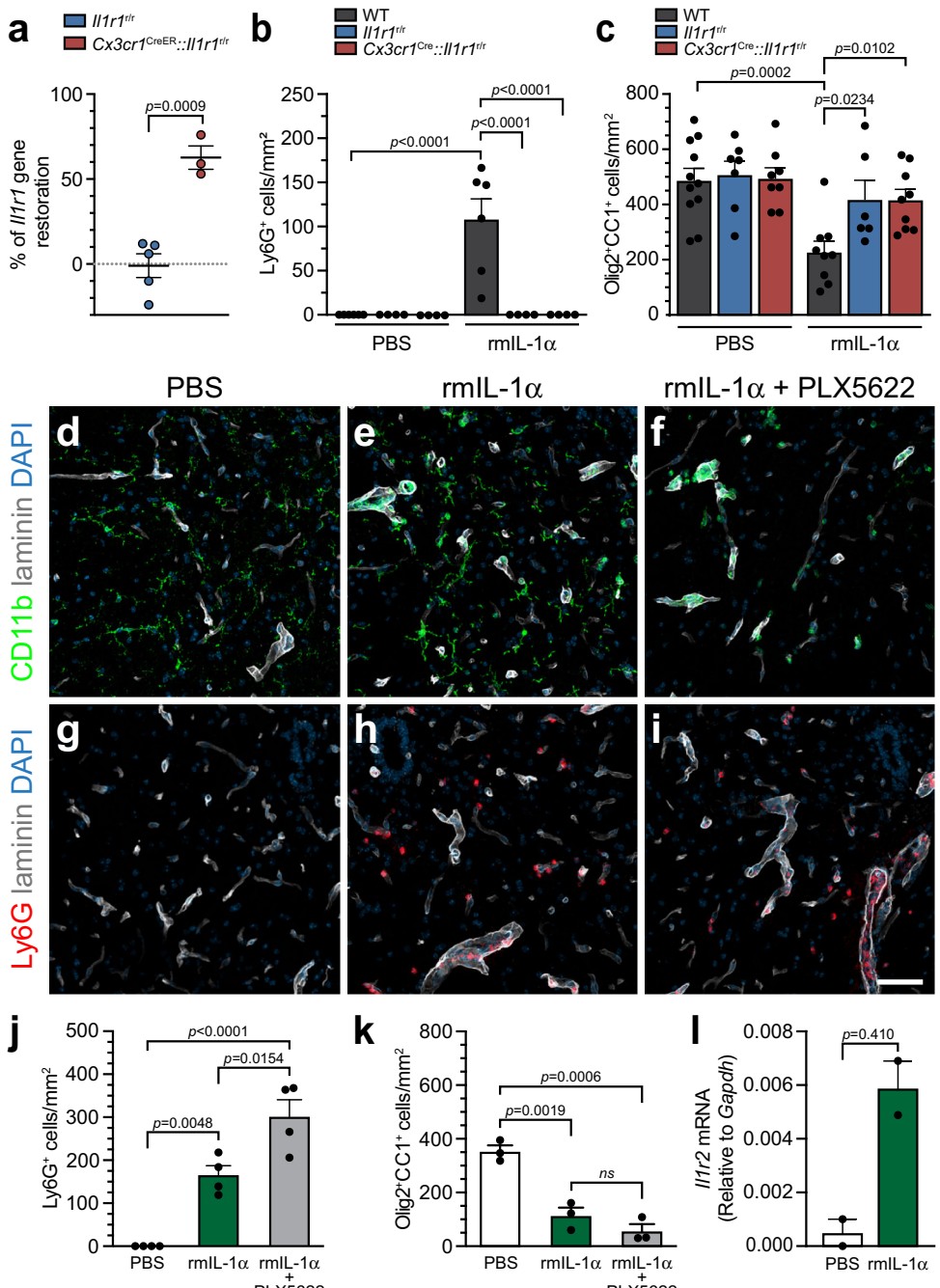

**Fig. 6 | Microglia alleviate IL-1α-mediated neuroinflammation and oligoden-drocyte loss independently of their expression of IL-1R1. a** Quantification of restoration of *Il1r1* gene expression in primary microglia isolated from the unin-jured spinal cord of adult *Il1r1*^r/r mice expressing an IL-1R1-knockout phenotype (set to 0%) and *Cx3cr1*^CreER*::Il1r1*^r/r mice at 30 days post-tamoxifen treatment (*n* = 5 *Il1r1*^r/r, *n* = 3 *Cx3cr1*^CreER*::Il1r1*^r/r). **b** Quantification of the number of Ly6G+ neutrophils that infiltrated the spinal cord of WT, *Il1r1*^r/r and *Cx3cr1*^CreER*::Il1r1*^r/r mice at 24 h post-injection of either PBS or rmIL-1α (*n* = 6 WT + PBS, *n* = 4 *Il1r1*^r/r + PBS, *n* = 4 *Cx3cr1*^CreER*::Il1r1*^r/r + PBS, *n* = 6 WT + rmIL-1α, *n* = 4 *Il1r1*^r/r + rmIL-1α, *n* = 4 *Cx3cr1*^CreER*::Il1r1*^r/r + rmIL-1α). **c** Quantification of the number of Olig2+ CC1+ mature oligodendrocytes in the spinal cord white matter of WT, *Il1r1*^r/r and *Cx3cr1*^CreER*::Il1r1*^r/r mice at 24 h post-injection (*n* = 11 WT + PBS, *n* = 7 *Il1r1*^r/r + PBS, *n* = 8 *Cx3cr1*^CreER*::Il1r1*^r/r + PBS, *n* = 9 WT + rmIL-1α, *n* = 6 *Il1r1*^r/r + rmIL-1α, *n* = 9 *Cx3cr1*^CreER*::Il1r1*^r/r + rmIL-1α). **d–i** Confocal images showing the presence of CD11b+ cells (green cells in

**d–f**; CD11b stains microglia, macrophages and neutrophils) and Ly6G+ neutrophils (red cells, **g–i**) in the spinal cord of C57BL/6 mice injected with either PBS (**d**, **g**), rmIL-1α (**e**, **h**) or rmIL-1α + PLX5622 (**f**, **i**) at 24 h post-injection. **j** Quantification of the number of Ly6G+ neutrophils that infiltrated the spinal cord of C57BL/6 mice at 24 h post-injection of either PBS, rmIL-1α, or rmIL-1α + PLX5622 (*n* = 4 mice/ group). **k** Quantification of the number of Olig2+ CC1+ mature oligodendrocytes in the spinal cord white matter of C57BL/6 mice at 24 h post-injection of either PBS, rmIL-1α, or rmIL-1α + PLX5622 (*n* = 3 mice/group). **l** Relative expression of *Il1r2* gene in mice injected with PBS or rmIL-1α, as determined by quantitative real-time PCR (*n* = 2 per group, where each n is a pool of 4 mice). Data are presented as means +/− SEM and statistical significance determined by either a two-tailed Student's *t*-test (**a**, **l**), one-way ANOVA (**j**, **k**), or two-way ANOVA (**b**, **c**) followed by a Bonferroni post-hoc test. Pairwise comparisons and *p*-values are indicated in the graphs. Scale bars: (**d–i**, in **i**) 50 μm.

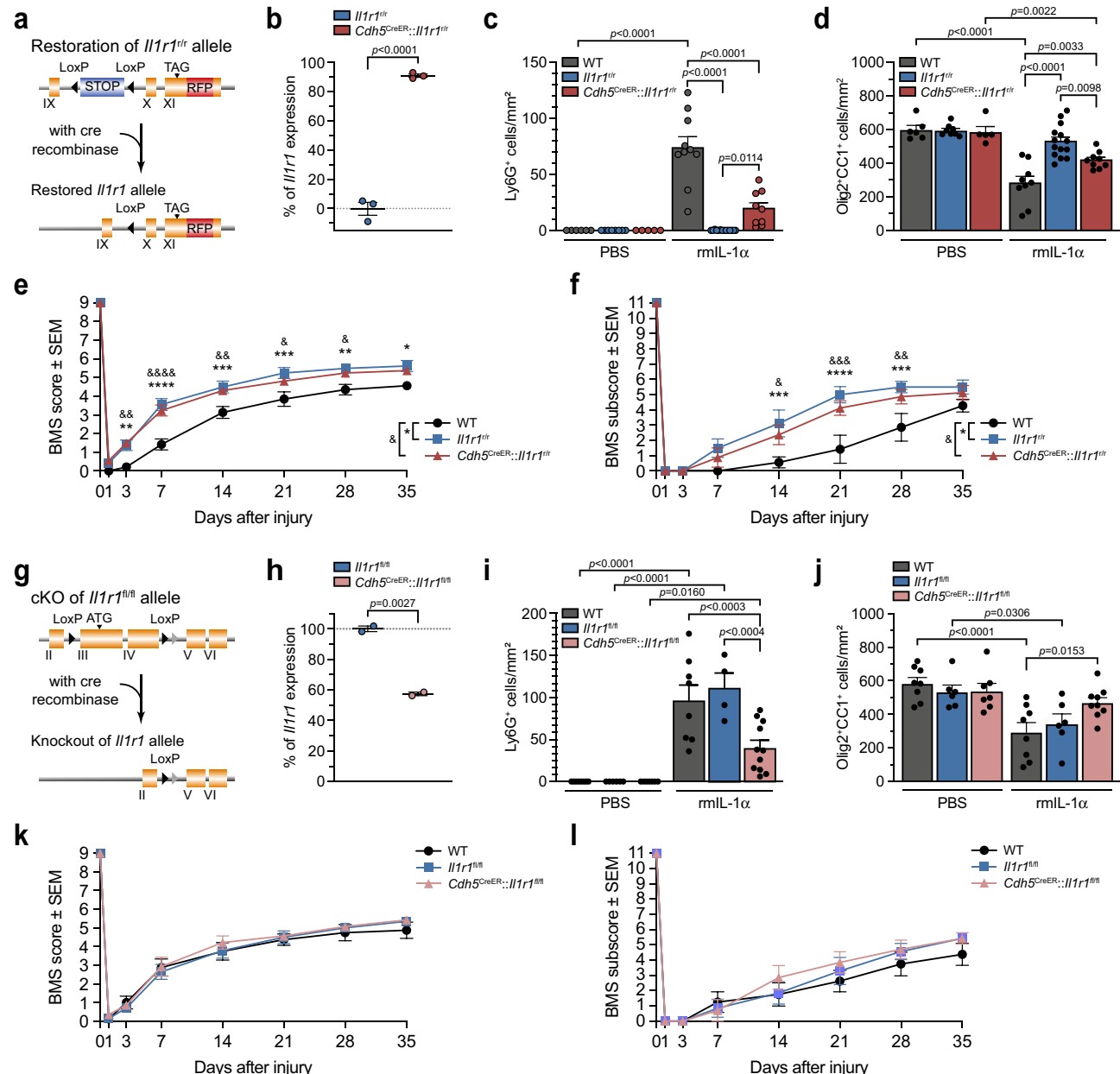

**Fig. 7 | IL-1α-induced neuroinflammation and oligodendrocyte loss is partly mediated by endothelial IL-1R1. a** Genetic design of the $Cdh5^{CreER}::Il1r1^{r/r}$ mouse line. **b** Quantification of $Il1r1$ gene expression in primary brain endothelial cells isolated from adult $Il1r1^{r/r}$ mice expressing an IL-1R1-knockout phenotype (set to 0%) and $Cdh5^{CreER}::Il1r1^{r/r}$ mice at 30 days post-tamoxifen treatment ($n = 3$ mice/group). **c** Quantification of Ly6G+ neutrophils in the spinal cord of WT, $Il1r1^{r/r}$ and $Cdh5^{CreER}::Il1r1^{r/r}$ mice at 24 h post-i.c.m. injection of PBS or rmIL-1α ($n = 6$ WT + PBS, $n = 8$ $Il1r1^{r/r}$ + PBS, $n = 5$ $Cdh5^{CreER}::Il1r1^{r/r}$ + PBS, $n = 10$ WT + rmIL-1α, $n = 14$ $Il1r1^{r/r}$ + rmIL-1α, $n = 9$ $Cdh5^{CreER}::Il1r1^{r/r}$ + rmIL-1α). **d** Quantification of Olig2+ CC1+ mature oligodendrocytes in the spinal cord at 24 h post-injection ($n = 6$ WT + PBS, $n = 8$ $Il1r1^{r/r}$ + PBS, $n = 5$ $Cdh5^{CreER}::Il1r1^{r/r}$ + PBS, $n = 9$ WT + rmIL-1α, $n = 14$ $Il1r1^{r/r}$ + rmIL-1α, $n = 9$ $Cdh5^{CreER}::Il1r1^{r/r}$ + rmIL-1α). **e, f** Locomotor function was assessed using the BMS score (**e**) and BMS subscore (**f**) after SCI ($n = 8$ mice/group). **g** Genetic design of the $Cdh5^{CreER}::Il1r1^{fl/fl}$ mouse line. **h** $Il1r1$ gene expression in brain endothelial cells

of adult $Il1r1^{fl/fl}$ mice, which normally express the $Il1r1$ gene (set to 100%), and $Cdh5^{CreER}::Il1r1^{fl/fl}$ mice at 30 days post-tamoxifen ($n = 2$ mice/group). **i** Quantification of spinal cord-infiltrated neutrophils at 24 h post-injection ($n = 8$ WT + PBS, $n = 6$ $Il1r1^{fl/fl}$ + PBS, $n = 7$ $Cdh5^{CreER}::Il1r1^{fl/fl}$ + PBS, $n = 8$ WT + rmIL-1α, $n = 4$ $Il1r1^{fl/fl}$ + rmIL-1α, $n = 11$ $Cdh5^{CreER}::Il1r1^{fl/fl}$ + rmIL-1α). **j** Quantification of mature oligodendrocytes at 24 h post-injection ($n = 8$ WT + PBS, $n = 6$ $Il1r1^{fl/fl}$ + PBS, $n = 7$ $Cdh5^{CreER}::Il1r1^{fl/fl}$ + PBS, $n = 8$ WT + rmIL-1α, $n = 6$ $Il1r1^{fl/fl}$ + rmIL-1α, $n = 9$ $Cdh5^{CreER}::Il1r1^{fl/fl}$ + rmIL-1α). **k, l** BMS scores (**k**) and subscores (**l**) after SCI ($n = 7$–8 mice/group). Data are means +/− SEM and statistical significance was determined by a two-tailed Student's $t$-test (**b, h**), two-way ANOVA (**c, d, I, j**), or two-way repeated measures ANOVA (**e, f, k, l**) followed by Bonferroni post-hoc test. **e, f** ****$p < 0.0001$, ***$p < 0.001$, **$p < 0.01$, *$p < 0.05$; &&&&$p < 0.0001$, &&&$p < 0.001$, &&$p < 0.01$, &$p < 0.05$. Other pairwise comparisons and $p$-values are indicated in graphs.

$Cdh5^{CreER}::Il1r1^{fl/fl}$ mice. In these mice, $Il1r1$ gene expression was reduced by nearly half in CNS ECs when compared to $Il1r1^{fl/fl}$ control mice (Fig. 7g, h). Further, conditional knockout (cKO) of IL-1R1 in CNS ECs resulted in decreased recruitment and infiltration of neutrophils in the spinal cord of mice at 24 h post-i.c.m. injection of IL-1α (Fig. 7i & Supplementary Fig. 5e, f), proving that endothelial IL-1R1

signaling is relevant in IL-1α mediated neuroinflammation. Further, the IL-1α-mediated decrease in the number of mature spinal cord OLs was prevented by genetic deletion of the $Il1r1$ gene in $Cdh5^{CreER}::Il1r1^{fl/fl}$ mice (Supplementary Fig. 5g, h). We counted on average $538.9 \pm 45.6$ Olig2+CC1+cells/mm² in the spinal cord white matter of $Cdh5^{CreER}::Il1r1^{fl/fl}$ mice injected with PBS compared to

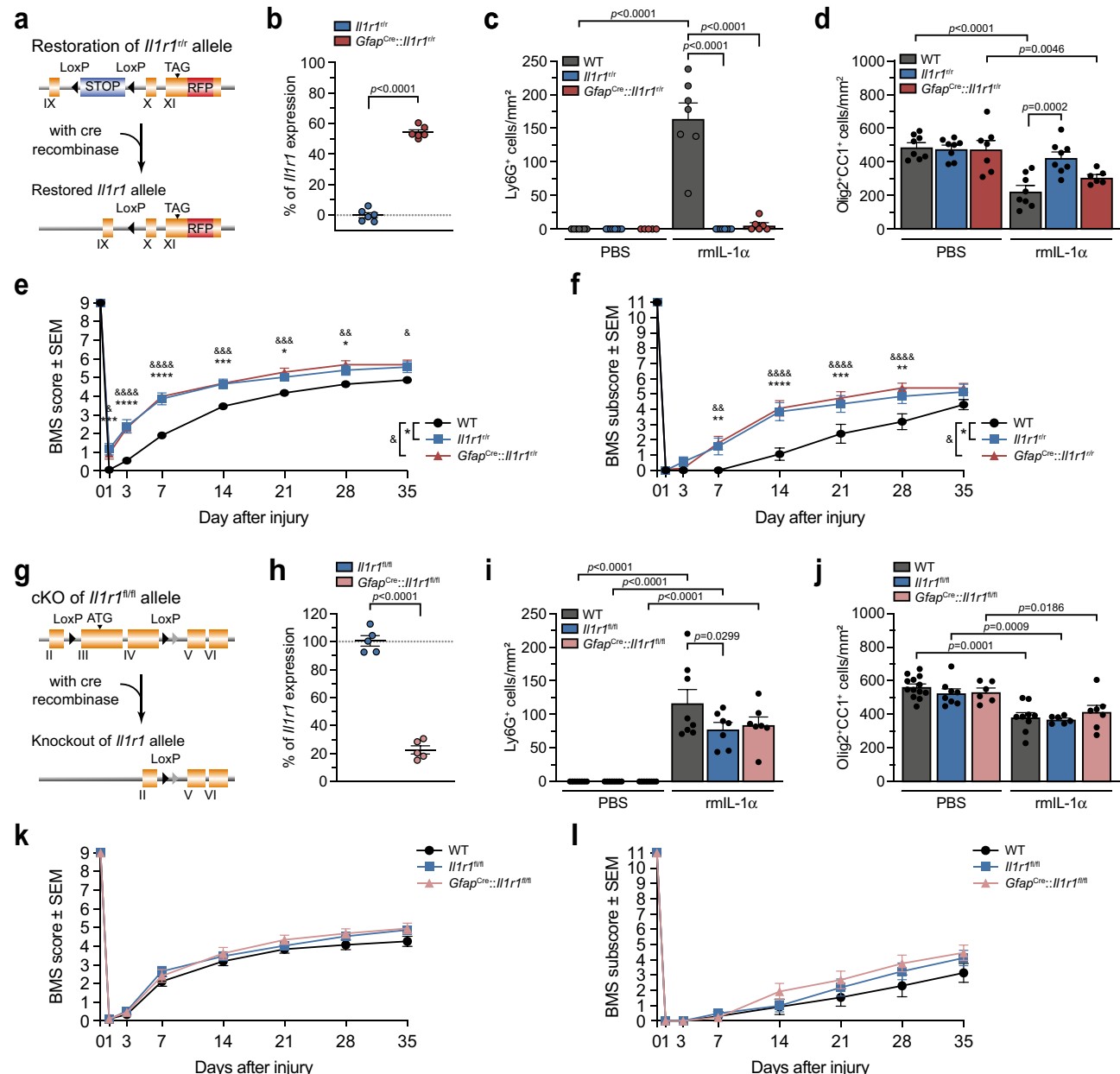

**Fig. 8 | IL-1α-induced neuroinflammation and oligodendrocyte loss is partly mediated by astrocytic IL-1R1. a** Genetic design of the *Gfap*^Cre^::*Il1r1*^r/r^ mouse line. **b** Quantification of *Il1r1* gene expression in primary brain astrocytes from adult *Il1r1*^r/r^ mice expressing an IL-1R1-knockout phenotype (set to 0%) and *Gfap*^Cre^::*Il1r1*^r/r^ mice (*n* = 6 mice/group). **c** Quantification of Ly6G⁺ neutrophils that infiltrated the spinal cord of WT, *Il1r1*^r/r^ and *Gfap*^Cre^::*Il1r1*^r/r^ mice at 24 h post-i.c.m. injection of PBS or rmIL-1α (*n* = 7 WT + PBS, *n* = 8 *Il1r1*^r/r^ + PBS, *n* = 5 *Gfap*^Cre^::*Il1r1*^r/r^ + PBS, *n* = 7 WT + rmIL-1α, *n* = 8 *Il1r1*^r/r^ + rmIL-1α, *n* = 6 *Gfap*^Cre^::*Il1r1*^r/r^ + rmIL-1α). **d** Quantification of Olig2⁺ CC1⁺ mature oligodendrocytes in the spinal cord at 24 h post-injection (*n* = 8 WT + PBS, *n* = 8 *Il1r1*^r/r^ + PBS, *n* = 7 *Gfap*^Cre^::*Il1r1*^r/r^ + PBS, *n* = 8 WT + rmIL-1α, *n* = 8 *Il1r1*^r/r^ + rmIL-1α, *n* = 6 *Gfap*^Cre^::*Il1r1*^r/r^ + rmIL-1α). **e, f** Locomotor function was assessed using the BMS score (**e**) and BMS subscore (**f**) after SCI (*n* = 16 WT, *n* = 15 *Il1r1*^r/r^, *n* = 15 *Gfap*^Cre^::*Il1r1*^r/r^). **g** Genetic design of the *Gfap*^Cre^::*Il1r1*^fl/fl^ mouse line. **h** *Il1r1* gene expression in primary brain astrocytes isolated from adult *Il1r1*^fl/fl^ mice, which normally express the *Il1r1* gene (set to 100%), and *Gfap*^Cre^::*Il1r1*^fl/fl^ mice (*n* = 5 mice/group). **i** Quantification of spinal cord-infiltrated neutrophils at 24 h post-injection (*n* = 8 WT + PBS, *n* = 8 *Il1r1*^fl/fl^ + PBS, *n* = 8 *Gfap*^Cre^::*Il1r1*^fl/fl^ + PBS, *n* = 8 WT + rmIL-1α, *n* = 7 *Il1r1*^fl/fl^ + rmIL-1α, *n* = 7 *Gfap*^Cre^::*Il1r1*^fl/fl^ + rmIL-1α). **j** Quantification of mature OLs at 24 h post-injection (*n* = 13 WT + PBS, *n* = 8 *Il1r1*^fl/fl^ + PBS, *n* = 6 *Gfap*^Cre^::*Il1r1*^fl/fl^ + PBS, *n* = 9 WT + rmIL-1α, *n* = 6 *Il1r1*^fl/fl^ + rmIL-1α, *n* = 7 *Gfap*^Cre^::*Il1r1*^fl/fl^ + rmIL-1α). **k, l** BMS scores (**k**) and subscores (**l**) after SCI (*n* = 13 WT, *n* = 16 *Il1r1*^fl/fl^, *n* = 13 *Gfap*^Cre^::*Il1r1*^fl/fl^). All data are means +/− SEM and statistical significance was determined by a two-tailed Student's *t*-test (**b, h**), two-way ANOVA (**c, d, i, j**), or two-way repeated measures ANOVA (**e, f, k, l**) followed by Bonferroni post-hoc test. **e, f** ****$p < 0.0001$, ***$p < 0.001$, **$p < 0.01$, *$p < 0.05$; &&&&$p < 0.0001$, &&&$p < 0.001$, &&$p < 0.01$, &$p < 0.05$. Other pairwise comparisons and *p*-values are indicated in graphs.

469.2 ± 30.7 Olig2⁺CC1⁺cells/mm² in those injected with IL-1α (Fig. 7j). This decrease was not statistically significant, and did not recapitulate the magnitude of the OL cell loss detected in the two control mouse lines in response to IL-1α (i.e., 45–50% decrease). These loss-of-function effects caused by cKO of IL-1R1 in CNS ECs did not translate into any changes in terms of locomotor recovery (Fig. 7k, l). Thus, endothelial IL-1R1 signaling is at least partly responsible for the infiltration of neutrophils and the death of mature OLs induced by IL-1α.

## IL-1α induces OL cell death in part through activation of IL-1R1 signaling in astrocytes

In light of recent studies reporting on the existence of neurotoxic A1 astrocytes in various CNS disorders[26,49], we next investigated the

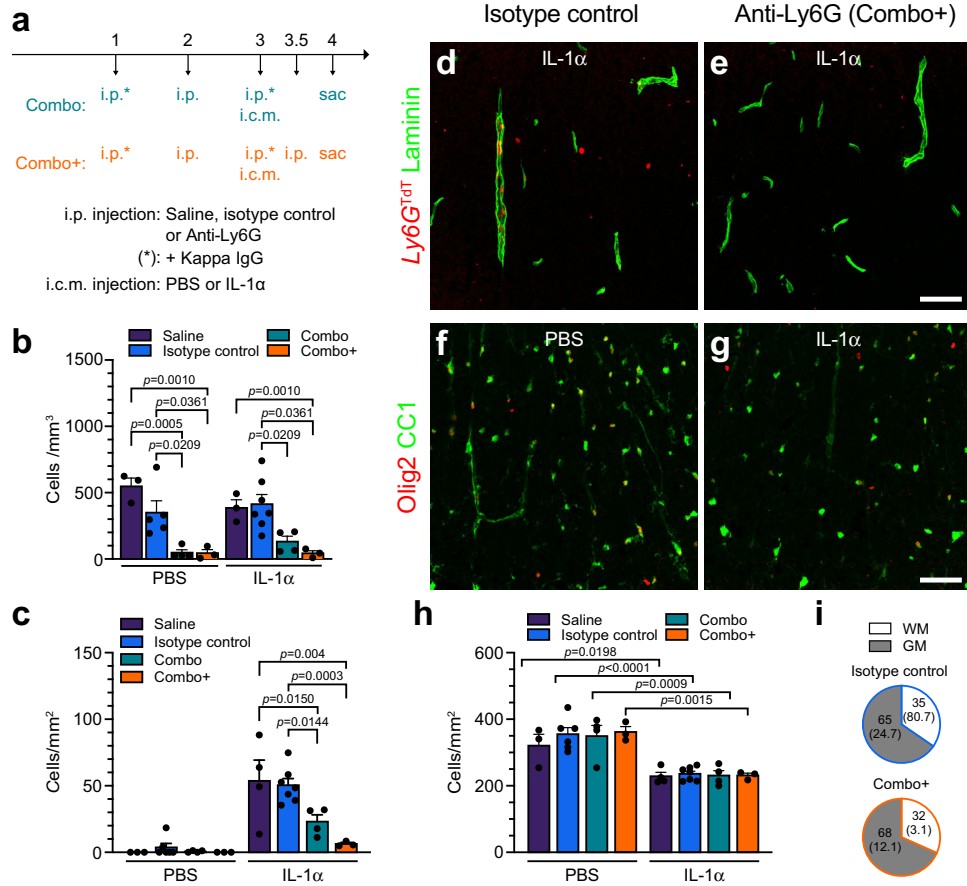

**Fig. 9 | Depletion of neutrophils does not alter IL-1α-mediated oligodendrocyte loss. a** Diagram showing the experimental design and timeline of the depletion study. Abbreviations: Combo, combination strategy; Combo⁺, adapted combination strategy; i.c.m., intra-cisterna magna; i.p., intraperitoneal; sac, sacrifice. **b** Quantification by flow cytometry of the number of Ly6Gᵀᵈᵀ⁺ neutrophils in the blood of PBS- and IL-1α-injected *Ly6g*^Cre-TdT^::*R26-TdT* mice pretreated with either the Combo⁺ strategy, Combo strategy, isotype control antibodies or saline. Mice were killed at 24 h post-i.c.m. injection (*n* = 3 Saline + PBS, *n* = 5 Isotype control + PBS, *n* = 4 Combo + PBS, *n* = 3 Combo⁺ + PBS, *n* = 3 Saline + IL-1α, *n* = 7 Isotype control + IL-1α, *n* = 4 Combo + IL-1α, *n* = 3 Combo⁺ + IL-1α). **c** Quantification of Ly6Gᵀᵈᵀ⁺ neutrophils that infiltrated the spinal cord at 24 h post-i.c.m. injection (*n* = 3 Saline + PBS, *n* = 6 Isotype control + PBS, *n* = 4 Combo + PBS, *n* = 3 Combo⁺ + PBS, *n* = 4 Saline + IL-1α, *n* = 7 Isotype control + IL-1α, *n* = 4 Combo + IL-1α, *n* = 3 Combo⁺ + IL-1α). **d, e** Confocal images showing Ly6Gᵀᵈᵀ⁺ cells (red) and laminin (green) in the

spinal cord of *Ly6g*^Cre-TdT^::*R26-TdT* mice injected i.c.m. with rmIL-1α, and pretreated i.p. with either isotype control antibodies (**d**) or the Combo⁺ treatment (**e**). **f, g** Confocal immunofluorescence imaging of Olig2 and CC1 in the spinal cord white matter of *Ly6g*^Cre-TdT^::*R26-TdT* mice injected i.c.m. with either IL-1α or PBS, and treated with either isotype control antibodies (**f**) or the Combo⁺ strategy (**g**). **h** Quantification of Olig2⁺ CC1⁺ mature OLs in the spinal cord white matter of mice (*n* = 3 Saline + PBS, *n* = 6 Isotype control + PBS, *n* = 4 Combo + PBS, *n* = 3 Combo⁺ + PBS, *n* = 4 Saline + IL-1α, *n* = 7 Isotype control + IL-1α, *n* = 4 Combo + IL-1α, *n* = 3 Combo⁺ + IL-1α). **i** Percentage and total number (in parentheses) of Ly6Gᵀᵈᵀ⁺ neutrophils in the spinal cord gray matter (GM) versus white matter (WM). All data are mean values +/− SEM and statistical significance was determined by a two-way ANOVA followed by Bonferroni post-hoc test (**b, c, h**). Pairwise comparisons and p-values are indicated in graphs. Scale bars: (**d, e**, in **e**) 50 μm, (**f, g**, in **g**) 50 μm.

contribution of astrocytic IL-1R1 signaling in IL-1α-mediated OL cell death. To examine this, *Gfap*^Cre^ mice were crossed with either *Il1r1*^r/r^ or *Il1r1*^fl/fl^ mice to restore or delete, respectively, IL-1R1 expression in astrocytes (Fig. 8). In brain astrocytes of adult *Gfap*^Cre^::*Il1r1*^r/r^ mice, *Il1r1* gene expression was restored to approximately 60% of normal levels (Fig. 8b). This level of restoration of IL-1R1 expression in astrocytes was not sufficient to reestablish IL-1α-mediated neutrophil infiltration, as virtually no Ly6G⁺ cells were observed in the spinal cord of *Gfap*^Cre^::*Il1r1*^r/r^ mice at 24 h post-injection (Fig. 8c & Supplementary Fig. 7a, b). Despite the absence of neutrophil recruitment, this level of IL-1R1 restoration in astrocytes partially reinstituted OL cell death in the presence of IL-1α injection (Supplementary Fig. 7c, d). In *Gfap*^Cre^::*Il1r1*^r/r^ mice, the number of mature OLs decreased by 35% following central injection of IL-1α, passing from 476.4 ± 50.8 to 308.2 ± 16.6 Olig2⁺CC1⁺ cells/mm² (Fig. 8d). In comparison, 55% of the mature OL population died in response to central delivery of IL-1α in WT mice, passing from 489.0 ± 225.1 to 225.6 ± 34.0 Olig2⁺CC1⁺cells/mm² in the thoracic spinal cord white matter. While WT mice performed

significantly worse than IL-1R1-deficient (*Il1r1*^r/r^) mice on the BMS open-field locomotor scale after SCI, *Gfap*^Cre^::*Il1r1*^r/r^ mice performed identical to *Il1r1*^r/r^ mice (Fig. 8e, f). Most of these observations were confirmed via another transgenic mouse model, namely *Gfap*^Cre^::*Il1r1*^fl/fl^ mice (Fig. 8g). In agreement with previous findings, knockout of 80% of *Il1r1* expression in astrocytes failed to reduce neutrophil infiltration (Fig. 8h, i & Supplementary Fig. 7e, f). However, partial deletion of IL-1R1 expression in astrocytes of IL-1α-treated *Gfap*^Cre^::*Il1r1*^fl/fl^ mice did not restore OL cell counts back to normal levels, i.e., those seen in PBS-treated animals (Fig. 8j & Supplementary Fig. 7g, h). Indeed, despite a slightly increased trend in OL viability in the cKO mouse line compared to control groups after i.c.m. injection of IL-1α, these results did not reach significance. Moreover, partial deletion of the *Il1r1* gene in astrocytes alone did not promote functional recovery in the context of SCI (Fig. 8k, l). Therefore, combined targeting of both astrocytic and endothelial IL-1R1, or their common downstream effector molecules, could prove synergistic and produce improved outcomes in terms of preventing secondary degeneration and locomotor deficits after SCI.

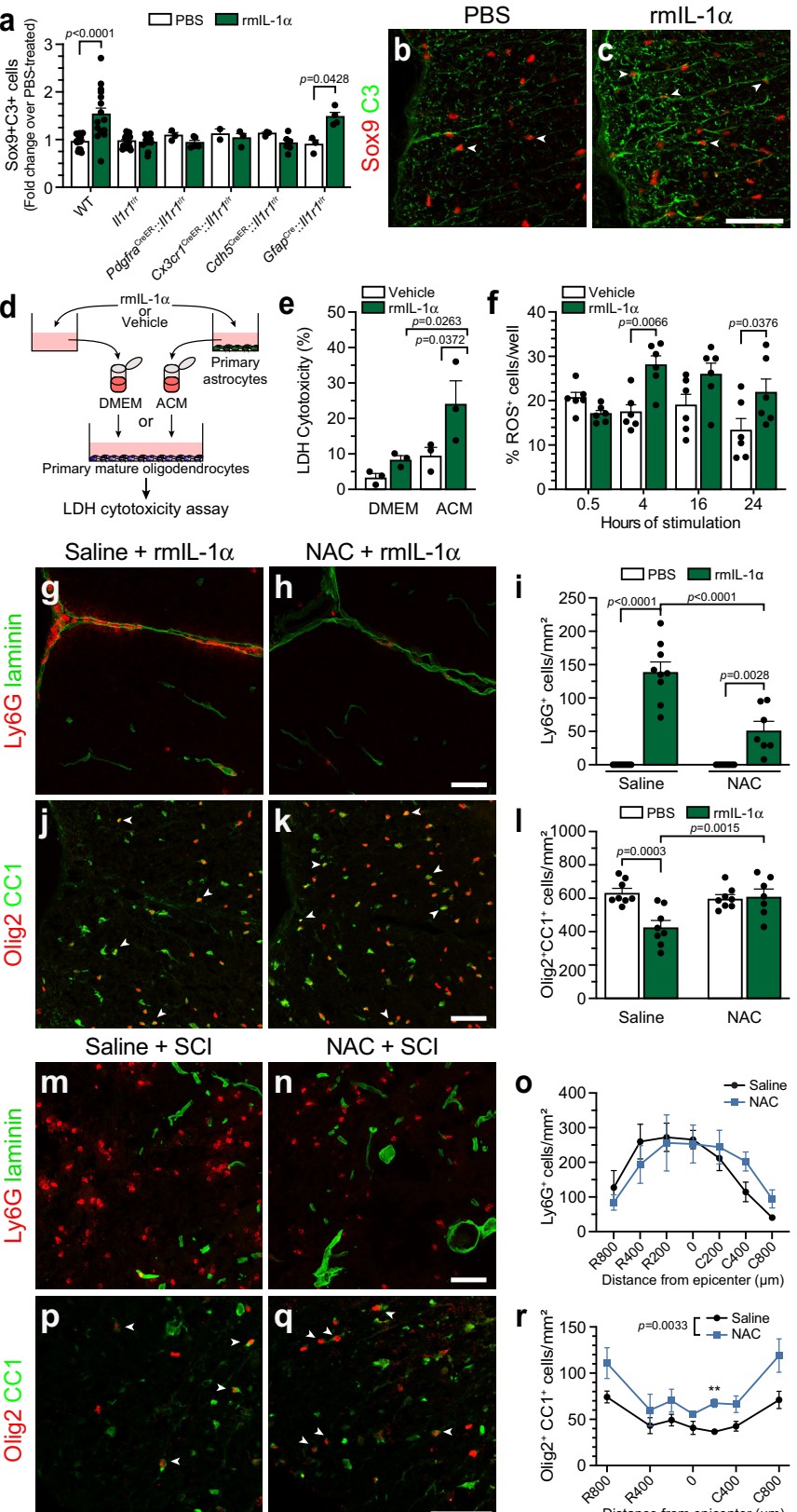

## Neutrophils are not responsible for the killing of mature OLs

To determine whether neutrophils play a role in IL-1α-mediated OL loss, we took advantage of the optimized neutrophil-specific "Combo" depletion strategy, recently developed by Boivin and colleagues[50]. As shown in Supplementary Fig. 2, within 4 h of i.c.m. injection of IL-1α, several proliferative and chemoattractant factors targeting neutrophils are released in the blood. To account for the depletion of newly formed neutrophils, an adapted version of the Combo strategy, consisting of an extra dose of the anti-Ly6G antibody at 12 h post-IL-1α administration, hereafter referred to as "Combo+", was also tested (Fig. 9a). Seeing as treatment with the anti-Ly6G antibody results in the masking of surface Ly6G antigens, thus preventing their subsequent

**Fig. 10 | Reactive oxygen species released by astrocytes in response to IL-1α induce oligodendrocyte death. a** Quantification of Sox9⁺C3⁺ astrocytes in the spinal cord of IL-1R1-deficient (*Il1r1*ʳ/ʳ) and cell-specific IL-1R1 conditional restored mice injected i.c.m. with PBS or rmIL-1α and killed at day 1 (*n* = 15 WT + PBS, *n* = 16 WT + rmIL-1α, *n* = 13 *Il1r1*ʳ/ʳ + PBS, *n* = 11 *Il1r1*ʳ/ʳ + rmIL-1α, *n* = 3 *Pdgfra*ᶜʳᵉᴱᴿ::*Il1r1*ʳ/ʳ + PBS, *n* = 5 *Pdgfra*ᶜʳᵉᴱᴿ::*Il1r1*ʳ/ʳ + rmIL-1α, *n* = 2 *Cx3cr1*ᶜʳᵉᴱᴿ::*Il1r1*ʳ/ʳ + PBS, *n* = 3 *Cx3cr1*ᶜʳᵉᴱᴿ::*Il1r1*ʳ/ʳ + rmIL-1α, *n* = 3 *Cdh5*ᶜʳᵉᴱᴿ::*Il1r1*ʳ/ʳ + PBS, *n* = 6 *Cdh5*ᶜʳᵉᴱᴿ::*Il1r1*ʳ/ʳ + rmIL-1α, *n* = 3 *Gfap*ᶜʳᵉ::*Il1r1*ʳ/ʳ + PBS, *n* = 4 *Gfap*ᶜʳᵉ::*Il1r1*ʳ/ʳ + rmIL-1α). **b, c** Sox9 (red) and C3 (green) immunostainings in the spinal cord of mice injected with PBS or rmIL-1α. White arrowheads point to double-labeled cells. **d** Experimental design for the lactate dehydrogenase (LDH) assay. Primary astrocytes were cultured in presence of PBS (vehicle) or rmIL-1α. Primary mature OLs were then incubated in DMEM containing (or not) rmIL-1α, or conditioned medium derived from astrocytes (ACM) stimulated with vehicle or IL-1α. **e** Quantification of OL loss using LDH assay (*n* = 3 wells/condition). **f** Quantification of reactive oxygen species (ROS) production in primary astrocytes stimulated with vehicle or rmIL-1α (*n* = 6 wells/ condition). **g, h** Immunofluorescence showing Ly6G⁺ neutrophils (red) in the spinal cord of C57BL/6 mice injected i.c.m. with rmIL-1α and i.p. with N-acetyl-L-cysteine (NAC) or saline. Mice were killed at 24 h post-i.c.m. injection. **i** Quantification of spinal cord-infiltrated Ly6G⁺ neutrophils (n = 8 Saline+PBS, n = 9 Saline+rmIL-1α, *n* = 8 NAC + PBS, *n* = 7 NAC + rmIL-1α). **j, k** Confocal immunofluorescence showing Olig2 (red) and CC1 (green) in the spinal cord of mice injected i.c.m. with rmIL-1α and i.p. with NAC or saline at 24 h. **l** Quantification of mature OLs in the spinal cord (*n* = 8 mice/group). **m, n** Immunostaining of Ly6G⁺ neutrophils (red) at the lesion epicenter at day 1 post-SCI in C57BL/6 mice treated with NAC or saline. **o** Quantification of spinal cord-infiltrated neutrophils (*n* = 5 mice/group). **p, q** Immunostaining for Olig2 (red) and CC1 (green) at the lesion epicenter at day 1 post-SCI. **r** Quantification of mature OLs in the injured spinal cord (*n* = 5 mice/ group). Data are means (+/−SEM) and statistical significance determined by two-way ANOVA followed by Bonferroni post-hoc test (**a, e, f, i, l, o, r**). **\*\****p* < 0.01 (**r**). Other pairwise comparisons and *p*-values are indicated in graphs. All scale bars: 50 μm.

detection by flow cytometry or immunofluorescence, all neutrophil depletion experiments were performed in *Ly6g*ᶜʳᵉ⁻ᵀᵈᵀ::*R26-TdT* transgenic mice expressing the TdT fluorescent protein specifically in neutrophils. As shown in Fig. 9b, the Combo⁺ strategy led to an eradication of >95% of Ly6G-TdT⁺ blood neutrophils compared to control groups at 24 h post-injection of IL-1α. Likewise, the total number of neutrophils was reduced by ~90% and ~85% in the spinal cord of *Ly6g*ᶜʳᵉ⁻ᵀᵈᵀ::*R26-TdT* mice injected with IL-1α and treated with the Combo⁺ strategy, compared to those treated with either saline or the isotype control antibody (Fig. 9c–e). The original Combo treatment reduced the total number of spinal cord-infiltrating neutrophils by only 55% compared to controls. Despite the near complete elimination of infiltrating neutrophils in the spinal cord of IL-1α-injected mice, the Combo⁺ treatment did not prevent OL loss (Fig. 9f–h). Furthermore, we discovered that ~70% of the few remaining neutrophils were localized in the spinal cord gray matter, leaving on average only 3.1 ± 1.1 neutrophils/mm² in the spinal cord white matter where we quantified Olig2⁺ CC1⁺ OLs (Fig. 9i). Altogether, these results suggest that neutrophils do not play a role in the death of mature spinal cord OLs observed after central IL-1α administration.

### IL-1α induces a reactive astrocyte phenotype that leads to the killing of mature OLs through the release of reactive oxygen species

The complement component 3 (C3) protein was recently shown to be upregulated in a subtype of reactive astrocytes observed to have potentially neurotoxic effects in various CNS disorders[26]. We have thus examined whether central administration of rmIL-1α increases C3 expression in spinal cord astrocytes. We found that i.c.m. delivery of IL-1α in C57BL/6 mice at day 1 post-injection increased the total number of Sox9⁺ cells expressing C3 by nearly two-fold, an effect that was only replicated by the restoration of IL-1R1 expression specifically in astrocytes and not other cell types (Fig. 10a–c). This shows that IL-1α is a potent inducer of astrocytic C3 upregulation, likely associated with a toxic reactive phenotype. Next, we performed cell culture experiments using primary murine astrocytes and OLs to determine whether OL death is induced by release of toxic factors via astrocytes upon IL-1α stimulation, rather than toxicity induced through cell contact mediated mechanisms (Fig. 10d). Consistent with previous data demonstrating that i.c.m. delivery of IL-1α to *Pdgfra*ᶜʳᵉᴱᴿ::*Il1r1*ʳ/ʳ mice did not result in death of mature spinal cord OLs (Fig. 5), we found that the addition of rmIL-1α to the control medium resulted in minor, non-significant toxicity for OLs in vitro (Fig. 10e). However, the transfer of conditioned medium from IL-1α-stimulated astrocytes to cultured primary OLs was sufficient in evoking their death, as assessed using the LDH cytotoxicity assay. No significant cytotoxicity was detected on primary mature OLs when incubated with astrocyte-conditioned medium from untreated astrocytes. This indicates that even if IL-1α

exposure might result in low level OL death, IL-1α-stimulated astrocytes clearly release factors that are lethal to mature OLs.

We next aimed to determine the identity of the molecules harming OLs. Considering the susceptibility of OLs and neurons (axons) to reactive oxygen species (ROS)[51,52], we hypothesized that ROS could be implicated in this mechanism of cell death. To assess the potential involvement of ROS in cell death of mature spinal cord OLs, we first examined whether IL-1α can trigger the production of ROS by astrocytes in vitro. For this, we cultured primary murine astrocytes in the presence of either rmIL-1α or vehicle, and then measured the production of ROS at various time points using the CellRox assay. We found that astrocytes begin expressing ROS at 4 h post-treatment with IL-1α, and this ROS production is sustained over the total observation period (i.e., 24 h; Fig. 10f). We then attempted to validate this proposition in vivo using IL-1α-injected C57BL/6 mice either pretreated or not with N-acetyl-L-cysteine (NAC), a potent antagonist of ROS activity. Pretreatment with the NAC compound was sufficient to partially neutralize the effect of IL-1α on neutrophil infiltration (Fig. 10g–i). More importantly, the NAC pretreatment completely prevented the loss of mature spinal cord OLs associated with central IL-1α administration (Fig. 10j–l). Inhibition of ROS may therefore be an effective target to mitigate inflammation and block secondary degeneration of mature OLs after SCI. This hypothesis was further tested in a mouse model of contusion SCI. Although NAC administration the day before and during the first 24 h post-SCI failed to reduce neutrophil infiltration at day 1 after SCI (Fig. 10m–o), it successfully reduced the death of mature OLs caudal to the site of contusion injury when compared to saline-treated animals (Fig. 10p–r). Together, these data show that IL-1α activates astrocytes, resulting in the release of toxic ROS, and contributing to secondary pathogenesis in SCI.

## Discussion

It has long been recognized that the release of DAMPs initiates inflammation following tissue injury, but their identity and effects are just beginning to be understood in the context of the injured CNS. Here, we investigated the role of the alarmin IL-1α in the SCI environment. We found that IL-1α is primarily derived from damaged microglia located in the lesion core, and that peak expression of this alarmin correlates with the death of microglia, occurring within the first 24 h of SCI. Further, deletion of the *Il1a* or *Il1r1* gene in mice not only reduced the infiltration of innate immune cells, but significantly diminished the death of mature OLs at sites of SCI and improved locomotor recovery. Delivery of IL-1α i.c.m. mimicked the profile of cytokines/chemokines released in the plasma post-SCI, triggered the recruitment and infiltration of neutrophils, and led to a dramatic loss of mature OLs along the rostrocaudal axis of the spinal cord. Using sophisticated transgenic mouse lines inducing cell-specific restoration or deletion of the gene coding for IL-1R1 in CNS-resident cell populations, we found that OL

cell death was indirectly mediated and involved both astrocytes and ECs. Last but not least, we revealed that OL loss was effectuated through the release of ROS by IL-1α-stimulated reactive astrocytes.

IL-1α is a proinflammatory cytokine that contributes to inflammation in various disorders through activation of its cell-surface receptor, IL-1R1[53]. In addition, IL-1α is ubiquitously expressed as a precursor protein of about 31 kDa (pro-IL-1α), and is translocated from the cytoplasm to the cell nucleus under inflammatory conditions, where it acts as a proinflammatory activator of transcription[54,55]. Pro-IL-1α does not require proteolytic cleavage to be activated[56], despite the fact that proteolysis increases its biological potency[57]. Accordingly, the rapid release of IL-1α from necrotic cells, but not apoptotic cells following sterile injury to peripheral tissues makes it an ideal DAMP[41,58,59]. We recently reported that IL-1α triggers neuroinflammation after SCI through its production via myeloid cells that either reside in the spinal cord or infiltrate from the bloodstream in response to injury[8]. Taking advantage of Cre-inducible *Cx3cr1*[CreER]*::R26-TdT* and *LysM-eGFP::Cx3cr1*[CreER]*::R26-TdT* reporter mice, we have extended these data to show that dead and damaged microglia, but not monocyte-derived macrophages, are the main source of IL-1α in the early acute phase of SCI.

It is known that IL-1α can bind to at least three specific cell-surface receptors, namely IL-1R1, τIL-1R1 and IL-1R2[46]. However, the identity of the CNS resident cells expressing each type of receptor has been hampered by the low in situ levels of the proteins, and difficulties related to the isolation of cell types purely. Although there is no doubt that IL-1R1 is strongly expressed by CNS ECs and certain types of neurons as shown by immunofluorescence microscopy[20,40], evidence supporting the expression of IL-1 receptors in microglia, astrocytes, OL lineage cells, and perivascular macrophages has often been indirect and linked only to the overall response of these cells to IL-1 treatment[60–63]. Here, the expression of IL-1R1 was analyzed by immunoblotting membrane proteins extracted from specific primary cell cultures derived from the normal mouse brain. Our immunoblotting findings confirm that IL-1R1 or τIL-1R1 is expressed by microglia, astrocytes and OLs, despite our failure to detect this protein by way of immunofluorescence staining using paraformaldehyde (PFA)-fixed brain and spinal cord tissue sections. We interpret this to mean that IL-1R1 is weakly expressed in glial cells. Supporting this, Pinteaux et al. have previously reported that murine microglial cell cultures express low levels of IL-1R1 mRNA, but high levels of the decoy IL-1R2, conferring to these cells a high resistance to IL-1 cytokines[64]. Our data showed that IL-1R2 expression (and thus resistance to IL-1α) is dramatically increased in microglia following i.c.m. injection of IL-1α, which may explain why we failed to detect expression of the transcription factor Fos, a marker of cell activity, in microglia after central administration of IL-1α. In contrast, both astrocytes and mature OLs demonstrated activation throughout the mouse spinal cord as early as 1 h post-injection, pointing to these cells as potential direct or indirect targets of IL-1α.

We previously found that in mice, IL-1α plays a critical role in SCI. First, we showed that IL-1R1/MyD88 signaling is essential for the expression of chemokines CXCL1, CXCL2 and CCL2 by astrocytes and the subsequent recruitment of neutrophils and proinflammatory Ly6C[hi] monocytes at sites of SCI[65]. Accordingly, deficiency in IL-1α led to a reduction in the infiltration of innate immune myeloid cells, reduced lesion volume and improved functional recovery after SCI, effects that correlated with an increased survival of mature OLs[8]. Until now, however, a direct causal effect relationship had yet to be demonstrated, and the in vivo mechanisms underlying the effects of IL-1α on OL loss remained obscure. Our findings shed light on these issues by revealing that IL-1α alone is sufficient in triggering a cascade of cellular and molecular events leading to the rapid death (within 24 h) of nearly 40% of mature OLs in the mouse spinal cord. Using a diphtheria toxin (DT) receptor-based strategy to selectively ablate mature

OLs, Oluich et al. demonstrated that a loss of approximately 25% and 40% of mature OLs in the brain and spinal cord, respectively, resulted in severe clinical dysfunction with an ascending spastic paralysis and ultimately leading to fatal respiratory impairment within 3 weeks of DT administration[66]. In this study, no evidence was found that OPCs compensate for the loss of mature OLs. While two other groups independently demonstrated that genetically-induced death of mature OLs is associated with severe ataxia and a tremor that correlates with signs of demyelination, impaired axonal conduction and even sometimes death, they also found that surviving mice recovered from their neurological deficits and displayed OL replenishment and remyelination[67,68]. The replenishment of the OL population was determined to occur between 5 and 10 weeks following the toxin-mediated death of mature OLs, followed by complete attenuation of motor deficits by week 11[67]. Likewise, targeted laser-induced ablation of a single mature cortical OL resulted in the emergence of a newly matured OL with an approximate turnaround of 11 weeks[69]. This prompted us to investigate in our own model whether OPCs proliferate and differentiate to replace original OLs, and if so, the timing of this process. Strikingly, we found that the number of mature OLs returned to baseline after 3–5 days post-injection of IL-1α. BrdU pulse experiments and Ki67 immunostaining allowed us to confirm both the proliferation of OPCs, and their subsequent differentiation into newly matured Olig2[+] CC1[+] OLs. This rapid turnover is reminiscent of changes observed in the acutely injured spinal cord, where robust replacement of mature OLs by OPCs can be seen[32,70,71]. Although the loss, and ensuing regeneration of OLs from OPCs is present in cases of SCI and i.c.m. IL-1α injection, the timing and scale of these responses may differ between models. This may be because the inflammatory response that develops after SCI is different than after acute central delivery of IL-1α. Unlike in the IL-1α injection model, SCI results in the formation of a fibrotic/glial scar that encompasses the primary damage, but simultaneously prevents spinal cord repair. Still, whether remyelination positively impacts locomotor recovery after SCI remains open for debate[72]. A better understanding of the mechanisms driving the replacement of myelin and OL loss would be very beneficial, contributing to the understanding of various other pathological contexts. Collectively, these data suggest that SCI triggers a local response in the CNS, characterized by the release of alarmins such as IL-1α that in turn induce the death of mature OLs in the first 24 h. Their rapid replacement afterwards by newly matured OLs may allow for greater efficiency in remyelinating the injured spinal cord.

We discovered that concomitant injection of IL-1α and anakinra, an IL-1 receptor antagonist, was sufficient to abolish the effects of IL-1α on neutrophil infiltration and OL death. Accordingly, our data demonstrated that OLs were protected, and functional recovery improved in *Il1a*[−/−] and *Il1r1*[−/−] mice compared to WT mice after SCI. This prompted us to further investigate the mechanisms behind IL-1α-mediated OL loss using various cell-specific IL-1R1 restored and knockout mouse lines. Our in vivo experiments revealed that the effects of IL-1α on neutrophils and mature OLs are indirect and mediated through activation of IL-1R1 signaling in CNS ECs and astrocytes. The idea that endothelial IL-1R1, but not microglial IL-1R1, is necessary for mediating the central effects of IL-1β on sickness behavior and leukocyte recruitment was recently proposed by the Quan laboratory[73]. Moreover, Liddelow and colleagues recently found that microglia-derived cytokines, including IL-1α, determine whether astrocytes will exert toxic or pro-survival effects on differentiated OLs (but not OPCs) and neurons in culture[26]. This implies that astrocytic IL-1R1 may contribute to degeneration in a variety of neurodegenerative diseases and CNS injuries. Under all activation conditions tested, microglia alone were found to be insufficient in inducing neuronal death[26]. Whether microglia are protective or pathologic remains a matter of intense debate, and is most likely context-dependent. Depletion of microglia in mice injected i.c.m. with IL-1α increased

inflammation and OL loss, an effect that we associated with the protective overexpression of the decoy IL-1R2 found in microglia. In a mouse model of contusion SCI, microglia proved to be a key cellular component of the developing scar that protects neural tissue post-SCI[29]. Using a mouse model of ischemic SCI, the neuroprotection conferred by repeated LPS treatment was partly reversed by specific deletion of microglial or endothelial IL-1R1[74].

It is important to keep in mind that despite being a powerful tool to decipher molecular mechanisms in vivo, Cre-reporter mouse lines have some limitations, the most important being that recombination in a specific cell type is often incomplete, particularly in the case of tamoxifen-inducible CreER mice. This is especially limiting when aiming to knockout a gene of interest in a specific cell type in vivo, rather than restoring it. As knockouts are only partial, it cannot be determined with certainty whether the responses observed are due to the remaining receptors expressed by the targeted cell population, or those expressed on another cell types. Here, we circumvented part of this problem by using restored and knockout mouse models along with primary cell lines, thus ensuring an additional level of confidence and reproducibility of our results. Using these mouse lines, we demonstrate that the IL-1α released by injured microglia in SCI triggers neutrophil recruitment and secondary degeneration of OLs through IL-1R1 expression in cells forming the neurovascular unit, namely ECs and astrocytes.

The role of astrocytes in SCI, and more generally in CNS injury and diseases, has recently been a great topic of discussion[27]. While some studies suggest that reactive astrocytes protect the injured spinal cord via their role in glial scar formation[28,75], evidence also suggests that they may inhibit axonal regeneration and exacerbate secondary tissue damage[76–79]. The reality remains that pending the right conditions of activation and/or given the right molecular cues, astrocytes may adopt an axon growth-supporting phenotype along with their contribution to scar formation[80]. Overall, our results support the idea that astrocytes are highly adaptable cells that adjust to their changing microenvironment. When activated by IL-1α, we found that astrocytes produce and release ROS, which in turn induces death of mature OLs. It must be noted, however, that the secondary injury cascade occurring after traumatic SCI is complex and includes various proinflammatory pathways. Our findings regarding the effects of IL-1α likely apply to the first 24 h after the injury, when the cytokine is present. Accordingly, treatment of mice with the antioxidant NAC prevented the loss of OLs induced in cases of i.c.m. delivery of IL-1α and SCI. The idea that cells of OL lineage are particularly vulnerable to oxidative damage after CNS injury has been suggested before[81]. Some studies have reported improved functional recovery in NAC-treated rodents after traumatic brain or spinal cord injury[82–84]. Collectively, these findings suggest that IL-1α-activated astrocytes release toxic ROS that play a crucial role in secondary degeneration of OLs after SCI. It must be noted, however, that the secondary injury cascade occurring after traumatic SCI is complex and likely involves multiple pathways.

Recently, Liddelow and colleagues described a subtype of reactive astrocytes, termed A1 astrocytes, capable of inducing neurotoxicity[26]. More specifically, they found that the supernatant of astrocytes activated by a cocktail of microglia-derived proteins, namely IL-1α, TNF and C1q, was sufficient to induce cell death in primary cultures of neurons and mature OLs, but not OPCs or other CNS cell types. In an LPS model of systemic inflammation, each global knockout mouse line for either IL-1α, TNF or C1q showed a significant decrease in the A1 astrocytic response, but only the triple-knockout ($Il1a^{-/-}$ $Tnf^{-/-}$ $C1qa^{-/-}$) mice had no sign of astrocyte reactivity. They also found that reactive astrocytes drive the death of retinal ganglion cells in cases when the optic nerve is crushed and in the bead occlusion model of glaucoma, an effect that was prevented in $Il1a^{-/-}$ $Tnf^{-/-}$ $C1qa^{-/-}$ triple knockout mice[85]. Since TNF and C1q are also produced at sites of SCI during the

early acute phase[86,87], it is possible that the loss of mature OLs detected in our transgenic mouse models would have been further prevented by targeting these three genes at the same time. In future work, it may be interesting to validate the relevance of these immune molecules and their receptors, individually or in combination, in animal models of SCI using cell-specific conditional gene targeting strategies. Likewise, it will be important to identify other downstream effector molecules like ROS that directly mediate CNS cell toxicity, as the therapeutic efficacy of targeting a single molecule may be limited in the complex secondary damage following CNS injury. Supporting this line of research is a recent study by Guttenplan et al., revealing that long-chain saturated lipids partially contribute to astrocyte-mediated toxicity in vivo in the retinal injury model[88].

Aside from reactive astrocytes, another possible source of ROS could be the innate immune cells infiltrating the spinal cord during the early acute phase of SCI. Activated neutrophils have long been recognized to produce large amounts of ROS that may, for example, increase vascular permeability and incite neutrophil infiltration into tissues[89]. Neutrophils have also been suspected to promote neurotoxicity through the release of ROS, TNF and proteases[90]. Of relevance here is the fact that neutrophils acquire neurotoxic properties following transmigration across the IL-1-stimulated CNS endothelium[91]. To definitively answer the question of whether neutrophils have contributed to OL loss in response to central IL-1α delivery, we resorted to a double antibody-based depletion strategy that enhanced neutrophil elimination via anti-Ly6G[50], a finding that we confirmed both in the blood and in SCI tissue using $Ly6g^{Cre-TdT}::R26-TdT$ reporter mice. Another key finding in our study shows that neutrophils are not implicated in the OL loss observed after i.c.m. injection of IL-1α. Further supporting this statement is our discovery that restoring IL-1R1 expression in astrocytes of IL-1α-injected $Gfap^{Cre}::Il1r1^{r/r}$ mice restored OL cell death comparable to WT levels, yet did not restore the infiltration of neutrophils. Taken together with our previous work, this study suggests that endothelial IL-1R1 signaling is required for neutrophil recruitment in the spinal cord under sterile inflammatory conditions, but that these innate immune cells do not contribute to OL loss in the context of neuroinflammation. Instead, astrocytes and CNS ECs appear to drive OL cell death via an IL-1R1-dependent release of ROS in the first 24 h, and other pathways that have yet to be identified in the post-acute phase.

SCI generates an extensive activation of several proinflammatory pathways, many of which do not resolve over the course of the pathology[92]. The administration of IL-1α i.c.m. allowed us to study the effects of this alarmin in the spinal cord, and to replicate histopathological markers up to 24 h after the injection, which is when this cytokine is present. Although this model is appropriate for studying the function of IL-1α, the administration of a single dose of this cytokine is not enough to replicate the chronicity and complexity of inflammation and the physical outcomes present in SCI. Future investigations will be necessary to know the effects of the IL-1α/IL-1R1 pathway in later stages of SCI, along with its role in injury when other complementary DAMPs and inflammatory pathways are triggered.

In summary, our results show that the alarmin IL-1α produced by damaged microglia drives neuroinflammation and OL loss after SCI. These responses are mediated in part through activation of endothelial and astrocytic IL-1R1 signaling, resulting in the release of toxic ROS. IL-1α inhibition may therefore prove to be a valuable strategy in preventing secondary degeneration after SCI.

## Methods

### Mice

A total of 571 mice of both sexes were used in this study. Male and/or female C57BL/6 mice were purchased from Charles River Laboratories or The Jackson Laboratory (JAX) at 8–10 weeks of age. $Cx3cr1^{CreER}$ mice

were obtained from the European Mouse Mutant Archive, with prior authorization from Dr. Steffen Jung (Rehovot, Israel), and genotyped as described[93]. Breeders for *Rosa26-tdTomato* (*R26-TdT*, also known as Ai9, stock #007905[94]), *Pdgfra*[CreER] (stock #018280[95]), *Gfap*[Cre] (stock #024098[96]), and *Il1r1*[fl/fl] (stock #028398[97]) mice were all purchased from JAX. *LysM-eGFP* knock-in mice were obtained from Dr. Gregory Dekaban (Robarts Institute, London, ON, Canada), with prior consent of Dr. Thomas Graf (Center for Genomic Regulation, Barcelona, Spain). *Cdh5*[CreER] mice (line #13073), in which the tamoxifen-inducible Cre recombinase is active in all ECs, were purchased from the Cancer Research Technology Repository at Taconic with prior consent of the creator of the mouse line[98], Dr. Ralf Adams (London Research Institute, UK). *Il1r1*[r/r] mice were obtained from Dr. Ning Quan[47]. *Ly6g*[Cre-TdT] mice were provided by Dr. Matthias Gunzer (University of Duisburg-Essen, Essen, Germany) and are described in detail elsewhere[99]. *Cx3cr1*[CreER], *Pdgfra*[CreER], *Gfap*[Cre], *Cdh5*[CreER] and *Ly6g*[Cre-TdT] mice were bred in-house and crossed with *LysM-eGFP, R26-TdT, Il1r1*[r/r] or *Il1r1*[fl/fl] mice at the Animal Facility of the Center de recherche du Center hospitalier universitaire (CRCHU) de Québec–Université Laval. Wild-type littermates generated from heterozygous matings were used as controls. Mice were housed in individually ventilated cages on racks connected to a central HEPA filtered air supply (30–70 air changes per hour). All animals were kept in a standard 12-h light-dark cycle and had free access to food and water at all times. Room temperature was maintained at $23 \pm 2\,°C$ with a relative humidity of $50 \pm 5\%$. All animal procedures were approved by the *Comité de protection des animaux de l'Université Laval* (protocols #CHU-20-675 and #CHU-21-860) and conducted in compliance with relevant ethical regulations and guidelines of the Canadian Council on Animal Care.

### Tamoxifen treatment
To induce Cre-mediated recombination in *Cx3cr1*[CreER]*::R26-TdT*, *Cx3cr1*[CreER]*::Il1r1*[r/r], *Cdh5*[CreER]*::Il1r1*[r/r], and *Cdh5*[CreER]*::Il1r1*[fl/fl] mouse lines, mice were treated orally with 10 mg of tamoxifen (dissolved in 1:10 ethanol/corn oil) twice at 2-day intervals starting at postnatal days (P) 30–32. To induce recombination in *Pdgfra*[CreER]*::Il1r1*[r/r] mice, animals were treated intraperitoneally with 2 mg of tamoxifen twice at 2-day intervals starting at P12–14.

### Spinal cord injury (SCI)
Mice were anesthetized with isoflurane and underwent a laminectomy at vertebral level T9–10, which corresponds to spinal segment T10–11. Briefly, the vertebral column was stabilized and a contusion of 50 kdyn was performed using the Infinite Horizon SCI device (Precision Systems & Instrumentation). Overlying muscular layers were then sutured and cutaneous layers stapled. Post-operatively, animals received manual bladder evacuation twice daily to prevent urinary tract infections.

### Behavioral analysis
Recovery of locomotor function after SCI was quantified in an open field using the BMS, according to the method developed by Basso and colleagues[100]. All groups of mice exhibited similar parameters in terms of impact force and spinal cord tissue displacement prior to BMS testing. All behavioral analyses were done blind with respect to the identity of the animals.

### Intra-cisterna magna injections
Mice were injected i.c.m. with either rmIL-1α (doses ranging from 20 to 100 ng/μL diluted in PBS, 5 μL injected per mouse, Peprotech catalog #211-11 A), rmIL-1β (20 to 100 ng/μL diluted in PBS, 5 μL/mouse, Peprotech catalog #211-11B), anakinra (also known as Kineret®, 100 μg/μL, 5 μL/mouse) or PBS (5 μL/mouse). The i.c.m. treatment consisted of a single injection using a pulled-glass micropipette connected to a 10-μL Hamilton syringe.

### Bromodeoxyuridine injections
To label proliferating cells, mice were intraperitoneally injected once daily with BrdU (50 mg/kg of body weight in 0.9% saline) for 3 consecutive days, starting on day 1 post-i.c.m. injection of either IL-1α or PBS. The third and final injection was performed 4 h prior to sacrifice.

### Microglia depletion
To eliminate microglia, mice were fed PLX-5622, a CSF1R inhibitor provided by Plexxikon and formulated at a dose of 1200 mg/kg in AIN-76A chow from Research Diets Inc., for 3 weeks before experimentation.

### Neutrophil depletion
Neutrophils were depleted through repeated i.p. injections of rat anti-mouse Ly6G antibody (50 μg, clone 1A8, BioXCell, BE0075-1) and mouse anti-rat IgG2a (50 μg, clone MAR 18.5, BioXCell, BE0122), using an adaptation of the depletion strategy proposed by ref. [50]. Rat IgG2a isotype (50 μg, clone 2A3, BioXCell, BE0089) and PBS served as controls. Mice were injected with 100 μl of antibody at 0.5 μg/μl or saline, according to the regimen protocol described in Fig. 9a.

### In vivo inhibition of ROS production
Mice received 4 intraperitoneal injections of saline or NAC (150 mg/kg) at 12 h before, 1 h before, 12 h after and 24 h after i.c.m. injection of either PBS or rmIL-1α. A similar dose and regimen of NAC (or saline) was given to SCI mice. Tissue was collected at 24 h post-i.c.m. injection/SCI for subsequent immunohistological analysis.

### Tissue processing
Mice were overdosed with a mixture of ketamine (400 mg/kg) and xylazine (40 mg/kg) and transcardially perfused with 1% PFA, pH 7.4, in PBS. Spinal cords were extracted from vertebral columns, post-fixed for an additional 48 h in 1% PFA at 4 °C, and then transferred into PBS + 20% sucrose for at least 24 h before tissue sectioning. For experiments involving cytokine injection i.c.m., spinal cords were blocked into 4-mm segments, corresponding to the upper cervical, mid-thoracic and upper lumbar levels. For SCI experiments, a spinal cord segment of 12 mm centered over the lesion site was divided in three equal segments (i.e., epicenter, rostral, and caudal segments), as before[87]. Thus for each animal, a total of three spinal cord segments were embedded in Shandon™ M-1 Embedding Matrix (Thermo Fisher Scientific) with tissue sections cut at a thickness of 14 μm using a cryostat (model CM3050S; Leica Biosystems). All sections were collected directly onto Surgipath X-tra® slides having a permanent positive charged surface (Leica Microsystems Canada) and stored at −20 °C until use.

### Immunostaining and quantification
Immunofluorescence labeling was performed according to our previously published method[87]. Primary antibodies used in this study come from the following sources (catalog numbers in parentheses) and were used at the indicated dilutions: rat anti-BrdU (1:1000, clone BU1/75 [ICR1], Abcam, ab6326), rat anti-C3 (1:100, clone 11H9, Abcam, ab11862), mouse anti-CC1 (1:1000, clone CC1, Abcam, ab16794), rat anti-CD11b (1:250, clone 5C6, AbD Serotec, MCA711), rabbit anti-c-Fos (1:500, clone 9F6, Cell signaling, #2250), mouse anti-GalC (1:800, clone mGalC, Millipore, MAB342), goat anti-Iba1 (1:1500, Novus Biologicals, NB100-1028), goat anti-IL-1α (1:100 dilution, R&D Systems, AF-400-NA), rabbit anti-Ki67 (1:200, Abcam, ab15580), rabbit anti-laminin (1:1000, Dako, Z0097), rat anti-Ly6G (1:2000, clone 1A8, BD Biosciences, #551459), mouse anti-NeuN (1:250, clone A60, Millipore, MAB377), rabbit anti-NG2 (1:100, Millipore, Ab5320), rat anti-NG2 (1:200, clone 546930, R&D Systems, MAB6689), mouse anti-O4 (1:400, clone O4, R&D Systems, MAB1326), goat anti-Olig2 (1:400, R&D Systems, AF2418), rabbit anti-P2ry12 (1:500, AnaSpec, AS-55043A), goat anti-Sox9 (1:250, R&D Systems, AF3075), and rabbit anti-Sox9 (1:1000,

Millipore, AB5535). For Ki67 immunofluorescence, antigen retrieval was performed using a sodium citrate buffer at 95 °C for 5 min. For BrdU, tissue sections were treated with HCl (2.0 N) for 30 min at 37 °C followed by 0.1 M sodium borate (pH 8.5) for 10 min at room temperature. Primary antibodies were visualized with the appropriate Alexa Fluor®-conjugated secondary antibodies from Thermo Fisher Scientific (1:250 dilution). DAPI (1 μg/ml, Thermo Fisher Scientific) was used for nuclear counterstaining. Sections were imaged on a Zeiss LSM800 confocal microscope system equipped with 405, 488, 561, and 640 nm lasers. Confocal images were acquired and mosaics created using the Zen Blue Edition software (v. 2.3, Carl Zeiss). For high resolution images, the Zeiss Airyscan module was used.

All stereological counts were performed using the BIOQUANT Life Science software (v. 18.5, Bioquant Image Analysis Corporation). The total number of IL-1α-expressing cells and IL-1α-expressing microglia (TdT$^+$) per cross section was counted at 20× magnification using mosaics created from 6–12 overlapping confocal images. For the quantification of activated glial cells (Fos$^+$Sox9$^+$ or Fos$^+$Olig2$^+$CC1$^+$ or Fos$^+$Iba1$^+$) and neurons (Fos$^+$NeuN$^+$), OLs (Olig2$^+$CC1$^+$ or Olig2$^+$NG2$^+$BrdU$^+$ or Olig2$^+$Ki67$^+$) and reactive inflammatory astrocytes (Sox9$^+$C3$^+$), the total number of immunolabeled cells in the spinal cord white matter (for glial cells) or gray matter (for neurons) per cross section was counted at 20×, 40× and 20× magnification, respectively. For the quantification of neutrophils, the total number of Ly6G$^+$ or Ly6G$^{TdT+}$ cells in the spinal cord blood vessels or parenchyma, delineated using pan-laminin immunostaining, was counted at 20× magnification in the entire spinal cord cross section. Only immunolabeled cells with a DAPI-stained nucleus were counted, and results were presented as the total number of immunolabeled cells per mm². All quantifications were done blind with respect to the identity of the animals.

### Isolation of specific CNS cell types
**OPCs and late OL progenitors (pro-OLs).** OPCs were isolated from the neonatal (P7–P9) mouse brain to perform cell culture studies, whereas pro-OLs were isolated from adult brains (6–8 weeks old mice) to assess the efficiency of Cre-mediated recombination. Mouse pups were anesthetized on ice and sacrificed by decapitation according to institutional guidelines. Adult mice were anesthetized and perfused intracardially with ice cold Hanks' balanced salt solution (HBSS) without Ca$^{2+}$/Mg$^{2+}$ to remove blood from the vasculature. Brains were extracted and cerebral cortices dissected and diced into small pieces, which were then incubated in a mixture of papain (0.9 mg/mL), cysteine (0.2 mg/mL) and DNase I (39 U/ml) at 37 °C for 20 min on an orbital shaker. The digestion was stopped by adding a trypsin inhibitor with a final concentration of 5 mg/mL, and the tissue was dissociated into a single-cell suspension by gentle mechanical trituration. OPCs and pro-OLs were isolated by immunopanning at room temperature using antibodies directed against PDGFRα (also known as CD140a; 1:300 dilution, clone APA5, BD Biosciences, #558774) and O4 (1:300, clone O4, R&D Systems, MAB1326) cell surface markers, respectively, following the methods of Emery and Dugas[101].

**Microglia.** Microglia were isolated from the adult mouse spinal cord at 8 weeks of age to determine the efficiency of Cre-mediated recombination or to assess gene expression by qRT-PCR. To sum up, mice were transcardially perfused with cold HBSS (without Ca$^{2+}$/Mg$^{2+}$) to remove immune cells from the vasculature, and their spinal cords were dissected out and gently mechanically homogenized on ice. Cells were next filtered through a 70-μm nylon mesh cell strainer (BD Biosciences), fractionated using 37–70% Percoll density gradient centrifugation and microglia-enriched fractions purified by cell sorting according to the methods and cell surface markers described below, and the gating strategy described in Supplementary Fig. 8.

**Endothelial cells (ECs).** ECs were isolated from the brain capillaries of mice aged 6–8 weeks. In brief, mouse brain tissue that was free of meninges was minced, homogenized and digested in a mixture of 0.7 mg/ml collagenase type II and 39 U/ml DNase I in Dulbecco's Modified Eagle Medium (DMEM) for 75 min at 37 °C. Myelin was removed by centrifugation at 1000 g for 20 min in 20% BSA-DMEM. The cell pellet was then incubated for another h at 37 °C with a mixture of 1 mg/ml collagenase-dispase and 39 U/ml DNAse I in DMEM. Microvascular ECs were separated on a 33% continuous Percoll gradient.

**Astrocytes.** To assess the efficiency of Cre-mediated recombination in astrocytes, astrocytes were isolated from the brain of adult mice using the semi-automated gentleMACS™ Octo Dissociator with Heaters, the Adult Brain Dissociation Kit and the Anti-ACSA-2 MicroBead Kit, according to the standardized methods and protocols developed by Miltenyi Biotec. Astrocytes were further enriched by cell sorting using the APC-conjugated anti-mouse ACSA-2 (1:50, clone REA969, Miltenyi Biotec, 130-116-245) and Alexa 488-conjugated mouse anti-Oligodendrocyte Marker O4 (1:50, clone O4, R&D Systems, FAB1326G) antibodies, according to the gating strategy described in Supplementary Fig. 8.

### Primary cell culture
**Oligodendrocytes.** OPCs were seeded in poly-L-lysine-coated cell culture plates in the presence of DMEM-Sato Base Growth Medium and their proliferation was induced by adding platelet-derived growth factor-AA (PDGF-AA,10 ng/mL) and basic fibroblast growth factor (bFGF, 20 ng/mL) in the absence of the T3 thyroid hormone, following a protocol adapted from Haines and colleagues[102]. Once a sufficient number was reached, OPCs were differentiated into mature OLs by adding the T3 hormone to the DMEM-Sato Base Growth Medium, in the absence of PDGF and bFGF. The differentiation of OPCs into mature OLs was confirmed by immunostaining against the O4 marker and assessed according to cell morphology.

**Endothelial cells.** Primary brain microvascular endothelial cells (BMECs) were plated on culture dishes coated with 10 μg/ml collagen type IV and 5 mg/ml gelatin and cultured in DMEM supplemented with 20% FBS, 1 ng/mL basic fibroblast growth factor, 100 μg/mL heparin, 1.4 μM hydrocortisone with antibiotics and antimycotics. Medium was supplemented with puromycin (10 μg/mL) for the first 2 days of culture, after which the concentration was adjusted to 4 μg/mL.

**Astrocytes.** Astrocytes were isolated from the cortex of P0-P2 mouse pups, as described by Schildge and colleagues[103], and used from passages 3 to 4. In summary, meninges were removed and the cortex isolated, minced and incubated for 40 min in a collagenase IV solution (750 U/mL) at 37 °C under occasional agitation. The suspension was centrifuged, resuspended in complete DMEM supplemented with 10% FBS, dissociated into a single-cell-suspension and then seeded in T75 flasks coated with poly-D-lysine in a 37 °C, 5% CO2 humidified incubator. After 7–8 days, cells were agitated on an orbital shaker at 37 °C at 180 rpm for 30 min to remove microglia. Medium was removed and replaced by fresh medium. Cells were again agitated at 37 °C at 240 rpm for 6 h to remove OPCs. Medium was removed and astrocytes were collected by trypsinization. Astrocytes were grown in complete DMEM either in 6-well plates (to generate astrocytes-conditioned medium, ACM) or directly into 96-well plates (for the ROS assay) coated with poly-L-lysine (0.1 mg/ml) at a density of 200,000 cells/well or 20,000 cells/well, respectively.

### Automated blood cell count, flow cytometry and cell sorting
Mice were overdosed with a mixture of ketamine-xylazine and their blood collected via cardiac puncture using a 22-gauge heparinized

syringe. Blood samples were immediately transferred into EDTA-coated microtubes (Sarstedt) and put on slow rotation at 5 rpm (using the Mini LabRoller™ Rotator) until processing was complete. Automated complete blood cell count was performed using the Scil Vet abc Plus+™ Analyzer (Scil Animal Care Company) following manufacturer's instructions. For flow cytometry, red blood cells were lysed and the remaining cells incubated with Mouse Fc Block (i.e., purified anti-mouse CD16/CD32; clone 2.4G2, BD Biosciences, #553141) for 15 min at 4 °C to prevent nonspecific binding. Multicolor labeling was then performed using the LIVE/DEAD™ Fixable Yellow Dead Cell Stain Kit (Thermo Fisher Scientific) and the following fluorescently-conjugated primary antibodies (all from BD Biosciences): PerCP-conjugated anti-CD45 (1:50 dilution, clone 30-F11, #557235), Alexa 700-conjugated anti-CD11b (1:50, clone M1/70, #557960), and BD Horizon™ V450-conjugated anti-Ly6C (1:83, clone AL-21, #56094). Data were acquired on a LSRII flow cytometer (BD Biosciences) using FACSDiva software (v. 6.1.3, BD Biosciences), and further analyzed using FlowJo software (v. 9.2, Tree Star Inc.).

For the purification of microglia required for DNA and mRNA analyses via quantitative real-time PCR (qPCR) and RT-PCR (qRT-PCR), respectively, microglia were isolated from the adult spinal cord as described above and then sorted using a BD FACSAria II (BD Biosciences). The following primary antibodies were used (all from BD Biosciences): PerCP-conjugated anti-CD45 (1:50 dilution, clone 30-F11, #557235), Alexa 700-conjugated anti-CD11b (1:50, clone M1/70, #557960), FITC-conjugated anti-Ly6C (1:83, clone AL-21, #553104), and PE-Cy7-conjugated anti-Ly6G (1:50, clone 1A8, #560601). Microglia were identified as CD45$^{int}$ CD11b$^+$ Ly6C$^-$ Ly6G$^-$ cells.

## Assessment of Cre-mediated recombination using qPCR

Genomic DNA was extracted from pro-OLs, microglia, ECs or astrocytes isolated from the mouse CNS using the GenElute Mammalian Genomic DNA Miniprep Kit (Sigma-Aldrich Canada Ltd.), according to the manufacturer's instructions. The following primers were used to determine the extent of Cre-mediated deletion of the inserted sequence, which contained a neocassette (Neo) and other interfering elements (flanked by *loxP* sites) causing disruption of *Il1r1* gene expression, and whose deletion restored the normal coding frame of the *Il1r1* gene in *Il1r1*$^{r/r}$ mice crossed with cell-specific Cre mice: *Neo*, 5'-gcttgggtggagaggctattc-3' and 5'-gcctcgtcttgcagttcattca-3' and *Il1r1* (intron 7), 5'-gcccttttcttacattctatttggtgc-3' and 5'-caagaaggagttaaccgggacatc-3'. Primers designed to amplify intron 7 of the *Il1r1* gene were used to normalize gene expression between animals. To determine the extent of deletion of the floxed *Il1r1* alleles in *Cdh5*$^{CreER}$::*Il1r1*$^{fl/fl}$ and *Gfap*$^{Cre}$::*Il1r1*$^{fl/fl}$ mice, qPCR amplification was conducted using primers spanning intron 2 and exon 3 of the *Il1r1* gene: *Il1r1* (intron 2-exon 3), 5'-cattgcttctcctttctctcttttaa-3' and 5'-gccgtgcattttatttggagta-3'. As above, intron 7 was amplified to normalize data. The sequences chosen were selected to match only the intended gene using the GeneTool software (v. 2.0, BioTools Inc.), and verified by BLAST analysis in GenBank. Amplification was performed using reagents of the KAPA SYBR ® FAST qPCR Master Mix Kit (Kapa Biosystems Ltd.) optimized for the LightCycler 480 (Roche Diagnostics), and by following these conditions for PCR reactions: 45 cycles, denaturation at 95 °C for 30 s, annealing/elongation and reading at either 62 °C (for amplification of the *Neo* sequence) or 58 °C (intron 2-exon 3 of the mouse *Il1r1* gene) for 30 s. A melting curve was performed to assess non-specific signal. Fold change relative to the WT group was calculated using the delta-delta Ct method. Quantitative real-time PCR measurements were performed by the CRCHU de Québec–Université Laval Gene Expression Platform and were compliant with MIQE guidelines.

## Assessment of gene expression using real time qRT-PCR

FACS-isolated spinal cord microglia were homogenized using Qiazol Lysis Reagent (Qiagen) and total RNA extracted using the miRNeasy Micro Kit On-column DNase (Qiagen), following the manufacturer's instructions. The quantity of total RNA was measured using a Nano-Drop ND-1000 Spectrophotometer (NanoDrop Technologies). First-strand cDNA synthesis was accomplished using 0.1–0.2 μg of isolated RNA in a reaction containing 200 U of Superscript IV RNase H-RT (Thermo Fisher Scientific), 300 ng of oligo-dT$_{18}$, 50 ng of random hexamers, 50 mM Tris-HCl pH 8.3, 75 mM KCl, 3 mM MgCl$_2$, 500 μM deoxynucleotides triphosphate, 5 mM dithiothreitol, and 40 U of Protector RNase Inhibitor (Roche Diagnostics) in a final volume of 50 μl. The reaction was incubated at 25 °C for 10 min and then at 50 °C for 20 min, followed by inactivation by incubation at 80 °C for 10 min. A PCR Purification Kit (Qiagen) was used to purify cDNA. cDNA corresponding to 10–18 ng of total RNA was used to perform fluorescent-based real-time PCR quantification using the LightCycler 480 (Roche Diagnostics). Reagents of the KAPA SYBR ® FAST qPCR Master Mix Kit (Kapa Biosystems Ltd.) optimized for the LightCycler 480 were used as described by the manufacturer. The primer pairs were: *Il1r2*, 5'-gagaccccacacgcctattga-3' and 5'-ggttccgtggttgttcctttga-3' and *Gapdh*, 5'-ggctgcccagaacatcatccct-3' and 5'-atgcctgcttcaccaccttcttg-3'. Once again, primer pairs were designed using the GeneTool 2.0 software (Biotools Inc.) and their specificity verified by blast in the GenBank database. The conditions for PCR reactions were: 45 cycles, denaturation at 95 °C for 30 s, annealing/elongation and reading at 60 °C for 30 s. A melting curve was performed to assess non-specific signal. Relative quantity was calculated using the delta Ct method. Normalization was performed with the delta-delta Ct method using the PBS group and the *Gapdh* gene was used as a reference. Real time qRT-PCR measurements were performed by the CRCHU de Québec–Université Laval Gene Expression Platform and were compliant with MIQE guidelines.

## Multiplex cytokine assay

The concentrations of 32 plasma cytokines and chemokines were measured using the Mouse Cytokine 32-Plex Discovery Assay, a multiplex laser bead assay from Eve Technologies. Blood samples were collected and prepared according to the manufacturer's instructions, and then shipped to Eve Technologies for cytokines and chemokines measurement.

## Immunoblotting analysis

For protein extraction, cells were washed twice with ice cold PBS and then incubated in ice cold RIPA lysis buffer (100 μL for $1 \times 10^6$ cells; 20 mM Tris-HCl pH 8.0, 2 mM EDTA, 10 mM EGTA, 1% Triton X-100) containing Protease and Phosphatase Inhibitor Cocktail (Sigma-Aldrich Canada Ltd.). Cells were scraped off the plate and the lysate immediately transferred to a microcentrifuge tube on ice. Homogenates were sonicated and incubated for 20 min at 4 °C under rotation to complete cell lysis. Tubes were centrifuged at 13,000 g for 20 min at 4 °C and the supernatant collected into new microtubes. Protein levels were determined using the BCA Protein Assay Kit (Sigma-Aldrich Canada Ltd.).

For quantification of protein levels, 3 to 10 μg of proteins were boiled and electrophoresed on SDS-acrylamide gel, followed by blotting on PVDF or nitrocellulose membranes (PerkinElmer). Membranes were blocked for 1 h with 5% dry milk in TBST buffer (50 mM Tris-HCl pH 8.0, 150 mM NaCl, 0.05% Tween-20) and then incubated overnight in 5% milk/TBST solution containing primary antibody. The following antibodies were used: mouse anti-actin (1:75000, clone C4, EMD Millipore, MAB1501), rabbit anti-GFAP (1:8000, Agilent Technologies (Dako), Z0334), and goat anti-IL-1R1 (1:1500, R&D Systems, AF-771). Membranes were incubated with secondary antibodies conjugated to horseradish peroxidase (1:2500, Vector Laboratories) and cross-reactive bands visualized by chemiluminescence (PerkinElmer). Uncropped and unprocessed scans of immunoblots are shown in Supplementary Fig. 9.

## Intravital imaging of BSCB permeability

Intravital fluorescence videomicroscopy was used to determine whether central delivery of IL-1α i.c.m. affects the permeability of the BSCB. Animals were anesthetized with 1–2% isoflurane (vol/vol) and a laminectomy performed to expose the thoracic spinal cord at the level of the intersection between the dorsal spinal vein and the rostral radiculomedullary vessel, as illustrated elsewhere[104]. Mice were then transferred to a custom-made stabilization device adapted for intravital imaging of the mouse spinal cord (described in ref.[105]) and body temperature maintained at 37 °C using a temperature controlling device (RWD Life Science Co.). Labeling of blood was achieved through tail vein injection of a 40-kDa dextran-Texas red fluorescent tracer (1.25% wt/vol in sterile PBS, Life Technologies) 30 min prior to the imaging session. The thoracic spinal cord region to be imaged was next placed under the 20× objective and Gel-Seal (GE Healthcare) carefully applied on the bone surrounding the laminectomy to create a watertight cavity. All images were acquired on a custom-made BLIQ Photonics upright microscope coupled with an OptoSplit II LS image splitter to enable the simultaneous and separate visualization of GFP (Ex 480/20 nm, BS 495 nm, Em 510/20 nm) and/or RFP (Ex 580/25 nm, BS 600 nm, Em 625/30 nm). The microscope is equipped with a Lumen-300 LED light source (Prior Scientific), TTL triggering, a high-speed sCMOS camera (Orca Flash V2, Hamamatsu) and controlled by the Nirvana software (v. 1.9.4, Bliq Photonics). To quantify the presence (or absence) of the fluorescent tracer inside the spinal cord parenchyma, the maximum fluorescence intensity (FMax) of Texas Red was first measured in the spinal cord parenchymal tissue, the average tissue background subtracted, and the resulting value divided by the FMax measured in the surrounding blood vessel minus the background signal. Quantitative analysis was performed using ImageJ (v. 1.44o, Wayne Rasband, NIH). Data are presented as a ratio between the FMax of Texas Red in the spinal cord parenchyma versus in the blood vessels.

## Detection of ROS in vitro

Production of ROS was assessed in living cells by using the fluorogenic probe CellROX® Green Reagent (Thermo Fisher Scientific), a cell-permeable dye that is non-fluorescent in a reduced state, but upon oxidation exhibits strong fluorescent signal with excitation/emission maxima of ~485/520 nm. Primary astrocytes, seeded in 96-well plates at 20,000 cells/well, were exposed to vehicle or rmIL-1α (10 ng/mL) for various time points ranging from 0–24 h. Cells were loaded with 5 μM CellROX® Green Reagent and 1 μg/ml Hoechst 33342 for nuclear counterstain for 30 min and then rinsed three times with PBS and finally fixed with PFA 4% for 15 min at room temperature. Imaging was then performed using the EVOS® FL Auto Imaging System (Thermo Fisher Scientific) and further analysed with Image J.

## In vitro cytotoxic assay

To study the effect of conditioned medium derived from astrocytes or ECs stimulated with either IL-1α or PBS on the survival of mature OLs, cell viability was measured using the Pierce Lactate Dehydrogenase (LDH) Cytotoxicity Assay Kit (Thermo Fisher Scientific), according to the manufacturer's instructions. In brief, primary astrocytes were plated in 6-well plates at a density of 100,000 cells/well and treated with either rmIL-1α (10 ng/mL) or PBS. Following a 24-h incubation period, the conditioned medium was removed and immediately transferred onto mature OLs plated at a density of 20,000 cells/well in 96-well plates for 24 h. The LDH reaction mixture was then added to the culture medium for 30 min at room temperature, and the reactions stopped and absorbances measured at 490 nm and 680 nm (background signal from the instrument) using the SpectraMax i3 (Molecular Devices).

## Statistical analysis

Statistical evaluations were performed using the Student's *t*-test or one- or two-way ANOVA or repeated-measures ANOVA where appropriate. The multiple comparisons adjustment was made using the Bonferroni correction. A Pearson's correlation was used to assess if there was an association between the infiltration of neutrophils and OL loss. All statistical analyses were performed using the GraphPad Prism software (v.9.4.1, GraphPad Software Inc.). A $p < 0.05$ was considered as statistically significant. All data in graphs are expressed as means ± SEM.

## Reporting summary

Further information on research design is available in the Nature Research Reporting Summary linked to this article.

## Data availability

Raw image data can be obtained from the corresponding author upon reasonable request. Requests will be treated within 2 weeks. All other source data are provided with this paper as a Source Data file available in Supplementary Information. Source data are provided with this paper.

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

## Acknowledgements

This study was supported by grants from the Canadian Institutes of Health Research (PJT-162387 to S.L.) and Wings for Life Spinal Cord Research Foundation (WFL-CA-13/20 to S.L.). This project has also been made possible with the financial support of Health Canada, through the Canada Brain Research Fund, an innovative partnership between the Government of Canada (through Health Canada) and Brain Canada, and the Barbara Turnbull Foundation for Spinal Cord Research. Salary support to F. B. was in part provided by the Fondation du CHU de Québec and the Center thématique de recherche en neurosciences (CTRN). We thank Nadia Fortin for her invaluable technical assistance. This manuscript is dedicated to the memory of our friends and colleagues Giamal N. Luheshi and Guy Drolet, who sadly passed away in 2020.

## Author contributions

F.B. and A.C.M. conceived the study, designed and performed most of the experiments, analyzed the data, drafted the figures and wrote the manuscript. D.B., M.K., and M.L. performed immunofluorescence and quantitative analyses, some in vitro experiments, as well as commented on the manuscript. B.M. participated to the experimental design and commented on the manuscript. A.B. contributed to writing the final manuscript. N.V. acquired microscopy images and edited all figures. M.G., X.L., E.B., and N.Q. provided materials and commented on the manuscript. S.L. conceived the study, designed the experiments, supervised the overall project and wrote the manuscript.

## Competing interests

The authors declare no competing interests.
