## [Peer Review File · Nature Communications]

The alarmin interleukin-1 α triggers secondary degeneration through reactive astrocytes and endothelium after spinal cord injuryREVIEWER COMMENTS

Reviewer #1 (Remarks to the Author):

The alarmin interleukin-1 α triggers secondary degeneration through reactive astrocytes and endothelium after spinal cord injury

General Impressions: There is a lot to unpack in this paper, and though it is generally well written the conceptual framework is at times hard to follow and tenuous. I would not say any of the findings reported here are surprising or novel, but this does add incrementally to the literature and does add to current knowledge regarding secondary injury after SCI. Whether this will eventually lead to any kind of meaningful treatment in the prevention of secondary injury remains to be seen, but there are some interesting ideas here. What this manuscript does do well is pull together data on IL-1 α in SCI that has been spread across dozens of previously published papers and present it all together in a single package.

The first five or so figures seem to only serve the purpose of demonstrating the basics, which is somewhat an unnecessary use of space. The story doesn't start to come together until figures 6 and 7. There are also entire sections dedicated to describing the details of supplemental figures, leading me to believe that they should not be supplemental or need to be presented differently.

Below I list my biggest concerns, followed by summaries of the sections and figures.

-These experiments rely on direct injection of IL-1 α as being representative of the SCI environment, however when repeated in an SCI model many of the observations did not replicate. While the mechanisms described may still be applicable, their therapeutic efficacy in the complex secondary injury cascade may be limited. Not addressed or acknowledged in the discussion.

-The effect of several of the KI/KO models is limited (i.e. less than 30% increase in gene expression or less than 20% decrease). Could this not influence the extent to which these models show a change in response compared to control? Authors claim the lack of effect indicates the cell type is not directly involved in the effects but I'm not convinced this data is strong enough to support that. This is touched on in the discussion: "As the knockout will only be partial, it cannot be determined with certainty whether the response observed is due to the receptors that are still expressed by the targeted cell population or another cell type. Here, we circumvented part of this problem by using restored and knockout mouse models as well as primary cell lines, thus ensuring an additional level of confidence and reproducibility to our results." The repeated lines tested are supportive, but still the issue remains.

-In the methods it says that male and female mice were used. Male and female rodents have been shown to have very different neuroimmune responses. It is not clear whether sexes were analyzed separately to take into account possible sex differences.

Introduction Summary

This lab has previous work demonstrating release of IL-1 α by microglia. Current goal is to determine the extent to which this release drives secondary degeneration via astrocytes and endothelial cells.

****This section did not provide enough overview of neuroinflammation and the role of cytokines/IL-1 α to convince me that their question is novel or brings new information to the field. There are similar papers in existence.****

This statement: The transition from primary to secondary damage is triggered, at least in part, by the release of the intracellular content of necrotic cells, in particular danger-associated molecular patterns (DAMPs, also referred to as alarmins) **could give a clearer picture of the importance of IL-1 α in the secondary process aside from a basic generalization.**

Methodology and Results Summary:

Animal model – mouse

Figure 1. Goal is to show that IL-1a is released by microglia and not macrophages. Used a tamoxifen conditional expression model to label microglia (under promoter Cx3cr1) with TdT. Induction 1 month prior to injury, animals sacrificed 4 hours post SCI to observe maximal cytokine release. Stained for IL1a. 95% of IL-1a expressing cells were TDT positive, suggesting microglial identity. Following microglia depletion the number of IL-1a cells significantly decreased. High resolution confocal images suggest IL-1a, TdT positive cells are damaged microglia based on abnormal morphology. **The addition of arrows to these images (C-L) to highlight relevant morphology would make it easier to interpret. These IHC images alone are insufficient to resolutely claim that the microglia are the primary releasers of IL-1a. Nothing has been done here to show that macrophages aren't also releasing it.**

Figure 2. Goal is to show what the cellular response is to IL-1a. Injected IL-1a into cisterna magna of WT mice. Looked for cfos expression (indicates transcriptional activation) in spinal cord with IHC. Significant increases in cfos in Olig2 (OL) and Sox9 (astro) positive cells compared to control PBS injected animals. Increase peaked at 4 hrs post injection and was no longer significant by 24hrs. **Olig2 alone is insufficient to resolutely indicate activation of OLs. Olig2 is not specific to OLs.**

Figure 3. Injected IL-1a into cisterna magna of WT mice. Looked for neurotrophil infiltration. Significant increases compared to PBS controls after 24 hours in spinal cord parenchyma and neurovascular space. Returned to normal by 3 days. Demonstrates order of events: IL-1a release followed by glial activation followed by neurotrophil infiltration. Also quantified OL cell count in Olig2 CC1 positive cells and found a significant decrease (40%) in the number of mature OLs. Data not shown: OL loss increased as IL-1a concentration increased.

****Figures 2 and 3 would be more impactful if combined.****

Figure 4. BrdU experiment to examine OPC proliferation following IL-1a injection. Olig2 Brdu positive cells increased significantly 3 days post IL-1a injection. Ki67 staining supported this. OL number returned to baseline by 5 days post injection. Takeaway is that robust proliferation is able to restore IL-1a mediated OL loss within days **1) This is an example where cytokine injection does not necessarily equal SCI, where evidence of full OL recovery is controversial and mixed. There is no context provided for how this finding is related to actual injury. Discussion could be more robust. 2) Olig2 alone is not specific for OLs. 3) The limitations/translatibility of this model could be discussed further.**

Figure 5. Goal is to determine the role of the receptor IL-1R1 following IL-1a release. Coadministered IL-1a injection with IL-1a receptor antagonist anakinra. This completely blocked infiltration of neurotrophils and protected against OL loss. Implicates this receptor as primary pro-inflammatory response to IL-1a.

Supp Figure 2. Goal to determine the role of the R1 receptor in an SCI model. WT, IL-1a KO, and IL-1aR KO mice underwent SCI and recovery followed over 1 month. Both KO groups had better locomotor recovery than WT. This further implicates the cytokine and its receptor in the inflammatory response. **This seems like a powerful finding and I'm not sure why it's in supplemental.**

Supp Figure 3. Goal to determine what cell types R1 is expressed in. Immunoblot analysis of IL-1R1 showed expression in endothelial cells, OPCs, astrocytes, and microglia. This line concerns me: "Intriguingly, the protein detected using the anti-IL-1R1 polyclonal antibody in OPCs and astrocytes was of a slightly lower molecular weight than the predicted 75-kDa IL-1R1 protein, suggesting the possibility that cells of the glial-restricted lineage could express a different form of the receptor. Along this line, we point out that IL-1R3 is identical to IL-1R1 at the C terminus, but with a shorter extracellular domain and a predicted molecular weight of about 66 kDa." **Is this suggesting that their immunoblot may have actually picked up IL-1R3 instead or in addition to IL-1R1? If this is a possibility, I think it needs to be determined for sure or

this should not be reported as IL-1R1 expression alone. **

Figure 6. This and subsequent figures explore the response of specific cell types to IL-1a. Use mice expressing an inducible Cre recombinase under the control of the *Pdgfra* promoter to conditionally restore *IL1r1* gene expression specifically in OPCs of an *IL1r1/3* KO. ****Is it problematic for both R3 and R1 to be knocked out since they've focused on R1? No characterization is done to determine R3 effects or expression.**** Report the KI shows a 20% increase in R1 expression determined by qPCR in both OPCs and O4 OLs. However, this did not reverse the loss of neurotrophil infiltration or OL death following injection with IL-1a. ****Figure C is missing the label for PBS vs injection.**** Claim that the lack of response to "recovered" KO lines to IL-1a means that IL-1a is not directly responsible for the OL death observed in Fig3. ****With only a 20% recovery of gene expression and 40% recombination, especially without any presentation of the actual qPCR data that allowed them to make this calculation, it seems equally likely that the lack of reversal could have happened because the gene was only partially restored. Furthermore, it is not clear to me how they calculated % recovery from the raw qPCR data and this information would be useful.****

Figure 7. Goal is to determine whether IL-1R1 has autoregulatory effect on microglia (i.e. release of IL-1a from microglia then binds to receptor on microglia) that contributes to secondary damage. Same inducible model as Figure 6, this time using *Cx3cr1* promoter for conditional R1 expression in microglia. Report 65% restoration in gene expression. Similar to Figure 6, this increase in gene expression in microglia did not correlate to a reversal in response to IL-1a for neurotrophil infiltration or OL death. ****Given this result with a higher induction rate, I am more inclined to believe the results in figure 6 and that this suggests both cell types do not directly mediate the effects of IL-1a. I don't think the possibility can be completely ruled out, however, that the increase in gene expression was just insufficient to observe a response. For both Figure 6 and 7, Figure C appears to have a slight downward trend, even though it doesn't reach significance.**** Next, used three groups of WT mice (PBS injection, IL-1a injection, IL-1a + microglia depletion) to determine the effects of microglia on response. Mice with microglia depleted had a potentiated reaction to IL-1a for neurotrophil infiltration and OL death compared to IL-1a injection only group. This suggests that microglia themselves actually have a protective effect. They hypothesize the presence of another IL-1a receptor on the microglia may act to sequester released IL-1a and test this with IL-1R2. qPCR demonstrates a substantial increase in R2 expression following IL-1a injection, supporting their hypothesis.

Figure 8. Same inducible model as Figure 6, this time using *Cdh5* promoter for conditional R1 expression in endothelial cells. Report 90% restoration in R1 gene expression. Also observed a 35% return in neurotrophil infiltration and a slight increase in OL death following IL-1a injection. ****Is it expected for a 90% restoration in receptor expression in the KO to have such a small return to baseline response? What else accounts for the still high differences between the inducible model and the WT?*** The same animal model following SCI did not return to WT locomotor scores. This suggests endothelial cells are partially involved in the response to IL-1a. To further explore this, they utilized a transgenic IL-1R1 KO (rather than the inducible knock-in from the last few figures) model that decreased receptor expression by 50% compared to control in endothelial cells. Knocking out this receptor in endothelial cells significantly decreased neurotrophil expression and OL death following IL-1a injection. This did not translate to locomotor recovery after SCI.

Figure 9. Inducible KI model described previously, this time using *GFAP* promoter for conditional R1 expression in astrocytes. Report 30% return to normal R1 expression levels. There was no change in neurotrophil infiltration however OL death increased. There was also no effect on the SCI model locomotor scores. Again this was repeated in a transgenic KO line in astrocytes. This line had reduced expression of the receptor by roughly 20%. KO of the IL-1R1 did not prevent neurotrophil infiltration but did decrease OL death. This did not translate to any locomotor recovery following SCI. Concludes that

astrocytes partially mediate the response to IL-1a. ****Same concerns mentioned previously – low efficacy of KO and KI make me wonder if part of the lack of an observed effect was for this reason and not related to the mechanisms of IL-1a action.****

Figure 10. Goal to determine whether IL-1a administration induces C3 expression in astrocytes. Saw two fold increase 24 hrs post injection. In culture, application of IL-1a to OLs had no toxic effect, however replacing with media from astrocytes cultured in IL-1a was toxic to OLs. Hypothesized the toxic effect involved reactive oxygen species (ROS). Used an assay to measure ROS released from astrocytes stimulated with IL-1a in culture and found significant increases in ROS producing cells. Treatment with ROS antagonist completely prevented OL loss. Applied ROS antagonist to SCI animals and found similar reduction in OL death caudal to injury but no effect on neurotrophin infiltration.

Supp Figure 7: ROS antagonist decreased neurotrophin infiltration following IL-1a injection but not in SCI model.

Conclusions:

Pathway: SCI damages microglia, triggering release of IL-1a, which acts on astrocytes and endothelial cells to trigger a reactive neuroinflammatory response, including the release of ROS that directly damage OLs.

Blocking IL-1a receptor activity and/or ROS production are promising therapeutic targets for SCI

Reviewer #2 (Remarks to the Author):

The manuscript by Bretheau and colleagues provides a mechanistic investigation of the role of IL-1a – IL-1R1 signalling in oligodendrocyte death in two models of CNS inflammation, traumatic spinal cord injury and injection of IL-1a into the cisterna magna. The present study advances significantly from previously published work, addressing outstanding various knowledge gaps at both the cellular and molecular level. Overall, this is an elegant study, containing a very large body of work using numerous mouse lines/crosses to conditionally delete as well as restore expression of IL-1R1, thereby identifying relevant the cellular players and/or pathways.

A number of questions and suggested experiments that need addressing and/or doing is provided below:

Figure 1: The authors demonstrate that microglia are the source of IL-1a very early after SCI.

- (1) Is IL-1a also actively secreted rather than passively released by microglia?
- (2) If not, is its contribution to OL death only acute as microglia numbers are rapidly restored during the first week of SCI (previous work by this group), but OL loss peaks only at one week (Ref 25) and continues for weeks post injury?
- (3) Related to previous points, do the authors believe that IL-1a still drives OL death here, or would it be ROS produced via different pathways?
- (4) Inclusion of IL-1a protein levels and cellular sources at later time points post SCI would be desirable, including when microglia are depleted and for SCI experiments where IL-1R1 is conditionally removed (or restored) from select cell types.
- (5) To convincingly demonstrate IL-1a release by microglia, and the subsequent impact on OL death, the authors should aim to target/delete the ligand in microglia.

Figure 2: Immediate early gene protein Fos is upregulated in OLs following IL-1a injection.

- (6) Is SOX9 specifically expressed by astrocytes or do e.g. OPCs also express it? Can another astrocyte marker be used?
- (7) Also, was Fos expression only observed in in the white matter? No expression in neurons and/or microglia?

(8) How do IL-1a levels being injected compare to those seen after SCI in CSF as well as plasma? Are they comparable? The authors should measure plasma levels of IL-1a when this is injected into the cisterna magna.

Figure 3: IL-1a injection results in Ly6G infiltration and loss of OL.

(9) The authors convincingly show neutrophil recruitment and OL loss, but similar effects have been reported following injection of IL-1beta into the CNS. What is the specificity of the observed effect?

(10) Also, taken in conjunction with Figure 10, do the authors have insights as to when astrocytes begin to express ROS following IL-1a injection; the peak of Fos activation appears at 4h and the OL death is apparent at 24h post injection.

Figure 5: Delivery of Anakinra into the CSF blocks IL-1a induced neutrophil recruitment and OL death.

(11) Is anakinra also effective in SCI, i.e. under conditions where both IL-1a and IL-1beta are present?

Figure 6: Restoration of Il1r1 within the oligodendrocyte lineage does not allow for IL-1a-mediated neuroinflammation and OL loss.

(12) The effect on neutrophil infiltration is convincing but considering the low level of restoration (20-25%) of PDGFRa on OPCs and estimate of 40% on OL plus the variation and low animal numbers, the authors should increase groups size for figure 6C.

(13) It would be also helpful to colour code individual data points (see 6b).

Figure 7: Microglia alleviate IL-1a-mediated neuroinflammation and OL loss independent of IL-1R1 expression.

(14) Taken in conjunction with Figure 10 data and considering that microglia secreted factors drive A1 phenotypes, what happens to fos expression and C3 staining in astrocytes in absence of microglia?

Figures 8 & 9: IL-1a-induced neuroinflammation and OL loss is partly mediated by endothelial and astrocytic IL-1R1.

(15) Is IL-1R1 deleted here from CNS endothelial cells only or globally?

(16) What is the systemic response to both IL-1a injection and SCI under conditions where IL-1R1 is present, conditionally restored or deleted?

(17) Figure 8 and 9, data should be included the number of OLs, neutrophils, and also astrocyte activation after SCI.

(18) It would also be of interest to correlate OL number with BMS scores.

Figure 10:

(19) Does injection of rmIL-1a open the blood brain/spinal cord barrier? If so, C3 (and other proinflammatory factors) could come in from the circulation and contribute to OL death. This should be controlled for, also considering the observed effects in mice where IL-1R1 was selectively restored or deleted from endothelial cells.

(20) For the data presented in panel 10I, does NAC treatment, either alone or in combination with Anakinra, improve the neurological outcome from SCI?

(21) Considering that in vivo NAC treatment does not selectively target astrocytes, do cultured microglia (and other key cells in the inflammatory infiltrate after SCI) also produce ROS in response to IL-1a stimulation?

Reviewer #3 (Remarks to the Author):

The manuscript by Bretheau et al focuses on interleukin-1a (IL1a) as a key factor in neuroinflammation and oligodendrocyte (OL) loss after spinal cord injury (SCI). After determining that microglial cells are the main source of IL1a during the subacute phase after SCI, they demonstrate that intra cisternal magna delivery of IL1a induces glial

activation, neutrophil infiltration and mature OL death (replaced by OPC proliferation to compensate for OL death) through IL1R signalling in the mouse spinal cord. They further corroborate the detrimental role of alarmin IL1a within the context of SCI, demonstrating that lack of the cytokine or its receptor promotes better functional recovery after SCI. To demonstrate the cell types involved in OL death through IL1a signalling, the authors performed a careful and elegant series of gain and loss of function studies showing that detrimental effects of IL1a are mediated mostly through activation of endothelial and astrocytic IL-1R1 signalling. Finally, they identified toxic reactive oxygen species (ROS) expressed by astrocytes through IL1a signalling as one of the mediators responsible for OL death after SCI.

Although a role for IL1a in neuroinflammation after SCI has already been established (by the authors and others), the present study uses elegant methodology and transgenic mouse lines to advance our mechanistic understanding of the main cell types and mediators of these effects. The demonstration that astrocytes and endothelial cells are the main effectors in this detrimental role of IL1a by its receptor signalling is important and brings us a step forward in understanding pathological neuroinflammation after SCI, as well as identifying IL1a as a therapeutic target to prevent secondary damage after SCI. The data are convincing, and the studies are rigorous. I have three major points that should be taken into consideration by the authors, and several other minor points to consider.

Major points:

1. Provide stronger evidence for disregarding a direct neutrophil involvement in OL cell death by IL1a.

The importance of neutrophils in the development of the inflammatory response after SCI has been well documented (with early neutrophil recruitment, peaking at 18-24h after SCI, representing a key factor in the post-injury inflammatory response). Astrocytes are known to be involved in neutrophil recruitment after SCI, by the expression of CXCL1 and CXCL2 (demonstrated by the lead author and others; e.g. Pineau et al. 2010). These infiltrated neutrophils have also been previously associated with OL loss (Donnelly and Popovich, 2008; Satzer et al., 2015). Furthermore, in this article, the authors showed that in most cases when IL1a produced OL death, the recruitment of neutrophils was linked. Finally, neutrophils are one of the main sources of ROS during the early stages of the inflammatory response, and the authors identify ROS as one of main factors affecting viability of OL. For these reasons, neutrophils seem an obvious candidate that directly or indirectly could be involved in OL cell death. However, the authors claim that neutrophils are not involved directly in OL loss, supported by 2 different findings shown in supplementary figures 6 and 7. In my opinion, it is hard to disregard the link between neutrophils, ROS production and OL cell death based on the data in these supplementary figures.

In supplementary figure 6, the authors show that depletion of Ly6C/G+ (Gr-1) neutrophils and monocytes does not alter IL-1 α -mediated OL loss. There are some technical issues relating to non-specificity, since anti Gr1 not only affects neutrophil recruitment, but also depletes other cell types including monocytes. The use of alternative methods, such as 1A8 antibodies (anti-ly6G, which is more specific) requires more doses and could promote other side effects. The authors are clearly aware of the various issues and validated their results with LysMGFP mice, which is important and appreciated. However, leaving potential technical issues aside, I still am not 100% convinced about the affirmation that neutrophils are not involved in OL cell death by IL1a. The authors state that anti-Gr1 ab depleted neutrophil recruitment and the number of neutrophils were reduced in 70% compared with saline or isotype control after IL1a injection (supplementary 6h). However, the reduction reached no significance when compared with these groups and the depletion of neutrophils in the spinal cord was not complete. This remaining neutrophil infiltration could explain why Olig2+ cell counts were not re-established in comparison with saline treatment (vs IL1a injected mice). More consistent results should be provided to discard neutrophils as a mediator of OL cell death by IL1a. At the least, the authors could perform an in vitro analysis of

direct effects of IL1a on neutrophils and its effects on OL death. For example, evaluate the viability of OLs in culture, after application of supernatant from neutrophil cell cultures (wt and IL1r1ko) pre-activated with IL1a. Additionally, the authors could do more to explore the potential influence of other cell types on neutrophil activation and subsequent OL death. For example, whether IL1a activation of astrocytes could lead to neutrophil activation, and then neutrophils act as OL killers by an indirect effect of IL1a. As neutrophils are ROS producers, the authors could also evaluate the expression of ROS in neutrophils after IL1a activation. In supplementary figure 7 the authors discard a direct role of neutrophils in OL cell death by IL1a, by showing that an antagonist of ROS reduced OL death despite not reducing neutrophil recruitment after SCI. An alternative interpretation is that the lack of neutrophil effect on OL cell death is due to ROS inhibition, which may mask the potential effects of neutrophils on OL viability (since neutrophils also produce ROS, so by inhibiting ROS you are likely inhibiting a detrimental neutrophil effect on OL survival). As outlined above, additional studies to evaluate direct or indirect IL1a activation of neutrophils (which could produce ROS and contribute to OL death) will be important.

2. In relation to major point 1, the data in figure 9 (effects of restoration/deletion of IL-1R1 in astrocytes) reveals clearly that IL1a induces OL cell death in part through activation of IL-1R1 signalling in astrocytes. However, poor restoration effects could be the reason why it does not restore the infiltration of neutrophils. The authors could evaluate CXCL1, CXCL2 and CCL2 levels in each condition to determine whether the potential lack of these chemokines by the poor receptor restoration is responsible for the lack of neutrophil recruitment restoration. This would provide more in depth mechanistic information regarding the astrocyte-neutrophil-OL death axis.

3. Provide more detailed information on spatial and anatomical findings relating to OL death and neutrophil infiltration.

For most figures high power images of immunohistochemical staining are shown, but very little spatial or anatomical information is provided. In figure 2, for example, what region of the spinal cord is this? Was this effect seen across multiple segments at all levels? Was Fos activated in most white matter tracts? Can some low power images, and some spatial quantitation be added to more clearly see the location/distribution/extent of Fos reactivity across a tissue section?

Similar comment for fig 3 which shows high power images of Ly6G+ and Olig2+CCI+. In which regions/tracts/projections were these effects observed? This also applies to subsequent figures showing immunohistochemical findings. Are there any specific white matter tracts that are particularly vulnerable to IL1a-mediated OL death? Is it more pronounced in dorsal, lateral, ventral tracts? This seems important, particularly given the recent work by Floriddia et al showing that distinct oligodendrocyte populations have spatial preference and different responses to SCI (Nat Commun 11, 5860, 2020, DOI: 10.1038/s41467-020-19453-x).

Minor points:

1. In figure 2 the authors show that after IL1a i.c.m. injection, activation of oligodendrocytes and astrocytes peaked at 4 hr (using Fos expression as a marker of cell activation). This is a robust and clear finding. However, I miss the same analysis in microglial cells. Have the authors checked this in microglia? It is appreciated that evaluating c-fos expression and its upregulation after microglial activation could be problematic (in comparison to other neural cell types), but could authors use other marker(s) to analyse microglia activation at different time points after IL1a injection?

2. In figure 5, the authors further corroborate the role of IL1a in neutrophil recruitment and OL cell death through IL-1R1 signalling, since the use of IL1a inhibitor Anakinra abolished those effects. This data is clear. My concern is more related with the total Olig2 counting (cs/mm²) in animals treated with anakinra. Authors showed that in animals treated with anakinra Olig2 cell count is around 300 cells per mm², which is

significantly less than the other studies in the research paper (which showed at least 500 cells per mm²). Could this be a potential detrimental side effect of anakinra? Can the authors explain this reduction?

3. Supplementary figure 3 – add some labels to the figure to indicate what the bands are and what the lanes represent. The legend is also confusing, are lanes 1 and 6 both primary brain microvascular endothelial cells; and lanes 3 and 5 both primary astrocytes?

4. Some figures seem quite sparse and could be combined since the data are related, unless some additional data is provided (see major comments 1-3).

5. Additional minor points to consider in the text. Some statements seem to over-claim the findings and should be toned down e.g. line 440-441 “dramatic loss of mature OLs along the rostrocaudal axis of the spinal cord”. Unless some spatial characterisation is provided (see major point 2, above), this statement is not supported.

Similarly, Line 124, 145-146 and 924 “IL-1 α is released by injured microglia during the early acute phase of SCI”, “...these results show that damaged microglia release the alarmin IL-1 α during the early acute phase of SCI” and “Damaged microglia rapidly release IL-1 α at the site of spinal cord contusion in mice.” Immunohistochemical localisation does not demonstrate “release”.

The work of Didangelos on SCI and alarmins is relevant and should be cited.

Line 435-437 Please clarify the statement “We found that IL-1 α is released by microglia located in the lesion core, most of which are dead as early as 24 hours post-SCI.” Is something missing from this sentence, are they referring to OL death here?

RESPONSES TO THE REVIEWERS' COMMENTS:

Our responses to the Reviewers' comments are addressed point by point below. To facilitate the work of the Reviewers, all changes are highlighted in yellow in the revised manuscript. The authors have declared that no conflict of interest exists.

We thank the reviewers for their insightful comments and provide the following responses:

Reviewer 1:

General Impressions: There is a lot to unpack in this paper, and though it is generally well written the conceptual framework is at times hard to follow and tenuous. I would not say any of the findings reported here are surprising or novel, but this does add incrementally to the literature and does add to current knowledge regarding secondary injury after SCI. Whether this will eventually lead to any kind of meaningful treatment in the prevention of secondary injury remains to be seen, but there are some interesting ideas here. What this manuscript does do well is pull together data on IL-1 α in SCI that has been spread across dozens of previously published papers and present it all together in a single package.

The first five or so figures seem to only serve the purpose of demonstrating the basics, which is somewhat an unnecessary use of space. The story doesn't start to come together until figures 6 and 7. There are also entire sections dedicated to describing the details of supplemental figures, leading me to believe that they should not be supplemental or need to be presented differently.

We appreciate the reviewer's positive comments recognizing that this paper will both add incrementally to the literature, and to our understanding of the mechanisms responsible for secondary injury after SCI.

As suggested by Reviewer 1, we have reorganized the figures to make them more impactful and to ensure that the *Results* section is dedicated to describing the data of the main figures, with less focus on supplementary material. As a result of this, Fig. 1 was modified to include additional time points up to 35 days post-SCI, including when microglia are depleted, thus demonstrating that microglia, and not macrophages, are the predominant producers of the IL-1 α protein. Figures 2 and 3 were merged together and renamed as Fig. 2. Figure 5 was merged with Suppl. Figs. 2 & 3, which has now been renamed as Fig. 4. Supplementary Fig. 6 has become a main figure, renamed as Fig. 9, and was reinforced with additional immune cell depletion experiments using a more specific approach to deplete neutrophils, as requested by Reviewer 3. Figure 10 and Suppl. Fig. 7 were merged and renamed as Fig. 10. Finally, four

new supplementary figures (Suppl. Figs, 2-4 and 6) were added to the paper to give further details on our results, showing the effects of central IL-1 α delivery and/or SCI on plasma cytokine/chemokine levels, glial/neuronal cell activation, neutrophil infiltration and OL cell death as a function of sex or spinal cord anatomy, and blood-spinal cord barrier (BSCB) leakage.

These experiments rely on direct injection of IL-1 α as being representative of the SCI environment, however when repeated in an SCI model many of the observations did not replicate. While the mechanisms described may still be applicable, their therapeutic efficacy in the complex secondary injury cascade may be limited. Not addressed or acknowledged in the discussion.

We agree that acute central delivery of a single DAMP, in our case the injection of IL-1 α , likely does not result in a second injury cascade quite as complex as one produced by SCI. Although the existence of several other DAMPs (e.g. nucleic acids, nucleotide derivatives such as ATP, HMGB proteins, IL-33, etc.) in the injured spinal cord setting was clearly stated in the *Introduction* section, and now even more emphasized in the revised *Introduction* (see page 3, lines 66-81), we were guilty of not addressing this issue further in the *Discussion*. This has now been rectified as follows (page 25, lines 583-584): “It must be noted, however, that the secondary injury cascade occurring after traumatic SCI is complex and likely involves multiple pathways.” Also, on page 26 (lines 599-603): “Likewise, it will be important to identify other downstream effector molecules like ROS that directly mediate CNS cell toxicity, as the therapeutic efficacy of targeting a single molecule may be limited in the complex secondary damage following CNS injury. Supporting this line of research is a recent study by Guttenplan et al., revealing that long-chain saturated lipids partially contribute to astrocyte-mediated toxicity *in vivo* in the retinal injury model¹.”

Nevertheless, we point out that many of the observations made using the IL-1 α injection model are comparable to those found after SCI, as both models had a relatively identical time course in terms of plasma cytokine/chemokine levels, astrocyte reactivity, massive infiltration and recruitment of neutrophils, BSCB leakage, ROS production, and loss of OLs and their replacement (see responses below). We also point out that inhibition of IL-R1 signaling, either transgenically or pharmacologically, consistently improved recovery of locomotor function and reduced lesion volume after SCI in mice. Altogether, this suggests that IL-1 α is a major player in the SCI environment.

The effect of several of the KI/KO models is limited (i.e. less than 30% increase in gene expression or less than 20% decrease). Could this not influence the extent to which these models show a change in response compared to control? Authors claim the lack of effect indicates the cell type is not directly

involved in the effects but I'm not convinced this data is strong enough to support that. This is touched on in the discussion: "As the knockout will only be partial, it cannot be determined with certainty whether the response observed is due to the receptors that are still expressed by the targeted cell population or another cell type. Here, we circumvented part of this problem by using restored and knockout mouse models as well as primary cell lines, thus ensuring an additional level of confidence and reproducibility to our results." The repeated lines tested are supportive, but still the issue remains.

Initially lacking the appropriate technology, our original paper assessed the efficiency of Cre-mediated recombination in astrocytes and OPCs isolated from neonatal mice, rather than from adult mice of 2-3 months of age who were subjected to either IL-1 α injection or SCI. Therefore, the recombination efficiency was likely underestimated in our initial analyses, seeing as some of the mouse lines used constitutively expressed Cre recombinase (e.g. *Gfap*^{Cre} mice), and were able to recombine the gene of interest from development to adulthood. To address this issue, we performed additional experiments in which we isolated brain astrocytes from adult *Gfap*^{Cre::Il1r1^{fl/fl} and *Gfap*^{Cre::Il1r1^{r/r} mice, as well as from their respective controls, *Il1r1^{fl/fl}* and *Il1r1^{r/r}* mice, using the semi-automated gentleMACS™ Octo Dissociator and standardized protocols developed by Miltenyi Biotec, along with both the Adult Brain Dissociation Kit and Anti-ACSA-2 MicroBead Kit. Following the isolation of astrocytes from the brain of adult mice, genomic DNA was extracted and the floxed *Il1r1* allele or sequence causing *Il1r1* gene disruption was amplified using qPCR. As shown in the **modified Fig. 8b**, *Il1r1* gene expression was restored to approximately 60% of normal levels in brain astrocytes of *Gfap*^{Cre::Il1r1^{r/r} mice at 8-10 weeks of age compared to the previously reported 30% in neonates at postnatal days 0-2. Likewise, normal *Il1r1* gene expression was reduced by nearly 80% in brain astrocytes of *Gfap*^{Cre::Il1r1^{fl/fl} mice at 8-10 weeks of age compared to the previously reported 20% in neonates at postnatal days 0-2. The effect of the conditional knockout and restored models herein described is therefore far superior to previously reported in the original paper.}}}}

Regarding mouse lines, our paper highlights a common problem in the field of transgenic mice models, as only a few other papers (if any) validated the extent of gene deletion/restoration at the genomic DNA level in a cell-specific manner. The Cre mouse lines used in our research mirror those that other neuroscience research groups are using, thus the problem raised by Reviewer 1 is presumably common to all those using Cre-LoxP-based mouse models, it is just not acknowledged.

In the methods it says that male and female mice were used. Male and female rodents have been shown to have very different neuroimmune responses. It is not clear whether sexes were analyzed separately to take into account possible sex differences.

The reviewer is correct in pointing out that male and female rodents sometimes exhibit different neuroimmune responses. This was once again demonstrated by Tansley and colleagues in the February 2022 issue of Nature Communications, wherein sex differences resulted in differential gene expression of microglia in mouse and human spinal cords after peripheral nerve injury ². The **new Suppl. Fig. 4** now shows the total number of infiltrated Ly6G⁺ neutrophils and the total number of Olig2⁺CC1⁺ mature OLs in the spinal cord of C57BL/6 mice injected with either PBS or rmIL-1 α according to biological sex. Since no statistical difference was observed between sexes, data from mice of both sexes were pooled by experimental paradigm to increase power.

Introduction Summary: This lab has previous work demonstrating release of IL-1 α by microglia. Current goal is to determine the extent to which this release drives secondary degeneration via astrocytes and endothelial cells.

This section did not provide enough overview of neuroinflammation and the role of cytokines/IL-1 α to convince me that their question is novel or brings new information to the field. There are similar papers in existence.

As pointed out by the reviewer, our previous work has suggested the release of IL-1 α by microglia after SCI ³. This was based on the colocalization of the IL-1 α protein with markers of microglia/macrophages such as Iba1, CD11b and Cx3cr1-eGFP, and criteria such as cell morphology (e.g. ramification of processes) and the early onset of IL-1 α expression post-SCI. A previous study from Luheshi et al. ⁴ also arrived at the same conclusion, using the same tools and criteria in a mouse model of ischemic brain injury. However, the cell lineage tracing methods used here are more advanced, and allowed us to discriminate with certainty microglia from resident and infiltrating macrophages. These tracing methods were unavailable at the time of our first submission, thus it is only with our current work we can now claim that microglia are the main source of the alarmin IL-1 α .

The goal of the present study was to determine the extent to which IL-1 α release after SCI drives secondary degeneration and to identify the cellular and molecular mechanisms underlying this effect. As far as we know, no other papers have investigated this mechanism in the context of traumatic CNS injury. The principle that microglia-derived IL-1 α could drive a neurotoxic reactive astrocyte phenotype in various neurodegenerative diseases was recently suggested by Liddel and colleagues, and was

appropriately cited in our paper ⁵. This particular study and the following work by the same authors did not investigate, however, the role of IL-1 α in secondary degeneration occurring after SCI and brain trauma. Nonetheless, they did report that reactive astrocytes drive the death of retinal ganglion cells in cases when the optic nerve is crushed and in the bead occlusion model of glaucoma, an effect that was prevented in *Il1a*^{-/-} *Tnf*^{-/-} *Clqa*^{-/-} triple knockout mice ⁶. Furthermore, they showed that long-chain saturated lipids partially contribute to astrocyte-mediated toxicity *in vivo* in the retinal injury model ¹. Our work has highlighted yet another mechanism by which reactive astrocytes promote the death of mature OLs, this time in the context of CNS injury, through IL-1 α -mediated release of reactive oxygen species. The two studies discussed above, one of which was published after the submission of our paper, have been discussed in more detail in the revised manuscript (see lines 591-594 on page 26 and lines 601-603 on page 26). The text was further modified to provide a better overview of neuroinflammation and the role of IL-1 α in the context of CNS injury (page 3, lines 69-81): “Recent work in proteomics has predicted that the transition from primary to secondary damage after SCI and the accompanying neuroinflammatory responses are triggered by the release of danger-associated molecular patterns (DAMPs, also referred to as alarmins) from disrupted cells ⁷. The most prevalent DAMPs in the injured CNS are nucleic acids and nucleotide derivatives, including ATP, high-mobility group box (HMGB) proteins, S100 class of proteins, interleukin (IL)-33 and IL-1 α ^{3, 8, 9, 10, 11, 12}. However, blocking these individual DAMPs or their receptors has only shown modest improvements in protecting CNS cells from death and improving functional recovery after injury ¹². In some cases, these interventions can worsen neurological outcomes ^{11, 13}, with many conflicting studies reporting either beneficial or detrimental effects in mutant mouse lines with SCI ^{14, 15, 16}. The only exceptions to this ambiguity are IL-1 α and ATP and their receptors, whose neutralization *in vivo* consistently reduced neuroinflammation and secondary damage and improved neurological outcome after CNS injury ^{3, 10, 17, 18}. The relationship between these processes remains poorly understood and requires further investigation.”

This statement: The transition from primary to secondary damage is triggered, at least in part, by the release of the intracellular content of necrotic cells, in particular danger-associated molecular patterns (DAMPs, also referred to as alarmins)

could give a clearer picture of the importance of IL-1a in the secondary process aside from a basic generalization.

Recent work in proteomics has predicted that the transition from primary to secondary damage, and the accompanying neuroinflammatory responses after SCI are triggered by the release of DAMPs from

disrupted cells⁷. DAMPs constitute a large family of endogenous molecules, yet the most prevalent among them in the injured CNS are IL-1 α , IL-33, nucleotide derivatives (e.g. ATP), high-mobility group box 1 (HMGB1), and S100 class of proteins (e.g. S100A4). Blocking these individual DAMPs or their receptors has only shown modest improvements in protecting CNS cells from cell death and improving functional recovery after injury¹², and in some cases these interventions can worsen neurological outcomes^{11, 13}, with many conflicting studies reporting either beneficial or detrimental effects in mutant mouse lines with SCI^{14, 15, 16}. The sole exceptions to this are IL-1 α and ATP. Blocking these DAMPs or their signaling receptors always yields unambiguous outcomes, resulting in reduced neuroinflammation and secondary damage and improving neurological outcome^{3, 10, 17, 18}. To summarize, we agree with the reviewer and have now revised the *Introduction* section to address these issues (see page 3, lines 69-81).

Figure 1: Goal is to show that IL-1a is released by microglia and not macrophages. Used a tamoxifen conditional expression model to label microglia (under promoter *Cx3cr1*) with TdT. Induction 1 month prior to injury, animals sacrificed 4 hours post SCI to observe maximal cytokine release. Stained for IL1a. 95% of IL-1a expressing cells were TdT positive, suggesting microglial identity. Following microglia depletion the number of IL-1a cells significantly decreased. High resolution confocal images suggest IL-1a, TdT positive cells are damaged microglia based on abnormal morphology.

The addition of arrows to these images (C-L) to highlight relevant morphology would make it easier to interpret. These IHC images alone are insufficient to resolutely claim that the microglia are the primary releasers of IL-1a. Nothing has been done here to show that macrophages aren't also releasing it.

Arrows have been added to the **modified Fig. 1** to highlight relevant cell morphologies reminiscent of activated or damaged cells. Figure Legend 1 has been revised accordingly.

Regarding the second point, we have performed additional SCI experiments in *LysM-eGFP::Cx3cr1^{CreER}::Rosa26-TdT* triple transgenic reporter mice and performed immunofluorescence confocal microscopy against IL-1 α on spinal cord tissue sections derived from these animals. As shown in the **modified Fig. 1**, the IL-1 α protein was almost always colocalized with TdT⁺ microglia, but not detected in blood-derived eGFP⁺ neutrophils and monocytes, thus pointing to microglia as its main (if not only) cellular source after SCI. Several later time points post-SCI were also added to this analysis, as requested by Reviewer 2 (see Point #4), demonstrating that this situation did not change over time.

Figure 2: Goal is to show what the cellular response is to IL-1a. Injected IL-1a into cisterna magna of WT mice. Looked for *cfos* expression (indicates transcriptional activation) in spinal cord with IHC.

Significant increases in cfos in Olig2 (OL) and Sox9 (astro) positive cells compared to control PBS injected animals. Increase peaked at 4 hrs post injection and was no longer significant by 24hrs. ****Olig2 alone is insufficient to resolutely indicate activation of OLs. Olig2 is not specific to OLs.****

We have now included both the Olig2 and CC1 markers in the **new Fig. 2** and **new Suppl. Fig. 3** to confirm that the Fos protein is expressed by mature OLs following central IL-1 α administration.

Figure 3: Injected IL-1a into cisterna magna of WT mice. Looked for neutrophil infiltration. Significant increases compared to PBS controls after 24 hours in spinal cord parenchyma and neurovascular space. Returned to normal by 3 days. Demonstrates order of events: IL-1a release followed by glial activation followed by neurotrophil infiltration. Also quantified OL cell count in Olig2 CC1 positive cells and found a significant decrease (40%) in the number of mature OLs. Data not shown: OL loss increased as IL-1a concentration increased.

****Figures 2 and 3 would be more impactful if combined.****

Done, please see the **new Fig. 2**.

Figure 4: BrdU experiment to examine OPC proliferation following IL-1a injection. Olig2 Brdu positive cells increased significantly 3 days post IL-1a injection. Ki67 staining supported this. OL number returned to baseline by 5 days post injection. Takeaway is that robust proliferation is able to restore IL-1a mediated OL loss within days.

****1) This is an example where cytokine injection does not necessarily equal SCI, where evidence of full OL recovery is controversial and mixed. There is no context provided for how this finding is related to actual injury. Discussion could be more robust. 2) Olig2 alone is not specific for OLs. 3) The limitations/translatibility of this model could be discussed further.****

As pointed out earlier, we agree that the secondary injury cascade that develops after SCI is likely more complex than the responses observed after acute central delivery of a single DAMP, and that IL-1 α injection does not necessarily equal SCI. Nevertheless, many of the observations made using the IL-1 α injection model are similar to what is seen after SCI (see our response to the reviewer's general impressions). We argue here that this is also the case for the loss of OLs and their near complete restoration within days, as this pathophysiological process has been described before in mice subjected to contusive SCI. Indeed, a landmark study by Lytle and Wrathall¹⁹ has reported the following: "Total NG2⁺ cells was decreased at 24 h after injury, not only at the epicentre but at least as far as 2 mm rostral

and caudal to it (Fig. 6; Table 2). By 3 DPI, the NG2⁺ cell count was back to normal at most locations but was significantly higher (2-fold higher than at 24 h) in residual white matter 1–2 mm rostral to the epicentre. By 7 DPI, total NG2⁺ cell density was significantly higher than in uninjured control tissue from 0.5 mm caudal through to at least 2 mm rostral to the epicentre”. Taken together with our original data revealing that the number of proliferating (BrdU or Ki67) Olig2⁺ cells was increased several-fold in IL-1 α -injected mice at 3 days (**modified Fig. 3**), our new data showing that >85% of these proliferating Olig2⁺ cells at 3 days post-injection are NG2⁺ indicate that these two animal models have more in common than previously acknowledged. Along this line, Lytle and Wrathall also made the following additional observation: “No BrdU labeling was observed in the white matter of uninjured controls (not shown), and very little in the residual white matter at 1 DPI (Fig. 7, panel A1). However, at 3 DPI, many BrdU⁺ nuclei were detected (Fig. 7, panel A3), including those of cells that expressed NG2 or Cd11b (Fig. 7, panels B1 and 2). The maximum number of NG2⁺ BrdU⁺ cells was observed 1.5 mm rostral to the injury epicentre, at 3 DPI. At this location and time, 55% of BrdU⁺ cells were NG2⁺ but only 12.5% were Cd11b⁺. By 7 DPI, the number of NG2⁺ BrdU⁺ cells, while still greater than normal, had dropped to levels seen at 1 DPI. Temporally, the overall density of newly proliferated cells in the ventrolateral white matter in the cord adjacent to the injury site increased up to 3 DPI and remained at the same level at 4 DPI, before falling at 7 DPI. By 4 weeks after SCI, evidence of macroglial loss in the residual white matter at the lesion epicentre and at tissue adjacent to it had largely or completely disappeared. In white matter at the epicentre, the density of CC1⁺ OLs appeared closer to normal....”. Using a spinal contusion model in mice and rats, the McTigue laboratory has shown that proliferating OLs can be detected in tissues adjacent to lesion cavities within 3 days of SCI^{20,21}. By day 14 post-injury, the number of mature OLs in lesion borders equaled or even exceed those in the surrounding spared tissues or normal intact tissues^{15,20,22}. Therefore, although the timing and scale of these responses may change between the two models, we claim that OL loss followed by recovery is a normal process in both models of SCI and i.c.m. IL-1 α injection. The *Discussion* section has been modified as follows to provide more context into how this finding relates to actual SCI (page 23, lines 523-535): “BrdU pulse experiments and Ki67 immunostaining allowed us to confirm both the proliferation of OPCs, and their subsequent differentiation into newly matured Olig2⁺ CC1⁺ OLs. This rapid turnover is reminiscent of changes observed in the acutely injured spinal cord, where robust replacement of mature OLs by OPCs can be seen^{19,20,23}. Although the loss, and ensuing regeneration of OLs from OPCs is present in cases of SCI and i.c.m. IL-1 α injection, the timing and scale of these responses may differ between models. However, whether remyelination positively impacts locomotor recovery after SCI remains open for debate²⁴. A better understanding of the mechanisms driving the replacement of myelin and OL loss would be very

beneficial, contributing to the understanding of various other pathological contexts. Collectively, these data suggest that SCI triggers a local response in the CNS, characterized by the release of alarmins such as IL-1 α that in turn induce the death of mature OLs. Their rapid replacement afterwards by newly matured OLs may allow for greater efficiency in remyelinating the injured spinal cord.”

Regarding the second point, as indicated above, we have performed additional immunostaining against Olig2, NG2 and BrdU to address the issue of specificity raised by the reviewer (**modified Fig. 3**).

Suppl. Figure 2: Goal to determine the role of the R1 receptor in an SCI model. WT, IL-1a KO, and IL-1aR KO mice underwent SCI and recovery followed over 1 month. Both KO groups had better locomotor recovery than WT. This further implicates the cytokine and its receptor in the inflammatory response.
This seems like a powerful finding and I’m not sure why it’s in supplemental.

We agree. The data originally presented in Suppl. Figs 2-3 were combined with those of Fig. 5 and are now part of the **new Fig. 4**, displaying our findings more impactfully.

Suppl. Figure 3: Goal to determine what cell types R1 is expressed in. Immunoblot analysis of IL-1R1 showed expression in endothelial cells, OPCs, astrocytes, and microglia. This line concerns me: “Intriguingly, the protein detected using the anti-IL-1R1 polyclonal antibody in OPCs and astrocytes was of a slightly lower molecular weight than the predicted 75-kDa IL-1R1 protein, suggesting the possibility that cells of the glial-restricted lineage could express a different form of the receptor. Along this line, we point out that IL-1R3 is identical to IL-1R1 at the C terminus, but with a shorter extracellular domain and a predicted molecular weight of about 66 kDa.”
Is this suggesting that their immunoblot may have actually picked up IL-1R3 instead or in addition to IL-1R1? If this is a possibility, I think it needs to be determined for sure or this should not be reported as IL-1R1 expression alone.

We apologize for the confusion created in our description. By referring to IL-1R3, we meant that our team possibly detected a truncated splice variant of the mouse IL-1R1 receptor. This truncated form of IL-1R1 was first identified by one of the authors of this work, Dr. Ning Quan, and was referred to as IL-1R3 in the original paper describing and identifying it (Qian et al., PNAS, 2012). IL-1R3 expression is driven by an internal promoter in the *Il1r1* gene. We understand that the choice of the name IL-1R3 may have been unwise and confused the reviewer, seeing as several other groups in the field of IL-1 biology refer to the IL-1 receptor antagonist (IL-1Ra) when discussing IL-1R3 (IL-1Ra is coded by a completely

different gene). Therefore, we have modified the manuscript as follows to clarify this point (pages 11-12, lines 261-266): “Intriguingly, the protein detected using the anti-IL-1R1 polyclonal antibody in OPCs and astrocytes was of a slightly lower molecular weight than the predicted 75-kDa IL-1R1 protein, suggesting the possibility that cells of the glial-restricted lineage could express a truncated splice variant of the mouse IL-1R1 receptor, referred hereafter to as τ IL-1R1. Along this line, we point out that the Quan laboratory has previously shown that τ IL-1R1 is identical to IL-1R1 at the C terminus, but with a shortened extracellular domain and a predicted molecular weight of about 66 kDa²⁵.” Also on page 21 (lines 486-487): “Our immunoblotting findings confirm that IL-1R1 or a truncated splice variant of IL-1R1 is expressed by microglia, astrocytes and OLs...”

Figure 6: This and subsequent figures explore the response of specific cell types to IL-1a. Use mice expressing an inducible Cre recombinase under the control of the *Pdgfra* promoter to conditionally restore *IL1r1* gene expression specifically in OPCs of an *IL1r1/3* KO.

Is it problematic for both R3 and R1 to be knocked out since they've focused on R1? No characterization is done to determine R3 effects or expression.

** Report the KI shows a 20% increase in R1 expression determined by qPCR in both OPCs and O4 OLs. However, this did not reverse the loss of neutrophil infiltration or OL death following injection with IL-1a.**

Figure C is missing the label for PBS vs injection.

Claim that the lack of response to “recovered” KO lines to IL-1a means that IL-1a is not directly responsible for the OL death observed in Fig. 3. With only a 20% recovery of gene expression and 40% recombination, especially without any presentation of the actual qPCR data that allowed them to make this calculation, it seems equally likely that the lack of reversal could have happened because the gene was only partially restored. Furthermore, it is not clear to me how they calculated % recovery from the raw qPCR data and this information would be useful.

Please see response to the previous comment above for clarification on IL-1R3 versus the truncated form of IL-1R1. The labels for PBS and rmIL-1 α were added to panel C.

The issue of the poor recombination efficiency estimated for adult astrocytes and OLs, based on calculations made from the isolated glial cells of neonatal mice in the original paper, is an important matter that we have now addressed through additional experiments. As previously discussed in our response to the reviewer's general impressions, recombination efficiency was misjudged in our initial estimations because of an experimental design flaw in which neonatal astrocytes and OLs were used to

assess gene expression changes at the genomic level instead of adult glial cells. This design choice was dictated by our lack of expertise in isolating adult CNS astrocytes and OLs. To properly address this question, we have since developed a procedure that combines the semi-automated gentleMACS™ Octo Dissociator and standardized protocols developed by Miltenyi Biotec. As shown in the **modified Fig. 8**, *Il1r1* gene expression was restored to approximately 60% of normal levels in brain astrocytes of *Gfap*^{Cre::Il1r1^{r/r}} mice at 8-10 weeks of age compared to the previously reported 30% in neonates at postnatal days (P) 0-2. Likewise, *Il1r1* gene expression was reduced by nearly 80% of normal levels in brain astrocytes of *Gfap*^{Cre::Il1r1^{fl/fl}} mice at 8-10 weeks of age compared to the previously reported 20% in neonates. The effect of the conditional knockout and restored models used herein is thus far superior to that previously described in the original manuscript. Unfortunately, due to the pandemic, our animal research facility suffered major cuts in staff and available services and some of our “less promising” mouse lines had to be relinquished, this being the case for the *Pdgfra*^{CreER::Il1r1^{r/r}} mice.

Although we could not reassess recombination efficiency in adult OLs of *Pdgfra*^{CreER::Il1r1^{r/r}} mice, and efficiency of OLs isolated from P0-2 neonates was estimated to be about 40%, we point out that treatment of primary OLs with rmIL-1 α was not toxic in cell culture experiments (see **modified Fig. 10E**). Thus, despite a possible low recombination efficiency in OLs of *Pdgfra*^{CreER::Il1r1^{r/r}}, we are confident that OL cell death is not directly mediated by IL-1 α .

Figure 7: Goal is to determine whether IL-1R1 has autoregulatory effect on microglia (i.e. release of IL-1 α from microglia then binds to receptor on microglia) that contributes to secondary damage. Same inducible model as Figure 6, this time using Cx3cr1 promoter for conditional R1 expression in microglia. Report 65% restoration in gene expression. Similar to Figure 6, this increase in gene expression in microglia did not correlate to a reversal in response to IL-1 α for neutrophil infiltration or OL death. ****Given this result with a higher induction rate, I am more inclined to believe the results in figure 6 and that this suggests both cell types do not directly mediate the effects of IL-1 α . I don't think the possibility can be completely ruled out, however, that the increase in gene expression was just insufficient to observe a response. For both Figure 6 and 7, Figure C appears to have a slight downward trend, even though it doesn't reach significance.****

As mentioned above, we have tempered our conclusions as follows to fully address the reviewer's critique (page 24, lines 557-565): “It is important to keep in mind that despite being a powerful tool to decipher molecular mechanisms *in vivo*, Cre-reporter mouse lines have some limitations, the most important being that recombination in a specific cell type is often incomplete, particularly in the case of

tamoxifen-inducible CreER mice. This is especially limiting when aiming to knockout a gene of interest in a specific cell type *in vivo*, rather than restoring it. As knockouts are only partial, it cannot be determined with certainty whether the responses observed are due to the remaining receptors expressed by the targeted cell population, or those expressed on another cell types. Here, we circumvented part of this problem by using restored and knockout mouse models along with primary cell lines, thus ensuring an additional level of confidence and reproducibility of our results.”

Figure 8: Same inducible model as Figure 6, this time using Cdh5 promoter for conditional R1 expression in endothelial cells. Report 90% restoration in R1 gene expression. Also observed a 35% return in neutrophil infiltration and a slight increase in OL death following IL-1a injection.

**Is it expected for a 90% restoration in receptor expression in the KO to have such a small return to baseline response? What else accounts for the still high differences between the inducible model and the WT? **

The same animal model following SCI did not return to WT locomotor scores. This suggests endothelial cells are partially involved in the response to IL-1a. To further explore this, they utilized a transgenic IL-1R1 KO (rather than the inducible knock-in from the last few figures) model that decreased receptor expression by 50% compared to control in endothelial cells. Knocking out this receptor in endothelial cells significantly decreased neutrophil expression and OL death following IL-1a injection. This did not translate to locomotor recovery after SCI.

To address the reviewer’s comment, the experiment in which *Cdh5^{CreER}::Il1r1^{+/+}* mice were injected with either rmIL-1 α or PBS was replicated. Despite that and a 90% restoration in *Il1r1* gene expression in endothelial cells of *Cdh5^{CreER}::Il1r1^{+/+}* mice, we still observed a 25% return in neutrophil infiltration and a 25% increase in OL cell death (vs. ~50% in the WT + rmIL-1 α group) following i.c.m. delivery of IL-1 α . This partial but significant effect of endothelial IL-1R1 in IL-1 α -mediated neutrophil infiltration and OL loss, combined with the partial effect reported for astrocytic IL-1R1, indicates a redundant role of IL-1R1 in these two cell types. It thus suggests the existence of a complex interplay between endothelial and astrocytic IL-1R1 underlying the effects of IL-1 α in neuroinflammation and secondary damage after SCI. The text has been modified as follows to make this clearer (page 27, lines 620-622): “Instead, astrocytes and CNS ECs appear to drive OL cell death via an IL-1R1-dependent release of ROS and other molecules that have yet to be identified.” Also, on page 25 (lines 583-584): “It must be noted, however, that the secondary injury cascade occurring after traumatic SCI is complex and likely involves multiple pathways.”

Figure 9: Inducible KI model described previously, this time using GFAP promoter for conditional R1 expression in astrocytes. Report 30% return to normal R1 expression levels. There was no change in neurotrophil infiltration however OL death increased. There was also no effect on the SCI model locomotor scores. Again this was repeated in a transgenic KO line in astrocytes. This line had reduced expression of the receptor by roughly 20%. KO of the IL-1R1 did not prevent neurotrophil infiltration but did decrease OL death. This did not translate to any locomotor recovery following SCI. Concludes that astrocytes partially mediate the response to IL-1a.

Same concerns mentioned previously – low efficacy of KO and KI make me wonder if part of the lack of an observed effect was for this reason and not related to the mechanisms of IL-1a action.

The scope of this criticism is considerably reduced in light of the new experiments we have performed, in which *Il1r1* gene expression was restored to ~60% of normal levels in astrocytes of *Gfap^{Cre}::Il1r1^{r/r}* mice, while it was reduced by ~80% in astrocytes of adult *Gfap^{Cre}::Il1r1^{fl/fl}* mice. We refer to our response to the comment related to Fig. 6 for more information and details.

Reviewer 2:

The manuscript by Bretheau and colleagues provides a mechanistic investigation of the role of IL-1a – IL-1R1 signalling in oligodendrocyte death in two models of CNS inflammation, traumatic spinal cord injury and injection of IL-1a into the cisterna magna. The present study advances significantly from previously published work, addressing outstanding various knowledge gaps at both the cellular and molecular level. Overall, this is an elegant study, containing a very large body of work using numerous mouse lines/crosses to conditionally delete as well as restore expression of IL-1R1, thereby identifying relevant the cellular players and/or pathways. A number of questions and suggested experiments that need addressing and/or doing is provided below:

Figure 1: The authors demonstrate that microglia are the source of IL-1a very early after SCI.

- (1) Is IL-1a also actively secreted rather than passively released by microglia?
- (2) If not, is its contribution to OL death only acute as microglia numbers are rapidly restored during the first week of SCI (previous work by this group), but OL loss peaks only at one week (Ref 25) and continues for weeks post injury?

(3) Related to previous points, do the authors believe that IL-1 α still drives OL death here, or would it be ROS produced via different pathways?

(4) Inclusion of IL-1 α protein levels and cellular sources at later time points post SCI would be desirable, including when microglia are depleted and for SCI experiments where IL-1R1 is conditionally removed (or restored) from select cell types.

(5) To convincingly demonstrate IL-1 α release by microglia, and the subsequent impact on OL death, the authors should aim to target/delete the ligand in microglia.

We thank the reviewer for these positive comments regarding the elegance of our study and its importance in advancing the vast field of CNS inflammation. The reviewer brings up an interesting set of questions, which have been addressed as follows:

- 1) The reviewer asks whether IL-1 α is actively secreted rather than passively released by microglia. Although we agree that this is an interesting point, we would need to know more about the mechanisms regulating active versus passive release of IL-1 α in microglia to be able to definitely answer this question. Unfortunately, we know very little about said mechanisms at this point in time. We refer to the work of the known leaders in this subject to demonstrate the complexity of this question, and why it cannot be answered without performing a separate and time-consuming study^{26, 27, 28}.
- 2) The reviewer asks whether the effect of the alarmin IL-1 α was more acute than chronic, and whether this correlates with the death of mature OLs after SCI. This is a very relevant point that we began to address in our responses to Reviewer 1 (see responses to comments related to Figures #1 and 4). First, we have performed additional experiments using C57BL/6 mice and *LysM-eGFP::Cx3cr1^{CreER}::Rosa26-TdT* reporter mice that were euthanized at several later time points post-SCI to investigate the time course of IL-1 α protein expression (see **modified Fig. 1**). These new data revealed that IL-1 α protein was expressed almost exclusively by microglia exhibiting morphological profiles of cell activation, damage or death during the first day post-SCI. This suggests an acute effect of IL-1 α on neuroinflammation and death of mature OLs after SCI, in line with expectations of the role of an alarmin. Second, as stated above, the work of Lytle and Wrathall¹⁹ has established that day 1 post-SCI is the timepoint wherein most mature CC1⁺ OLs are lost, a drop that slightly continues until day 7 during the chronic phase of SCI before being restored to values found in uninjured mice. These findings have been confirmed by several groups including ours³, with some of the best experts in the field even suggesting that OL counts return to normal near the lesion epicenter by day 14 post-

SCI^{15, 20}. Therefore, although we do not refute the scientific evidence that OLs continue to die for weeks after SCI, and that other mechanisms may be involved in this continuous death, we point out that the most important decrease in OL cell numbers occurs at day 1 post-SCI, supporting our claim that IL-1 α plays a key role in this process. This has been clarified in the revised *Discussion* section (page 23, lines 527-529): “Although the loss, and ensuing regeneration of OLs from OPCs is present in cases of SCI and i.c.m. IL-1 α injection, the timing and scale of these responses may differ between models.” Also on page 25 (lines 583-584): “It must be noted, however, that the secondary injury cascade occurring after traumatic SCI is complex and likely involves multiple pathways.”

- 3) As explained above, we do not make claims about the potential role of IL-1 α in inducing OL cell death beyond the times we have specifically detected its expression, even if the triggering of a complex pathological cascade leading to cell death and involving multiple cell types could take hours or even days to manifest.
- 4) In the **modified Fig. 1**, we now provide new immunofluorescence data showing a complete time course of IL-1 α protein expression at later time points post-SCI, and we identify its cellular source by confocal microscopy. As requested by Reviewer 2, we also generated new *in vivo* tissue from spinal cord injured *Cx3cr1^{CreER}::R26-TdT* mice depleted in microglia using the PLX5622 diet to further validate the cellular source of IL-1 α (see **modified Fig. 1F**).
- 5) The reviewer asks us to generate another mouse model by crossing the *Cx3cr1^{CreER}* line for microglia-specific gene deletion with the *Il1a^{fl/fl}* conditional line generated by the laboratory of Dr. Emmanuel Pinteaux or Dr. Susan Burke^{29, 30}, which are not available from a commercial source. In order to generate and characterize these mice and then promptly evaluate signs of neuroinflammation, OL loss and functional recovery in our two models (i.e. central IL-1 α injection and SCI), years of additional work would be required and huge amount of new data would be produced. We truly appreciate the suggestion, yet we believe it is beyond the scope of this paper and would most likely not alter our conclusions.

Figure 2: Immediate early gene protein Fos is upregulated in OLs following IL-1 α injection.

- (6) Is SOX9 specifically expressed by astrocytes or do e.g. OPCs also express it? Can another astrocyte marker be used?
- (7) Also, was Fos expression only observed in in the white matter? No expression in neurons and/or microglia?

(8) How do IL-1a levels being injected compare to those seen after SCI in CSF as well as plasma? Are they comparable? The authors should measure plasma levels of IL-1a when this is injected into the cisterna magna.

(6-7) To resolve these issues, we performed additional immunofluorescence analyses in which we assessed the colocalization of the Fos protein with cell-specific markers and quantified the number of activated astrocytes (Sox9⁺ Fos⁺), OLs (Olig2⁺ CC1⁺ Fos⁺), microglia (Iba1⁺ Fos⁺), and neurons (NeuN⁺ Fos⁺). The additional data is presented in two new figures (**new Fig. 2** and **new Suppl. Fig. 3**). The *Results* section was modified as follows to reflect these additions (pages 7-8, lines 158-174): “Immunostaining for Fos in spinal cord tissue sections from mice injected with IL-1 α revealed that Sox9⁺ astrocytes are activated as early as 1 hour post-injection (Fig. 2A), while activation of Olig2⁺ CC1⁺ oligodendrocytes (OLs) was slightly delayed, occurring at 4 hours (Fig. 2B). The peak of activation for both glial cell types was observed at 4 hours, with a total of 187.7 ± 13.3 Fos⁺ Sox9⁺ astrocytes/mm² and 75.2 ± 1.5 Fos⁺ Olig2⁺ CC1⁺ OLs/mm² in the thoracic spinal cord white matter (Fig. 2A-B & Suppl. Fig. 3A-G). The number of activated astrocytes and OLs decreased afterward to 4.4 ± 1.4 cells/mm² and 19.3 ± 2.8 cells/mm², respectively, at 24 hours post-injection. In contrast, we found no evidence of microglia activation after IL-1 α injection, as demonstrated by the absence of colocalization of Fos with Iba1 immunostaining at 1, 4 and 24 hours after IL-1 α injection (Fig. 2C & Suppl. Fig. 3H-J). For all time points examined, nearly zero Fos⁺ cells were detected in the spinal cord white matter of PBS-injected mice. Unlike glial cells, neuronal activation peaked at 24 hours post IL-1 α injection, as determined by the upregulation of the Fos protein in NeuN⁺ cells of the spinal cord gray matter (Fig. 2D & Suppl. Fig. 3K-M). However, this was the final time point we analyzed, and peak neuronal activation may well have occurred later. Thus, it appears that cell type activation in response to IL-1 α injection follows a precise timeline, from astrocytes to OLs and then to neurons.”

Regarding the specificity of the Sox9 immunostaining, we refer to the work of the Nedergaard and Arenkiel groups who presented solid evidence that the transcription factor Sox9 is specifically coexpressed with Aldh1L1 in the adult CNS, indicating that Sox9 expression is restricted to astrocytes^{31, 32}. We utilized additional markers of astrocytes, such as GFAP and vimentin, but these stained foot processes rather than cells bodies of reactive astrocytes, thus preventing colocalization with the Sox9 and Fos proteins. While Aldh1L1 is likely the current best and most specific marker of astrocytes, there are no good commercial antibodies available to detect the Aldh1L1 protein, meaning that *Aldh1L1* fluorescent reporter mice should have been used. This was, however, incompatible with our current approach.

(8) As requested by both Reviewer 2 and Reviewer 3, we now provide in **new Suppl. Fig. 2** plasma levels of IL-1 α and 31 additional cytokines and chemokines under naïve conditions, after SCI or after IL-1 α injection at 3 different time points post-SCI/injection (1, 4 and 24 hours) measured using a multiplex laser bead assay and the Mouse Cytokine 32-Plex Discovery assay from Eve Technologies. This new dataset shows that SCI and central delivery of IL-1 α induce a similar profile of cytokines/chemokines in the plasma, although the time course was slightly accelerated in mice directly injected with the alarmin. We point out that it was not possible to extract CSF in sufficient amounts due to the prior i.c.m. puncture of the meninges, thus preventing us from measuring cytokine/chemokine levels in the CSF compartment.

Figure 3: IL-1 α injection results in Ly6G infiltration and loss of OL.

(9) The authors convincingly show neutrophil recruitment and OL loss, but similar effects have been reported following injection of IL-1 β into the CNS. What is the specificity of the observed effect?

(10) Also, taken in conjunction with Figure 10, do the authors have insights as to when astrocytes begin to express ROS following IL-1 α injection; the peak of Fos activation appears at 4h and the OL death is apparent at 24h post injection.

(9) This reviewer raises a very important point. To address this, we have performed additional *in vivo* experiments in which either rmIL-1 α or rmIL-1 β was injected at different concentrations into the CNS of C57BL/6 mice. As shown in the **new Suppl. Fig. 4I**, the concentration of IL-1 β had to be increased 2.5-fold that of IL-1 α in order to mimic its effect on OL loss. Still, IL-1 α and IL-1 β injected at 20 ng/ μ l induced recruitment of a similar number of neutrophils (**new Suppl. Fig. 4H**), suggesting a possible disconnect between the infiltration of neutrophils and OL cell death (see our response to Comment #1 of Reviewer 3). In the same regard, we previously showed that deletion of the *Il1b* gene has no long-term beneficial effect on functional recovery after SCI, while *Il1a*^{-/-} mice recover significantly better than WT mice over a 35-day period (see **new Fig. 4I-J**). This occurred despite the fact that the inflammatory response was equally compromised in both *Il1a*^{-/-} and *Il1b*^{-/-} mice³, thus reinforcing the idea that separate mechanisms regulate neutrophil recruitment and OL loss, and pointing to a conclusion that IL-1 α is the most important IL-1 cytokine in secondary degeneration. To reflect these additions, the *Results* section has been modified as follows (page 9, lines 211-215): “In contrast, the concentration of IL-1 β had to be increased 2.5-fold that of IL-1 α to mimic its effect on OL loss. Still, IL-1 α and IL-1 β injected at 20 ng/ μ l

induced similar neutrophil recruitment, suggesting separate mechanisms regulating infiltration of neutrophils and OL cell death (Pearson's correlation, $p=0.1813$; Suppl. Fig. 4J)."

(10) As requested, we have carried out additional *in vitro* experiments in which we cultured primary murine astrocytes and then measured ROS production upon IL-1 α stimulation. As shown in the **modified Fig. 10F**, astrocytes begin to express ROS at 4 hours post-treatment with IL-1 α and continue to express them until 24 hours. This new information is now provided in the revised *Results* section, as follows (page 19, lines 431-437): "To assess the potential involvement of ROS in cell death of mature spinal cord OLs, we first examined whether IL-1 α can trigger the production of ROS by astrocytes *in vitro*. For this, we cultured primary murine astrocytes in the presence of either rmIL-1 α or vehicle, and then measured the production of ROS at various time points using the CellRox assay. We found that astrocytes begin expressing ROS at 4 hours post-treatment with IL-1 α , and this ROS production is sustained over the total observation period (i.e. 24 hours; Fig 10F)."

Figure 5: Delivery of Anakinra into the CSF blocks IL-1a induced neutrophil recruitment and OL death.

(11) Is anakinra also effective in SCI, i.e. under conditions where both IL-1a and IL-1beta are present?

Yes, this has been demonstrated this in our previous work by showing that a single i.c.m. injection of anakinra given 15 minutes after SCI was effective at reducing locomotor deficits compared to treatment with PBS³.

Figure 6: Restoration of Il1r1 within the oligodendrocyte lineage does not allow for IL-1a-mediated neuroinflammation and OL loss.

(12) The effect on neutrophil infiltration is convincing but considering the low level of restoration (20-25%) of PDGFR α on OPCs and estimate of 40% on OL plus the variation and low animal numbers, the authors should increase groups size for figure 6C.

(13) It would be also helpful to colour code individual data points (see 6b).

(12) We have many reasons to believe that the recombination efficiency reported in our initial analyses on OPCs and pro-OLs isolated from the brain of *Pdgfra*^{CreER::Il1r1^{f/r}} mice was underestimated. We refer to our response to the "General impressions" comment of Reviewer 1 for a more detailed explanation. Unfortunately, because of constraints related to the pandemic, the *Pdgfra*^{CreER::Il1r1^{f/r}} mouse line was discontinued two years ago, meaning we were not able to increase group size for Fig. 5C. However, we point out that our *in vitro* data showing that incubation of primary mouse OLs with IL-1 α does not lead

to cell death (see **modified Fig. 10E**) supports our conclusion that IL-1R1 in OLs does not allow for IL-1 α -mediated OL loss. Thus, despite a possible low recombination efficiency in OLs of *Pdgfra*^{CreER}::*Il1r1*^{r/r} mice, we are confident that OL cell death is not directly mediated by IL-1 α . Still, we have tempered our conclusions as follows (page 24, lines 557-565): “It is important to keep in mind that despite being a powerful tool to decipher molecular mechanisms *in vivo*, Cre-reporter mouse lines have some limitations, the most important being that recombination in a specific cell type is often incomplete, particularly in the case of tamoxifen-inducible CreER mice. This is especially limiting when aiming to knockout a gene of interest in a specific cell type *in vivo*, rather than restoring it. As knockouts are only partial, it cannot be determined with certainty whether the responses observed are due to the remaining receptors expressed by the targeted cell population, or those expressed on another cell types. Here, we circumvented part of this problem by using restored and knockout mouse models along with primary cell lines, thus ensuring an additional level of confidence and reproducibility of our results.”

(13) Done.

Figure 7: Microglia alleviate IL-1 α -mediated neuroinflammation and OL loss independent of IL-1R1 expression.

(14) Taken in conjunction with Figure 10 data and considering that microglia secreted factors drive A1 phenotypes, what happens to fos expression and C3 staining in astrocytes in absence of microglia?

This reviewer suggests the examination of Fos and C3 expression in the absence of microglia due to the fact that these cells are the likely source driving toxic reactive astrocytes and OL cell death. However, the role of microglia is bypassed in the model of direct IL-1 α injection into the CNS, meaning depletion of microglia would not provide the appropriate answers for the reviewer. Thus, we have quantified the number of C3-expressing astrocytes in spinal cord tissue sections collected from C57BL/6 mice, IL-1R1-knockout (*Il1r1*^{-/-}) mice, and mice with conditional restoration of IL-1R1 expression in either OLs, microglia, ECs or astrocytes at 1 day after PBS or IL-1 α injection. As shown in the **modified Fig. 10A**, i.c.m. delivery of IL-1 α in C57BL/6 mice increased the total number of Sox9⁺ cells expressing C3 by nearly two-fold at day 1 post-injection, an effect that was only replicated by restoration of IL-1R1 expression in astrocytes. This once again shows that IL-1 α is a potent inducer of astrocytic C3 upregulation, which is associated with a toxic reactive phenotype. Please note that substantial changes were made to the *Results* (page 18) and *Figure Legends* (pages 52-53) sections to reflect these additions.

Figures 8 & 9: IL-1 α -induced neuroinflammation and OL loss is partly mediated by endothelial and astrocytic IL-1R1.

(15) Is IL-1R1 deleted here from CNS endothelial cells only or globally?

(16) What is the systemic response to both IL-1 α injection and SCI under conditions where IL-1R1 is present, conditionally restored or deleted?

(17) Figure 8 and 9, data should be included the number of OLs, neutrophils, and also astrocyte activation after SCI.

(18) It would also be of interest to correlate OL number with BMS scores.

The reviewer raised several important concerns that we have addressed as follows:

(15) IL-1R1 expression was restored or deleted specifically in ECs using a transgenic mouse line in which a tamoxifen-inducible Cre (CreERT2) sequence was inserted into a large phage artificial chromosome that includes the *Cadherin 5* (VE-Cadherin) locus. VE-Cadherin is an endothelium-specific cell–cell adhesion molecule located at adherens junctions. Although more abundantly expressed in the CNS, expression can be found in ECs of peripheral organs as well in this driver mouse line. This is now specified in the revised *Methods* section (page 28, lines 639-642): “*Cdh5*^{CreER} mice (line #13073), in which the tamoxifen-inducible Cre recombinase is active in all ECs, were purchased from the Cancer Research Technology Repository at Taconic with prior consent of the creator of the mouse line...”

(16) We have performed additional experiments and can now provide new data in which systemic plasma levels of 32 total cytokines and chemokines were measured under naive conditions and after SCI or IL-1 α injection at 3 different time points (1, 4 and 24 hours) post-SCI/injection (**new Suppl. Fig. 2**). This new dataset shows that SCI and central delivery of IL-1 α induce a similar profile of cytokines/chemokines in the plasma, with the time course being slightly accelerated in mice directly injected with the alarmin.

(17) The reviewer is asking that we perform additional experiments in which another 24 groups of 8 mice (WT + Sham, *Il1r1*^{r/r} + Sham, *Cdh5*^{CreER}::*Il1r1*^{r/r} + Sham, WT + SCI, *Il1r1*^{r/r} + SCI, *Cdh5*^{CreER}::*Il1r1*^{r/r} + SCI) (WT + Sham, *Il1r1*^{fl/fl} + Sham, *Cdh5*^{CreER}::*Il1r1*^{fl/fl} + Sham, WT + SCI, *Il1r1*^{fl/fl} + SCI, *Cdh5*^{CreER}::*Il1r1*^{fl/fl} + SCI) (WT + Sham, *Il1r1*^{r/r} + Sham, *Gfap*^{Cre}::*Il1r1*^{r/r} + Sham, WT + SCI, *Il1r1*^{r/r} + SCI, *Gfap*^{Cre}::*Il1r1*^{r/r} + SCI) (WT + Sham, *Il1r1*^{fl/fl} + Sham, *Gfap*^{Cre}::*Il1r1*^{fl/fl} + Sham, WT + SCI, *Il1r1*^{fl/fl} + SCI, *Gfap*^{Cre}::*Il1r1*^{fl/fl} + SCI) would be euthanized at both 4 hours and 24 hours post-sham/SCI surgery to quantify astrocyte activation (4h time point), neutrophil infiltration (24h time point) and OL loss (24h

time point) in function of the distance from the lesion epicenter. This suggestion would include surgeries, animal perfusions, tissue processing, immunostainings and complex quantification on an additional 384 mice. Generating these mice and quantifying these responses in various conditions would likely require an additional year of work (the quantification of OLs alone takes 30-45 minutes per section, and typically 7 sections are analyzed per SCI animal [the lesion epicenter and at least 3 sections both rostral and caudal]). Considering the complexity of the SCI model, the transgenic mouse lines used and the pathological mechanisms under investigation, we do not think these additional experiments would influence the main conclusions of our paper, and this sentiment is illustrated by the behavioral data presented in the original Figs. 8 and 9.

(18) Please refer to our response above, as such a correlation would require killing animals at various time points, immediately after assessment of the BMS score, for a large number of conditions, and thoroughly analyzing and quantifying the number of OLs at various distances from the lesion epicenter.

Figure 10:

(19) Does injection of rmIL-1 α open the blood brain/spinal cord barrier? If so, C3 (and other proinflammatory factors) could come in from the circulation and contribute to OL death. This should be controlled for, also considering the observed effects in mice where IL-1R1 was selectively restored or deleted from endothelial cells.

(20) For the data presented in panel 10I, does NAC treatment, either alone or in combination with Anakinra, improve the neurological outcome from SCI?

(21) Considering that in vivo NAC treatment does not selectively target astrocytes, do cultured microglia (and other key cells in the inflammatory infiltrate after SCI) also produce ROS in response to IL-1 α stimulation?

(19) The reviewer raises a very important point. As suggested, we have investigated whether central delivery of rmIL-1 α i.c.m. affects the permeability of the blood-spinal cord barrier (BSCB) using intravital fluorescence videomicroscopy. As shown in the **new Suppl. Fig. 6**, we found that the administration of IL-1 α induces leakage of the BSCB to a 40-kDa dextran-Texas red fluorescent tracer at 6 hours post-injection, and that this response is dependent on endothelial IL-1R1. We point out, however, that the possibility that C3 and/or other proinflammatory factors could have infiltrated from the blood circulation into the spinal cord and subsequently contributed to OL cell death is unlikely given our new data showing that BSCB breakdown induced by IL-1 α injection is dependent on the presence of

circulating neutrophils, and that depleting these cells using an adapted version of the anti-Ly6G-based “Combo” depletion approach developed by Boivin et al.³³ resulted in no impact on OL loss (see **new Fig. 9** and our response to Comment #1 of Reviewer 3 below).

(20-21) We have not been able to test this explicitly considering the extent of additional work that was required for this revision. It was clear that some experiments were more urgent than others, and we needed to prioritize experiments that would move forward the main goal of our study, which is to investigate the cellular and molecular mechanisms underlying IL-1 α -dependent effects in the spinal cords of injured mice. Along with this, and keeping in mind our new and old data exhibiting that blood-derived myeloid cells are not responsible for killing mature OLs, we have revised the text to state that, in addition to astrocytes, ECs could be yet another source of ROS production after IL-1 α stimulation. We refer to the revised *Discussion* section on page 27 (lines 621-62): “Instead, astrocytes and CNS ECs appear to drive OL cell death via an IL-1R1-dependent release of ROS and other molecules that have yet to be identified.”

Reviewer 3:

The manuscript by Bretheau et al focuses on interleukin-1 α (IL1a) as a key factor in neuroinflammation and oligodendrocyte (OL) loss after spinal cord injury (SCI). After determining that microglial cells are the main source of IL1a during the subacute phase after SCI, they demonstrate that intra cisternal magna delivery of IL1a induces glial activation, neutrophil infiltration and mature OL death (replaced by OPC proliferation to compensate for OL death) through IL1R signalling in the mouse spinal cord. They further corroborate the detrimental role of alarmin Il1a within the context of SCI, demonstrating that lack of the cytokine or its receptor promotes better functional recovery after SCI. To demonstrate the cell types involved in OL death through IL1a signalling, the authors performed a careful and elegant series of gain and loss of function studies showing that detrimental effects of IL1a are mediated mostly through activation of endothelial and astrocytic IL-1R1 signalling. Finally, they identified toxic reactive oxygen species (ROS) expressed by astrocytes through IL1a signalling as one of the mediators responsible for OL death after SCI.

Although a role for IL1a in neuroinflammation after SCI has already been established (by the authors and others), the present study uses elegant methodology and transgenic mouse lines to advance our mechanistic understanding of the main cell types and mediators of these effects. The demonstration that astrocytes and endothelial cells are the main effectors in this detrimental role of IL1a by its receptor

signalling is important and brings us a step forward in understanding pathological neuroinflammation after SCI, as well as identifying IL1a as a therapeutic target to prevent secondary damage after SCI. The data are convincing, and the studies are rigorous. I have three major points that should be taken into consideration by the authors, and several other minor points to consider.

1) Provide stronger evidence for disregarding a direct neutrophil involvement in OL cell death by IL1a. The importance of neutrophils in the development of the inflammatory response after SCI has been well documented (with early neutrophil recruitment, peaking at 18-24h after SCI, representing a key factor in the post-injury inflammatory response). Astrocytes are known to be involved in neutrophil recruitment after SCI, by the expression of CXCL1 and CXCL2 (demonstrated by the lead author and others; e.g. Pineau et al. 2010). These infiltrated neutrophils have also been previously associated with OL loss (Donnelly and Popovich, 2008; Satzer et al., 2015). Furthermore, in this article, the authors showed that in most cases when IL1a produced OL death, the recruitment of neutrophils was linked. Finally, neutrophils are one of the main sources of ROS during the early stages of the inflammatory response, and the authors identify ROS as one of main factors affecting viability of OL. For these reasons, neutrophils seem an obvious candidate that directly or indirectly could be involved in OL cell death. However, the authors claim that neutrophils are not involved directly in OL loss, supported by 2 different findings shown in supplementary figures 6 and 7. In my opinion, it is hard to disregard the link between neutrophils, ROS production and OL cell death based on the data in these supplementary figures.

In supplementary figure 6, the authors show that depletion of Ly6C/G+ (Gr-1) neutrophils and monocytes does not alter IL-1 α -mediated OL loss. There are some technical issues relating to non-specificity, since anti Gr1 not only affects neutrophil recruitment, but also depletes other cell types including monocytes. The use of alternative methods, such as 1A8 antibodies (anti-ly6G, which is more specific) requires more doses and could promote other side effects. The authors are clearly aware of the various issues and validated their results with LysM-GFP mice, which is important and appreciated. However, leaving potential technical issues aside, I still am not 100% convinced about the affirmation that neutrophils are not involved in OL cell death by IL1a. The authors state that anti-Gr1 ab depleted neutrophil recruitment and the number of neutrophils were reduced in 70% compared with saline or isotype control after IL1a injection (supplementary 6h). However, the reduction reached no significance when compared with these groups and the depletion of neutrophils in the spinal cord was not complete. This remaining neutrophil infiltration could explain why Olig2+ cell counts were not re-established in comparison with saline treatment (vs IL1a injected mice). More consistent results should be provided to

discard neutrophils as a mediator of OL cell death by IL1a. At the least, the authors could perform an *in vitro* analysis of direct effects of IL1a on neutrophils and its effects on OL death. For example, evaluate the viability of OLs in culture, after application of supernatant from neutrophil cell cultures (wt and IL1r1ko) pre-activated with IL1a. Additionally, the authors could do more to explore the potential influence of other cell types on neutrophil activation and subsequent OL death. For example, whether IL1a activation of astrocytes could lead to neutrophil activation, and then neutrophils act as OL killers by an indirect effect of IL1a. As neutrophils are ROS producers, the authors could also evaluate the expression of ROS in neutrophils after IL1a activation. In supplementary figure 7 the authors discard a direct role of neutrophils in OL cell death by IL1a, by showing that an antagonist of ROS reduced OL death despite not reducing neutrophil recruitment after SCI. An alternative interpretation is that the lack of neutrophil effect on OL cell death is due to ROS inhibition, which may mask the potential effects of neutrophils on OL viability (since neutrophils also produce ROS, so by inhibiting ROS you are likely inhibiting a detrimental neutrophil effect on OL survival). As outlined above, additional studies to evaluate direct or indirect IL1a activation of neutrophils (which could produce ROS and contribute to OL death) will be important.

We thank the reviewer for these positive comments, and for recognizing that our data are convincing and high quality, and acknowledging our aim to advance the mechanistic understanding of the main cell types and mediators of neuroinflammation and secondary degeneration after SCI.

A major critique of the reviewer is that we did not provide enough evidence for disregarding a direct involvement of neutrophils in OL loss induced by IL-1 α . As correctly pointed out by the reviewer, many papers have suggested that neutrophils play a key role in the development of the inflammatory response after SCI and their presence is associated with OL loss. We also agree with the reviewer that our neutrophil depletion strategy based on repeated injections of anti-Gr-1 antibodies was not specific, since other cell types including monocytes may have been depleted, and as confirmed in the original Suppl. Fig. 6B-C. The original approach also proved ineffective in fully depleting the neutrophil population, with a depletion rate of approximately 70%. To fully address previous issues of specificity and efficacy, we carried out a series of *in vivo* experiments to test the optimized neutrophil-specific depletion strategy recently developed by Boivin and colleagues³³. This method, referred to as the combination (Combo) depletion strategy, is a double antibody-based depletion strategy that enhances neutrophil elimination in the long term via anti-Ly6G, both in blood and in tissues of interest. As shown in the **new Fig. 9**, both the original Combo depletion strategy and an adapted version of it were tested in parallel in *Ly6g*^{Cre-TdT::R26-TdT} reporter mice (also known as Catchup^{IVM-red} mice) injected with either

PBS or rmIL-1 α i.c.m. to first validate neutrophil depletion efficacy and specificity. *Ly6g^{Cre-TdT}::R26-TdT* reporter mice were used because Ly6G immunofluorescence staining cannot be performed following treatment with the anti-Ly6G antibody. In brief, we found that our adapted version of the Combo protocol, referred to as the “Combo⁺” depletion strategy in the revised manuscript, led to an eradication of >95% of Ly6G-TdT⁺ blood neutrophils compared to control groups at 24 hours post-injection of IL-1 α (**new Fig. 9B**). Likewise, the total number of neutrophils was reduced by ~90% and ~85% in the spinal cord of IL-1 α -injected *Ly6g^{Cre-TdT}::R26-TdT* mice treated with the Combo⁺ strategy compared to those treated with either saline or the isotype control antibody (**new Fig. 9C**). Thus, the Combo⁺ treatment allowed us to decrease the total number of spinal cord-infiltrating neutrophils by ~90% compared to ~55% in the original Combo treatment and ~70% in the anti-Gr-1 antibody alone. Despite the virtually complete elimination of infiltrating neutrophils in the spinal cord of IL-1 α -injected mice, the Combo⁺ treatment did not prevent OL loss (**new Fig. 9H**). Furthermore, we discovered that 80% of the few remaining neutrophils were localized in the spinal cord gray matter, leaving on average only about 3.1 ± 1.1 neutrophils/mm² in the spinal cord white matter where OL loss was quantified (**new Fig. 9H**). Altogether, these results strongly suggest that neutrophils do not play a role in the death of mature spinal cord OLs observed after IL-1 α injection in the CNS. Please note that substantial changes were made to the *Results* (page 17), *Discussion* (pages 26-27), *Methods* (page 30), and *Figure Legends* (page 52) sections to incorporate and discuss these new data.

2) In relation to major point 1, the data in figure 9 (effects of restoration/deletion of IL-1R1 in astrocytes) reveals clearly that IL1 α induces OL cell death in part through activation of IL-1R1 signalling in astrocytes. However, poor restoration effects could be the reason why it does not restore the infiltration of neutrophils. The authors could evaluate CXCL1, CXCL2 and CCL2 levels in each condition to determine whether the potential lack of these chemokines by the poor receptor restoration is responsible for the lack of neutrophil recruitment restoration. This would provide more in-depth mechanistic information regarding the astrocyte-neutrophil-OL death axis.

The issue of poor recombination efficiency in select mouse lines was also raised by Reviewers 1 and 2. After careful consideration of possible causes, it was determined that the low efficiency was the result of poor experimental design choice, seeing as recombination was initially estimated using genomic DNA extracted from neonatal glial cells, rather than from astrocytes and OLs isolated from the adult CNS of our mouse lines. To fully address this matter, we have since developed the expertise to isolate astrocytes and OLs from the adult mouse spinal cord and brain using the semi-automated gentleMACS™ Octo

Dissociator and standardized protocols developed by Miltenyi Biotec. As shown in the **new Fig. 8**, we are happy to report that *Il1r1* gene expression was restored to approximately 60% of normal levels in astrocytes of *Gfap^{Cre}::Il1r1^{+/+}* mice at 8-10 weeks of age, while it was reduced by nearly 80% of normal levels in astrocytes of adult *Gfap^{Cre}::Il1r1^{fl/fl}* mice. The conditional knockout and restored models used herein are therefore far superior to those previously used. For more detailed information, please refer to our responses to the “General Impressions” Comment from Reviewer 1, as well as our responses to comments related to Fig. 6 raised by both Reviewers 1 and 2 (found above).

3) Provide more detailed information on spatial and anatomical findings relating to OL death and neutrophil infiltration. For most figures high power images of immunohistochemical staining are shown, but very little spatial or anatomical information is provided. In figure 2, for example, what region of the spinal cord is this? Was this effect seen across multiple segments at all levels? Was Fos activated in most white matter tracts? Can some low power images, and some spatial quantitation be added to more clearly see the location/distribution/extent of Fos reactivity across a tissue section? Similar comment for fig 3 which shows high power images of Ly6G+ and Olig2+CCI+. In which regions/tracts/projections were these effects observed? This also applies to subsequent figures showing immunohistochemical findings. Are there any specific white matter tracts that are particularly vulnerable to IL1a-mediated OL death? Is it more pronounced in dorsal, lateral, ventral tracts? This seems important, particularly given the recent work by Floriddia et al showing that distinct oligodendrocyte populations have spatial preference and different responses to SCI (Nat Commun 11, 5860, 2020, DOI: 10.1038/s41467-020-19453-x).

The reviewer brings up an interesting set of questions. The revised manuscript now includes results from additional immunohistochemical analyses showing the anatomical (cervical, thoracic, and lumbar spinal cord levels) and spatial (dorsal, lateral, and ventral white matter tracts) distribution of neutrophil and OL presence in the mouse spinal cord following administration of either PBS or IL-1 α into the CNS (see **new Suppl. Fig. 4**). Overall, these additional data show that IL-1 α injection into the cisterna magna mediates neutrophil infiltration and OL loss throughout the entire rostro-caudal axis, independent of the white matter tract undergoing analysis. We point out, however, that the effect of IL-1 α on OL cell death attenuates slightly the greater the distance from the injection site. The text has been modified to reflect these observations, and the work of Floriddia et al. is cited to justify the rationale behind the recommendation/analysis. We refer to the revised *Results* section on page 9 (lines 202-208): “Given the recent discovery by Floriddia and colleagues that distinct OL populations have spatial preferences in the spinal cord and exhibit different responses to injury³⁴, we next investigated whether mature OLs

vulnerable to IL-1 α were localized at a specific spinal level or white matter tract. IL-1 α -mediated OL loss was observed throughout the entire rostro-caudal axis, independent of the white matter tract analyzed (Suppl. Fig. 4D-G). However, oligodendrocyte death slightly diminished as the distance from the injection site increased.”

As requested, low power images are now presented as part of the **new Suppl. Fig. 3** to more clearly see the distribution and extent of Fos reactivity across a spinal cord cross section.

4) In figure 2 the authors show that after IL1a i.c.m. injection, activation of oligodendrocytes and astrocytes peaked at 4 hr (using Fos expression as a marker of cell activation). This is a robust and clear finding. However, I miss the same analysis in microglial cells. Have the authors checked this in microglia? It is appreciated that evaluating c-fos expression and its upregulation after microglial activation could be problematic (in comparison to other neural cell types), but could authors use other marker(s) to analyse microglia activation at different time points after IL1a injection?

As requested, data on microglia activation are now provided in the **new Fig. 2** and **new Suppl. Fig. 3**. This new analysis shows that microglia are not activated by IL-1 α upon central delivery.

5) In figure 5, the authors further corroborate the role of IL1a in neutrophil recruitment and OL cell death through IL-1R1 signalling, since the use of IL1a inhibitor Anakinra abolished those effects. This data is clear. My concern is more related with the total Olig2 counting (cs/mm²) in animals treated with anakinra. Authors showed that in animals treated with anakinra Olig2 cell count is around 300 cells per mm², which is significantly less than the other studies in the research paper (which showed at least 500 cells per mm²). Could this be a potential detrimental side effect of anakinra? Can the authors explain this reduction?

The reviewer is correct that OL numbers appear unusually low in this particular experiment. Possible explanations may include variations in animal perfusions, tissue processing, immunostaining, analysis of scanned tissue slides *versus* fluorescence quantification under the microscope, quantification by different observers, and so on. Throughout our study we ensured appropriate control groups in each experiment, even if it meant replicating the same control groups from one experiment to another, all in the aim of accounting for these variations.

6) Supplementary figure 3 – add some labels to the figure to indicate what the bands are and what the lanes represent. The legend is also confusing, are lanes 1 and 6 both primary brain microvascular endothelial cells; and lanes 3 and 5 both primary astrocytes?

We apologize for the confusion that we have created. Supplementary figure 3, which is now part of the **new Fig. 4**, has been modified according to the reviewer's requests. We point out that some samples were run in duplicate to ensure that side-by-side comparisons were possible even when separate gels were used.

7) Some figures seem quite sparse and could be combined since the data are related, unless some additional data is provided (see major comments 1-3).

We agree. To address this issue, several changes were implemented to make the figures more impactful: Figure 1 was modified to include additional time points up to 35 days post-SCI, including when microglia were depleted; Figures 2 and 3 were merged together and renamed as Fig. 2; Figure 5 was merged with Suppl. Figs. 2 & 3, which has now been renamed as Fig. 4; Supplementary Fig. 6 has become a main figure, renamed as Fig. 9, and was reinforced with additional immune cell depletion experiments using a more specific approach to deplete neutrophils, as discussed above (see Major Comment #1); Figure 10 and Suppl. Fig. 7 were merged and renamed as Fig. 10; Four new supplementary figures (Suppl. Figs. 2-4 and 6) were added to the paper to give further details on results showing the effects of either central IL-1 α delivery and/or SCI on plasma cytokine and chemokine levels, glial/neuronal cell activation, neutrophil infiltration and OL cell death as a function of sex or spinal cord anatomy, and blood-spinal cord barrier (BSCB) leakage.

8) Additional minor points to consider in the text. Some statements seem to over-claim the findings and should be toned down e.g. line 440-441 “dramatic loss of mature OLs along the rostrocaudal axis of the spinal cord”. Unless some spatial characterisation is provided (see major point 2, above), this statement is not supported.

Similarly, Line 124, 145-146 and 924 “IL-1 α is released by injured microglia during the early acute phase of SCI”, “...these results show that damaged microglia release the alarmin IL-1 α during the early acute phase of SCI” and “Damaged microglia rapidly release IL-1 α at the site of spinal cord contusion in mice.” Immunohistochemical localisation does not demonstrate “release”.

The work of Didangelos on SCI and alarmins is relevant and should be cited.

Line 435-437 Please clarify the statement “We found that IL-1 α is released by microglia located in the lesion core, most of which are dead as early as 24 hours post-SCI.” Is something missing from this sentence, are they referring to OL death here?

As requested by Reviewer 3, we now provide a new supplementary figure (**new Suppl. Fig. 4**) that acts as a spatial/anatomical characterization of the widespread effect of IL-1 α throughout the entire rostro-caudal axis of the spinal cord. Based on these new data, our statement regarding “dramatic loss of mature OLs along the rostrocaudal axis of the spinal cord” is now supported and justified.

The reviewer is right, and the word “release” has been exchanged for “produced” in these three sentences, as well as elsewhere in the text.

The work of Didangelos et al. is now cited as a key supporting reference in the revised *Introduction* section (page 3, lines 69-71): “Recent work in proteomics has predicted that the transition from primary to secondary damage after SCI and the accompanying neuroinflammatory responses are triggered by the release of danger-associated molecular patterns (DAMPs, also referred to as alarmins) from disrupted cells ⁷.” We apologize for this omission.

The sentence has been rewritten as follows to clarify its meaning (page 20, lines 452-455): “We found that IL-1 α is primarily derived from damaged microglia located in the lesion core, and that peak expression of this alarmin correlates with the death of microglia, occurring within the first 24 hours of SCI.”

Once again, we appreciate the comments of these reviewers and hope our responses and revisions address their points with satisfaction.

REFERENCES

1. Guttenplan KA, *et al.* Neurotoxic reactive astrocytes induce cell death via saturated lipids. *Nature* **599**, 102-107 (2021).
2. Tansley S, *et al.* Single-cell RNA sequencing reveals time- and sex-specific responses of mouse spinal cord microglia to peripheral nerve injury and links ApoE to chronic pain. *Nat Commun* **13**, 843 (2022).
3. Bastien D, *et al.* IL-1alpha gene deletion protects oligodendrocytes after spinal cord injury through upregulation of the survival factor Tox3. *J Neurosci* **35**, 10715-10730 (2015).
4. Luheshi NM, Kovacs KJ, Lopez-Castejon G, Brough D, Denes A. Interleukin-1alpha expression precedes IL-1beta after ischemic brain injury and is localised to areas of focal neuronal loss and penumbral tissues. *J Neuroinflammation* **8**, 186 (2011).
5. Liddelow SA, *et al.* Neurotoxic reactive astrocytes are induced by activated microglia. *Nature* **541**, 481-487 (2017).
6. Guttenplan KA, *et al.* Neurotoxic Reactive Astrocytes Drive Neuronal Death after Retinal Injury. *Cell Rep* **31**, 107776 (2020).
7. Didangelos A, Puglia M, Iberl M, Sanchez-Bellot C, Roschitzki B, Bradbury EJ. High-throughput proteomics reveal alarmins as amplifiers of tissue pathology and inflammation after spinal cord injury. *Sci Rep* **6**, 21607 (2016).
8. Nimmerjahn A, Kirchhoff F, Helmchen F. Resting microglial cells are highly dynamic surveillants of brain parenchyma in vivo. *Science* **308**, 1314-1318 (2005).
9. Davalos D, *et al.* ATP mediates rapid microglial response to local brain injury in vivo. *Nat Neurosci* **8**, 752-758 (2005).
10. de Rivero Vaccari JP, *et al.* P2X4 receptors influence inflammasome activation following spinal cord injury. *J Neurosci* **32**, 3058-3066 (2012).
11. Gadani SP, Walsh JT, Smirnov I, Zheng J, Kipnis J. The glia-derived alarmin IL-33 orchestrates the immune response and promotes recovery following CNS injury. *Neuron* **85**, 703-709 (2015).
12. Kigerl KA, Lai W, Wallace LM, Yang H, Popovich PG. High mobility group box-1 (HMGB1) is increased in injured mouse spinal cord and can elicit neurotoxic inflammation. *Brain Behav Immun* **72**, 22-33 (2018).

13. Dmytriyeva O, *et al.* The metastasis-promoting S100A4 protein confers neuroprotection in brain injury. *Nat Commun* **3**, 1197 (2012).
14. Kigerl KA, Lai W, Rivest S, Hart RP, Satoska AR, Popovich PG. Toll-like receptor (TLR)-2 and TLR-4 regulate inflammation, gliosis, and myelin sparing after spinal cord injury. *J Neurochem* **102**, 37-50 (2007).
15. Church JS, Kigerl KA, Lerch JK, Popovich PG, McTigue DM. TLR4 Deficiency Impairs Oligodendrocyte Formation in the Injured Spinal Cord. *J Neurosci* **36**, 6352-6364 (2016).
16. Impellizzeri D, *et al.* Role of Toll like receptor 4 signaling pathway in the secondary damage induced by experimental spinal cord injury. *Immunobiology* **220**, 1039-1049 (2015).
17. Wang X, *et al.* P2X7 receptor inhibition improves recovery after spinal cord injury. *Nat Med* **10**, 821-827 (2004).
18. Roth TL, Nayak D, Atanasijevic T, Koretsky AP, Latour LL, McGavern DB. Transcranial amelioration of inflammation and cell death after brain injury. *Nature* **505**, 223-228 (2014).
19. Lytle JM, Wrathall JR. Glial cell loss, proliferation and replacement in the contused murine spinal cord. *Eur J Neurosci* **25**, 1711-1724 (2007).
20. Tripathi R, McTigue DM. Prominent oligodendrocyte genesis along the border of spinal contusion lesions. *Glia* **55**, 698-711 (2007).
21. Hesp ZC, *et al.* Proliferating NG2-Cell-Dependent Angiogenesis and Scar Formation Alter Axon Growth and Functional Recovery After Spinal Cord Injury in Mice. *J Neurosci* **38**, 1366-1382 (2018).
22. McTigue DM, Tripathi RB. The life, death, and replacement of oligodendrocytes in the adult CNS. *J Neurochem* **107**, 1-19 (2008).
23. Hesp ZC, Goldstein EZ, Miranda CJ, Kaspar BK, McTigue DM. Chronic oligodendrogenesis and remyelination after spinal cord injury in mice and rats. *J Neurosci* **35**, 1274-1290 (2015).
24. Duncan GJ, *et al.* Locomotor recovery following contusive spinal cord injury does not require oligodendrocyte remyelination. *Nat Commun* **9**, 3066 (2018).
25. Qian J, *et al.* Interleukin-1R3 mediates interleukin-1-induced potassium current increase through fast activation of Akt kinase. *Proc Natl Acad Sci U S A* **109**, 12189-12194 (2012).

26. Tapia VS, *et al.* The three cytokines IL-1beta, IL-18, and IL-1alpha share related but distinct secretory routes. *J Biol Chem* **294**, 8325-8335 (2019).
27. Luheshi NM, Rothwell NJ, Brough D. The dynamics and mechanisms of interleukin-1alpha and beta nuclear import. *Traffic* **10**, 16-25 (2009).
28. Martin-Sanchez F, *et al.* Inflammasome-dependent IL-1beta release depends upon membrane permeabilisation. *Cell Death Differ* **23**, 1219-1231 (2016).
29. Bageghni SA, *et al.* Fibroblast-specific deletion of interleukin-1 receptor-1 reduces adverse cardiac remodeling following myocardial infarction. *JCI Insight* **5**, (2019).
30. Collier JJ, *et al.* Pancreatic, but not myeloid-cell, expression of interleukin-1alpha is required for maintenance of insulin secretion and whole body glucose homeostasis. *Mol Metab* **44**, 101140 (2021).
31. Sun W, *et al.* SOX9 Is an Astrocyte-Specific Nuclear Marker in the Adult Brain Outside the Neurogenic Regions. *J Neurosci* **37**, 4493-4507 (2017).
32. Ung K, *et al.* Olfactory bulb astrocytes mediate sensory circuit processing through Sox9 in the mouse brain. *Nat Commun* **12**, 5230 (2021).
33. Boivin G, *et al.* Durable and controlled depletion of neutrophils in mice. *Nat Commun* **11**, 2762 (2020).
34. Floriddia EM, *et al.* Distinct oligodendrocyte populations have spatial preference and different responses to spinal cord injury. *Nat Commun* **11**, 5860 (2020).

REVIEWERS' COMMENTS

Reviewer #2 (Remarks to the Author):

The authors performed a comprehensive revision of their original manuscript and have addressed my concerns. I would recommend the manuscript for publication.

One minor point, is that the term 'activation' is not defined and used in association with c-fos expression. What is the phenotype of c-fos expressing astrocytes? Are these pro-inflammatory, expressing ROS?

Reviewer #3 (Remarks to the Author):

The authors have done an excellent job in addressing all the comments and concerns. This high-quality and elegant study is a nice addition to the field and I have no further comments.

Reviewer #4 (Remarks to the Author):

The revised manuscript by Bretheau and colleagues describes in great detail how the alarmin interleukin-1a (IL-1a) triggers early-acute (neuro)inflammation and degenerative changes in the CNS.

The authors reconfirm a deleterious role for IL-1a in recovery from traumatic SCI and demonstrate that microglia are the main source of IL-1a early after SCI (<24 hours post-injury). To more precisely delineate the role of this particular alarmin, that is, separate from other signalling mediators that are also recognised as and/or part of the so-called danger- and/or damage-associated molecular patterns (DAMPs), they have established a model in which delivery of IL-1a into the CSF (intra-cisterna magna; icm) causes astrocyte activation, oligodendrocyte death and neutrophil recruitment. These findings mirror some aspects of the early inflammatory response to SCI (neutrophil recruitment) and the degenerative changes (oligodendrocyte death) that coincide with these.

The authors next perform a series of elegant experiments in which they use genetic approaches to either ablate or restore the main receptor for IL-1a in select cell types, further complemented by pharmacologic and in vitro studies. From these, they demonstrate / conclude that the deleterious effects of IL-1a signalling are mostly mediated via endothelial cells and reactive astrocytes. They also convincingly demonstrate through depletion experiments (anti-Ly6G treatment) that the recruitment of neutrophils to the CNS following IL-1a infusion is an 'epiphenomenon', that is, something that results from activation of the vascular endothelium but that this is not associated with and/or driving the death of oligodendrocytes.

Overall, there is a very high level of scientific rigour in most aspects of the manuscript and there are several important and/or new insights as to the role of IL-1a signalling in acute SCI (and neuroinflammation more broadly), and how this in turn influences secondary pathology and outcomes. A significant amount of work has been performed to address previously raised concerns, greatly improving the manuscript overall. Some questions and/or points of clarification do, however, remain as highlighted below in response to the comments previously raised by Reviewer 1 and the authors' rebuttal of these.

**** Reviewer 1, Point 1: General Impressions.**

"There is a lot to unpack...and conceptual framework is hard to follow sometimes...findings not surprising or novel....do add incrementally to the literature and does add to current knowledge regarding secondary injury after SCI....What the manuscript does do well is pull together data on IL-1a in SCI that has been spread across dozens of previously published papers and present it all together in a single package".

The authors have been very receptive to this particular comment, reorganising the manuscript and also several of the figures to aid the storyline whilst at the same time (better) highlighting existing knowledge gaps, and also how these are addressed in the current paper.

**** Reviewer 1, Point 2: Experiments rely on direct injection of IL-1a as being representative of the SCI environment, however when repeated in an SCI model many of the observations did not replicate...Mechanisms described may still be applicable, their therapeutic efficacy in the complex secondary injury cascade may be limited. Not addressed or acknowledged in the discussion."**

The revised manuscript is more explicit in acknowledging some of the limitations highlighted above. New data is also included to show that the systemic response is also remarkably similar between SCI and icm delivery of IL-1a based on plasmatic cytokine/chemokine levels (Fig. S2), which goes some

way to alleviating these concerns. The authors further correctly point out that early histopathological hallmarks are also similar between conditions (astrocyte activation, blood-spinal cord barrier leak and early inflammation), although it is also reasonable to assume that many of these changes are transient with IL-1a injection (as in fact shown) and much more severe and/or not resolving with SCI.

Overall, I feel that some of the limitations and/or outstanding questions around the findings could be better highlighted and/or emphasized.

- In response to the new/revised statement on page 596-597 (wrongly referred to as page 25, lines 538-584 in the letter) that *"it must be noted, however, that the secondary injury cascade occurring after traumatic SCI is complex and likely involves multiple pathways"*: This is true but it would be strengthened considerably if the authors would extend and/or (re)acknowledge here more strongly that their findings likely only (or mostly) apply to the first 24 hours post-SCI based on the observed temporal pattern of IL-1a expression / release. As acknowledged in their Introduction (line 109-110), oligodendrocyte death occurs for days (if not weeks) post-SCI, and mechanisms other than IL1a-IL1R1 signalling must therefore drive cell loss during the post-acute phase. The seemingly narrow temporal window has obvious implications for therapeutic approaches and clinical translation.

- Further, and long the same lines as the above, regarding *"Likewise, it will be important to identify other downstream effector molecules like ROS that directly mediate CNS cell toxicity.....(lines 612-617)"*. This is a fair point but it would also only apply if one was to target downstream consequences of IL-1R1 activation (as done here for NAC - Fig. 10). It does not address the problem that the temporal profile of IL-1a does not match established time courses of oligodendrocyte death during the post-acute phase of SCI, i.e. where IL-1a itself appears absent. All that we can conclude therefore is that IL-1a is a key mediator in initiating neuroinflammation very early after SCI and that this is associated with glial activation and oligodendrocyte loss.

This line of reasoning also logically explains as to why not all findings replicate and/or are not as robust in the SCI model compared to where IL-1a is delivered directly, as other cell types and pathways likely add and/or must drive the progressive secondary cell death during the subacute and intermediate-chronic phase of SCI. It is important in that regard to provide clarity as to what the treatment regimen was for the experiments where NAC was administered to SCI mice – this appears to be missing from the revised manuscript and is important to include as there are many other cellular sources and/or pathways other than IL-1a-IL-1R1 signalling that would contribute to ROS production in this pathology.

- Lastly, and perhaps a more minor point, the manuscript does not really consider whether the mode of oligodendrocyte cell death is the same for SCI and IL-1a infusion, and also whether the various other well-established alarmins / DAMPs mentioned in the introduction are released following IL-1a infusion and associated cell death. If occurring, then this does not necessarily take away from the findings of IL-1a having an initiating role, but more so would put a nuance on its potency. This should be acknowledged.

**** Reviewer 1, Point 3: The effect of several of the KI/KO models is limited (i.e. less than 30% increase in gene expression or less than 20% decrease). Could this not influence the extent to which these models show a change in response compared to control? Authors claim the lack of effect indicates the cell type is not directly involved in the effects but I am not convinced the data is strong enough to support that....**

Overall, it is fair for the authors to comment that some of the limitations around the use of conditional and inducible gene targeting strategies apply to many high-impact studies, yet they remain 'state-of-the-art' as no better alternatives are available. The authors do a good job in my view in acknowledging these limitations, and the combination of approaches where *Il1r1* is either selectively ablated or restored certainly is a strength of the paper.

Additional analyses have been performed to address this point, focusing specifically in recombination efficiency in relevant cell types isolated from the CNS of adult mice (as opposed to neonatal pups). The newly added data suggest that, at least for astrocytes, much higher levels (60-80%) of IL-1R1 ablation or reconstitution were achieved. This adds significantly to the strength of the observations around this particular cell type but does not necessarily address possible weaknesses and/or overreach regarding some other lines used (e.g. *Pdgfra-CreER*, see comments on author response to queries re: original Fig. 6 below).

**** Reviewer 1, Point 4: ...Male and female rodents have been shown to have very different neuroimmune responses. It is not clear whether sexes were analysed separately to take into account possible sex differences.**

The revised manuscript convincingly shows that there is no sex difference in the key readouts (neutrophil infiltration and oligodendrocyte death) following IL-1a infusion, and hence that pooling is appropriate.

**** Reviewer 1, Point 5: Lab has previous work demonstrating IL-1a release by microglia. Current goal is to determine the extent to which this release drives secondary degeneration via astrocytes and endothelial cells....did not provide enough overview of neuroinflammation and the role of cytokines /IL-1a to convince me the question is novel...similar papers in existence.**

Previous work from this group in SCI indeed showed that IL-1a is released by myeloid cells, with the current paper narrowing this down to microglia being the source. Considering the temporal profile of IL-1a expression / release and the fact that robust recruitment of monocytes and macrophages more so takes place beyond the 24 hour timepoints, this finding is logical and not surprising but it remained to be unequivocally confirmed.

Otherwise, whilst I do agree with Reviewer 1 that there is a significant body of literature on IL-1a in various forms of acquired CNS injury, there are many contextual nuances and/or differences between these pathologies and a careful dissection of how the alarmin IL-1a influences the outcome from traumatic SCI remained outstanding. It is this sort of careful dissection and mechanistic understanding as to how, when and where inflammatory mediators exert their effects that will ultimately be needed to successfully design and translate new targeted therapies into the clinic.

Overall, it could be argued that the authors have addressed this satisfactorily concern by rewriting aspects of their introduction, providing a contextualised overview of current insights, why alarmins are important targets for further investigation, and then also the knowledge gap, i.e. that mechanistic insights as to how specific alarmins drive secondary damage are incomplete, as highlighted in their response and the revised text.

**** Reviewer 1, Point 6: Statement...."transition from primary to secondary damage is triggered, at least in part, by the release of the intracellular content of necrotic cells....". Could give a clearer picture of the importance of IL-1a aside from basic generalisation.**

This point extends from the previous one and the authors have addressed this as part of their revision (see again author response and highlighted text in the introduction, in particular lines 70-73 and 79-83).

**** Reviewer 1, Point 7: ...Figure 1...Nothing has been done here to show that macrophages aren't also releasing it.**

This point has been adequately addressed, with the revised Figure 1 pointing out damaged and/or activated microglia. There is also data showing that IL-1a protein is present in tdTomato-positive microglia and not GFP-positive cells, which mostly represents immune infiltrate in LysM-eGFP mice (this is well established).

**** Reviewer 1, Point 8: ...Figure 2...Olig2 alone is insufficient to resolutely indicate activation of OLs. Olig2 is not specific to OLs.**

This has been adequately addressed by the inclusion of CC-1 staining, which is a well-established and accepted marker of OLs.

**** Reviewer 1, Point 9: ...Data not shown: OL loss increased as IL-1a concentration increased.... Figures 2 and 3 would be more impactful if combined.**

The comment regarding the relationship between IL-1a concentration and OL loss was not addressed. This may be of significance as it could explain as to why olig2+cc1+ cell counts following IL-1a infusion appear quite different between experiments (compare e.g. new Fig. 2H with new Fig. 3C).

Regardless, original Figures 2 and 3 have been merged as requested.

**** Reviewer 1, Point 10: BrdU experiment to examine OPC proliferation following IL-1a injection....an example where cytokine injection does not necessarily equal SCI...Olig2 alone is not specific for OLs....limitations/translatibility could be discussed further.**

The authors have provided a balanced argument / overview of relevant literature in their rebuttal that supports a similarly rapid OPC response to SCI, along with the subsequent maturation of these into mature oligodendrocytes. The Discussion has been amended accordingly (lines 535-547). Coming back to my earlier comment (in relation to the response to Reviewer 1, Point 2 above), this would be another opportunity to mention that the role of IL-1a in inducing the death of mature OLs is likely restricted to the first 24 hours of injury only.

Otherwise, as per point 8, the inclusion of CC-1 as a marker for mature OLs addressed the comment regarding staining specificity.

**** Reviewer 1, Point 11: Suppl. Fig 2....IL-1a and IL-1R1 KO mice recover significantly better from SCI....not sure why it's supplemental.**

This data has now been included in the main manuscript and is part of the new Figure 4.

One thing that stood out but wasn't further discussed is the fact that IL-1a KO mice recovery substantially better than IL1r1 KO mice here, both in the main BMS score and also the BMS sub-scores. Should they not at least be similar? Differences in SCI recovery also appear present between the respective WT cohorts, most obviously so in the BMS sub-scoring. Were all strains fully backcrossed onto a C57BL/6J background? I appreciate that each experiment is individually controlled but it does bring into question the variability of the lesion model. It would be good for the authors to comment on this and also to include relevant injury parameters (actual applied force and associated displacement between experiments / groups) in the Methods section.

**** Reviewer 1, Point 12: Suppl. Figure 3....immunoblot analysis of IL-1R1...could this have picked up IL-1R3 instead or in addition to IL1R1?**

The rebuttal addressed the confusion that may have arisen from using the term IL-1R3 (introduced into the literature for a truncated splice variant of IL1-R1 by a previous article from an independent group). The revised manuscript now refers to tIL-1R1 instead to avoid this confusion and/or concern about specificity.

Otherwise, the reference in the Discussion that "Our immunoblotting findings confirm that IL-1R1 or a truncated splice variant of IL-1R1 is expressed by microglia, astrocytes and OLS ..." could perhaps be sharpened and/or be more specific, as it seems an intriguing coincidence that only those cells that express the truncated form of IL-1R1 appear to express fos in response to IL-1a stimulation.

**** Reviewer 1, Point 13: Figure 6....restoring Il1r1 gene expression in OPCs...claim the lack of response means that IL-1a is not directly responsible for OL death (as now shown in Fig. 2h of the revised manuscript).**

Possible issues around low-ish recombination efficiency have been partially addressed through additional experiments, that is, for astrocytes but not OLS (impact of the pandemic on authors ability to do the latter is noted).

In absence of additional verification as to what the restoration efficiency was for OLS, the authors rebut by stating that "treatment of primary OLS was not toxic in cell culture experiments (see modified Fig. 10E)". This I am not convinced of, however, as there is a clear increase (more than a doubling) in LDH cytotoxicity visible in Fig. 10E where rm-IL1a is added to the DMEM (i.e. medium not conditioned by astrocytes). This does not take away from the fact that a much greater degree of cell death is induced when OLS are exposed to conditioned medium from astrocytes stimulated with rmIL-1a, but it would probably be significant if a direct comparison was done between both DMEM conditions (+/- IL1a). The authors would do better therefore to err on the side of caution, remove any ambiguity and simply state in the relevant section(s) of the Results and Discussion that a low level of OL death, directly resulting from IL-1a exposure, cannot be excluded at present because of the above limitations / observations.

**** Reviewer 1, Point 14: Figure 7 (now Fig. 6 in revised manuscript)...goal is to determine whether IL-1R1 has auto regulatory effect on microglia....don't think the possibility can be completely ruled out. For both Figure 6 and 7 (Fig. 5 and 6 in revised manuscript). Figure C appears to have a slight downward trend...**

The authors partially rebut this criticism by acknowledging some of the limitations around conditional gene targeting in their Discussion. The point that was originally made here by Reviewer 1 is, however, that partial restoration (not deletion) appears to lead to a downward trend in the number of OLs (compare red bars). This is particularly obvious for what is now Figure 5C, further emphasising the previous point to tone down statements that IL-1a does not directly induce OL death.

Otherwise, the level of recombination, spread of data points within/between groups and statistics otherwise make it reasonable to state that microglial IL-1R1 appears redundant, possibly so because of Il1r2 up regulation, as shown.

**** Reviewer 1, Point 15: Figure 8 (now Fig. 7 in revised manuscript)....conditional restoration of IL1-R1 in endothelial cells....small return of the baseline response? Neutrophil infiltration partially recovered and also partial attenuation of OL death. This suggests endothelial cells are involved in the response to IL-1a. Confirmed in conditional KO of IL-1R1 in endothelial cells, but no change in recovery this time.**

The authors have addressed the first part of these criticisms by repeating experiments in which rm-IL-1a is injected into mice where IL-1R1 expression is selectively restored in endothelial cells. The results confirm earlier findings regarding the effect size of this manipulation regarding neutrophil recruitment and OL death. It is fair to say on the basis of these experiments that downstream consequences of IL-1a signalling in other cell types must contribute to both phenomena. Consistent with this, neutrophil recruitment is not completely abolished when selectively knocking out IL-1R1 in endothelial cells (Fig. 7I), although it should also be noted that the loss of oligodendrocytes is pretty much prevented here (Fig. 7J).

What remains puzzling, and this was also highlighted by Reviewer 1 in their last comment, is why the functional outcomes from SCI are not in agreement with each other. Specifically, restoring IL-1R1 expression in endothelial cells partially 'rescues' the neutrophil response that is normally seen in response to rmIL-1a infusion and also induces cell death, yet, there is no impact in relation to function from which one can conclude that the degree of neutrophil recruitment and/or OL death was not sufficient to attenuate the augmented recovery seen with global IL-1R1 deficiency (red vs. blue lines in Fig. 7e and f), with both strains of mice recovering significantly better compared to the WT controls (black line). On the other hand, selectively knocking out IL-1R1 in endothelial cells only reduces neutrophil recruitment and OL death, which makes sense, yet no change is seen in SCI experiments in relation to the functional outcome. This could be discussed more clearly. I am less bothered by the fact that histopathological observations in the rmIL-1a infusion model do not correlate well here with SCI outcomes, but it does re-emphasize the point made earlier that the effects of IL-1a may be very acute only and that other pathways are more dominant and/or significant in relation to OL death in the post-acute phase.

At a minimum, I would recommend that the authors acknowledge this and provide a nuance to their statement that "...astrocytes and CNS ECs appear to drive OL death via an IL-1R1-dependent release of ROS and other molecules that have yet to be identified" (lines 635-636), adding something along the lines that, based on the evidence at hand, we can reasonably assume that this applies to the first

24 hours post-SCI only and then less so thereafter, in addition to clarifying perhaps in light of this that “other molecules” here implies other pathways, not other molecules downstream of IL-1R1.

A similar nuance can be provided to the cited statement in the rebuttal (line 596-597 of the revised manuscript) that”*secondary injury cascade after trauma SCI is complex and likely involves multiple pathways*”, again emphasizing the point perhaps that the latter are likely to be more dominant and/or driving OL loss in the post-acute phase.

**** Reviewer 1, Point 16: Figure 9 (relating Fig. 8 in the revised manuscript): ...using GFAP promoter to restore IL-1R1 expression in astrocytes...low efficacy of KO and KI make me wonder if part of the lack of an observed effect is for this reason and not related to the mechanism of IL-1a action.**

The revised manuscript contains new data to show that recombination efficiency was much higher (60-80% as opposed to the 20-30% that was stated in the original manuscript where cells from neonatal pups rather than adult mice were used). This indeed considerably reduces any of the initially expressed concern around interpretation / robustness of the findings.

The revised manuscript also contains data to show that only restoration of IL-1R1 in astrocytes appears to drive these cells more towards an A1 phenotype, that is, on the basis of C3 expression. This implies a direct effect of IL-1a on astrocytes as opposed to an indirect one where e.g. IL-1R1-mediated endothelial cell activation and the downstream consequences of this may (partially) drive astrocytic responses.

I would make one comment here though which is in relation to the statement that “*knockout of 80% of Ilr1 expression in astrocytes failed to reduce neutrophil infiltration but partially prevented OL death in response to IL-1a administration (Fig. 8H-J)*” (line 384-385). Overall, I do not disagree with the authors that there is likely a synergistic effect of IL-1a signalling in both astrocytes and ECs that contributes to SCI outcomes. That does not take away, however, that this statement appears to be very much overreaching and is not substantiated by what is actually shown in Fig. 8J. Looking at the spread of data points within and between groups, one can only conclude one thing based on what is presented, which is that the conditional deletion of IL-1R1 from astrocytes has no bearing on the extent of OL death in these experiments, or in the very least that there is no statistical basis / evidence for this. Consistent with that, restoring IL-1R1 expression in astrocytes also does not reinstate the expected level of OL death, with the findings not being significantly different from the WT controls (Fig. 8D, rmlL-1a, grey vs. red bar). The authors should tone down these conclusions as they distract from what is otherwise an excellent manuscript, unnecessarily creating an impression perhaps that there was preconceived bias in what the experimental outcome should be.

RE: Manuscript # NCOMMS-21-21054B

RESPONSES TO THE REVIEWERS' COMMENTS – ROUND 2:

Our responses to the Reviewers' comments are addressed point by point below. To facilitate the work of the Reviewers, all new changes are highlighted in yellow in the revised manuscript. The authors have declared that no conflict of interest exists.

We thank the reviewers for their insightful comments and provide the following responses:

Reviewer 2:

1) The authors performed a comprehensive revision of their original manuscript and have addressed my concerns. I would recommend the manuscript for publication.

We appreciate the kind words of Reviewer 2 and we are thankful for their thorough analysis of our manuscript.

2) One minor point is that the term 'activation' is not defined and used in association with c-fos expression. What is the phenotype of c-fos expressing astrocytes? Are these pro-inflammatory, expressing ROS?

We agree that the term “activated” could be more defined. Nonetheless, in our model, astrocytes express c-Fos during the first 4 hours after the administration of IL-1 α . This expression disappears completely at 24 hours post-injection. Meanwhile, the increase in C3⁺ astrocytes, which has been described as neurotoxic/neuroinflammatory (Liddel et al., *Nature*, 2017), appears 24 hours after IL-1 α injection. Seeing as these two signals do not overlap, it is quite challenging to characterize the phenotype of these c-Fos⁺ astrocytes at 4 hours post-injection.

Our data suggests that the direct activation of IL-1R1 in astrocytes by IL-1 α induces the transition of the astrocyte to its neuroinflammatory phenotype (Fig. 10a), triggering ROS production in cell culture (Fig. 10f). These factors lead us to believe that the activation observed at 4 hours post-injection elicited both proinflammatory and ROS-producing astrocytes. Supporting this view is the long-standing work of Michael Karin and colleagues, who showed that AP-1 transcription factors

including c-Fos are a key link between physiological and pathophysiological production of ROS, notably H₂O₂, and the control of cell proliferation and apoptosis (for review, see Karin and Shaulian, *IUBMB Life*, 2001). We agree with Reviewer 2 that it would be beneficial to test this hypothesis in the future.

To address this minor point, in lines 157-159 of page 7, we now define activation as a “marker of increased transcriptional activity”.

Reviewer 3:

1) The authors have done an excellent job in addressing all the comments and concerns. This high-quality and elegant study is a nice addition to the field and I have no further comments.

The positive feedback received from Reviewer 3 is truly appreciated, and we thank them for the time and care spent on our manuscript.

Reviewer 4:

Overall, there is a very high level of scientific rigour in most aspects of the manuscript and there are several important and/or new insights as to the role of IL-1a signalling in acute SCI (and neuroinflammation more broadly), and how this in turn influences secondary pathology and outcomes. A significant amount of work has been performed to address previously raised concerns, greatly improving the manuscript overall. Some questions and/or points of clarification do, however, remain as highlighted below in response to the comments previously raised by Reviewer 1 and the authors' rebuttal of these.

We thank Reviewer 4 for their constructive comments, as they have helped us improve our manuscript.

1) ** Reviewer 1, Point 1: General Impressions. “There is a lot to unpack...and conceptual framework is hard to follow sometimes...findings not surprising or novel....do add incrementally to the literature and does add to current knowledge regarding secondary injury after SCI....What the manuscript does do well is pull together data on IL-1a in SCI that has been spread across dozens of previously published papers and present it all together in a single package”. **

The authors have been very receptive to this particular comment, reorganising the manuscript and also several of the figures to aid the storyline whilst at the same time (better) highlighting existing knowledge gaps, and also how these are addressed in the current paper.

We are grateful for Reviewer 4's appreciation of our revised work, and acknowledge that their insightful comments, which have been addressed thoroughly below, have enhanced the quality and impact of our work.

2) ** Reviewer 1, Point 2: "Experiments rely on direct injection of IL-1 α as being representative of the SCI environment, however when repeated in an SCI model many of the observations did not replicate...Mechanisms described may still be applicable, their therapeutic efficacy in the complex secondary injury cascade may be limited. Not addressed or acknowledged in the discussion." **

The revised manuscript is more explicit in acknowledging some of the limitations highlighted above. New data is also included to show that the systemic response is also remarkably similar between SCI and icm delivery of IL-1 α based on plasmatic cytokine/chemokine levels (Fig. S2), alleviating these concerns. The authors further correctly point out that early histopathological hallmarks are also similar between conditions (astrocyte activation, blood-spinal cord barrier leakage and early inflammation), although it is also reasonable to assume that many of these changes are transient with IL-1 α injection (as in fact shown) and much more severe and/or not resolving with SCI.

Overall, I feel that some of the limitations and/or outstanding questions around the findings could be better highlighted and/or emphasized.

We agree that the secondary damage events occurring after SCI are much more severe than as witnessed with an acute central administration of IL-1 α . It is also true that the resolution phase of inflammation does not occur after SCI, that i.c.m injection of IL-1 α fails to reproduce the chronicity of inflammation, and that the activation of a single inflammatory pathway is not sufficient to replicate SCI. We agree that this can and should be emphasized in the final manuscript. To do so, we have added the following text to the *Discussion* (page 25, lines 585-588): "It must be noted, however, that the secondary injury cascade occurring after traumatic SCI is complex and includes various proinflammatory pathways. Our findings regarding the effects of IL-1 α likely

apply to the first 24 hours after the injury, when the cytokine is present.” Also, on page 27 (lines 634-642), the following paragraph was added to the *Discussion* to further address the point raised by both Reviewers 3 and 4: “SCI generates an extensive activation of several proinflammatory pathways, many of which do not resolve over the course of the pathology⁹². The administration of IL-1 α i.c.m. allowed us to study the effects of this alarmin in the spinal cord, and to replicate histopathological markers up to 24 hours after the injection, which is when this cytokine is present. Although this model is appropriate for studying the function of IL-1 α , the administration of a single dose of this cytokine is not enough to replicate the chronicity and complexity of inflammation and the physical outcomes present in SCI. Future investigations will be necessary to know the effects of the IL-1 α /IL-1R1 pathway in later stages of SCI, along with its role in injury when other complementary DAMPs and inflammatory pathways are triggered.”

2A) In response to the new/revised statement on page 596-597 (wrongly referred to as page 25, lines 538-584 in the letter) that “it must be noted, however, that the secondary injury cascade occurring after traumatic SCI is complex and likely involves multiple pathways”: This is true but it would be strengthened considerably if the authors would extend and/or (re)acknowledge here more strongly that their findings likely only (or mostly) apply to the first 24 hours post-SCI based on the observed temporal pattern of IL-1 α expression/release. As acknowledged in their Introduction (line 109-110), oligodendrocyte death occurs for days (if not weeks) post-SCI, and mechanisms other than IL1 α -IL1R1 signaling must therefore drive cell loss during the post-acute phase. The seemingly narrow temporal window has obvious implications for therapeutic approaches and clinical translation.

We agree that we should emphasize that IL-1 α is one of the many cytokines released after damage to the spinal cord, and that the effects observed in the manuscript are more relevant in the early stages of the pathology, so we have made the following changes to lines 634-642: “SCI generates an extensive activation of several proinflammatory pathways, many of which do not resolve over the course of the pathology⁹². The administration of IL-1 α i.c.m. allowed us to study the effects of this alarmin in the spinal cord, and to replicate histopathological markers up to 24 hours after the injection, which is when this cytokine is present. However, although this model is appropriate for studying the function of IL-1 α , the administration of a single dose of this cytokine is not enough to replicate the chronicity and complexity of inflammation and the physical outcomes present in

SCI. Future investigations will be necessary to know the effects of the IL-1 α /IL-1R1 pathway in later stages of SCI, along with its role in injury when other complementary DAMPs and inflammatory pathways are triggered.”

2B) Further, and along the same lines as the above, regarding “Likewise, it will be important to identify other downstream effector molecules like ROS that directly mediate CNS cell toxicity.....(lines 612- 617)”. This is a fair point but it would also only apply if one was to target downstream consequences of IL-1R1 activation (as done here for NAC - Fig. 10). It does not address the problem that the temporal profile of IL-1a does not match established time courses of oligodendrocyte death during the post-acute phase of SCI, i.e. where IL-1a itself appears absent. All that we can conclude therefore is that IL-1a is a key mediator in initiating neuroinflammation very early after SCI and that this is associated with glial activation and oligodendrocyte loss.

In conjunction with point 2A, we agree that our manuscript should indicate that our findings apply to the first 24 hours post-SCI. Together with the changes that address Point 2A, we added the following changes in lines 585-588: “It must be noted, however, that the secondary injury cascade occurring after traumatic SCI is complex and includes various proinflammatory pathways. Our findings regarding the effects of IL-1 α likely apply to the first 24 hours after the injury, when the cytokine is present.” Also, the following text was added to further clarify this point (lines 540-542): “Collectively, these data suggest that SCI triggers a local response in the CNS, characterized by the release of alarmins such as IL-1 α that in turn induce the death of mature OLs in the first 24 hours.”

2C) This line of reasoning also logically explains as to why not all findings replicate and/or are not as robust in the SCI model compared to where IL-1a is delivered directly, as other cell types and pathways likely add and/or must drive the progressive secondary cell death during the subacute and intermediate-chronic phase of SCI. It is important in that regard to provide clarity as to what the treatment regimen was for the experiments where NAC was administered to SCI mice – this appears to be missing from the revised manuscript and is important to include as there are many other cellular sources and/or pathways other than IL-1a-IL-1R1 signaling that would contribute to ROS production in this pathology.

We agree that the addition of specific information on the NAC treatment would allow for a better understanding and analysis of the data that we generated. In this regard, we added the following to lines 449-450, “Although NAC administration the day before and during the first 24 hours post-SCI failed to reduce neutrophil infiltration at day 1 after SCI (Fig. 10M-O) [...]” Furthermore, we appreciate the indication that the treatment of NAC in SCI was not present in the manuscript, and we apologize for this omission. This information is now described in the *Materials and Methods* section (page 32, lines 722-725): “Mice received 4 intraperitoneal injections of saline or NAC (150 mg/kg) at 12 hours before, 1 hour before, 12 hours after and 24 hours after i.c.m. injection of either PBS or rmIL-1 α . A similar dose and regimen of NAC (or saline) was given to SCI mice. Tissue was collected at 24 hours post-i.c.m. injection/SCI for subsequent immunohistological analysis”.

2D) Lastly, and perhaps a more minor point, the manuscript does not really consider whether the mode of oligodendrocyte cell death is the same for SCI and IL-1 α infusion, and also whether the various other well-established alarmins/DAMPs mentioned in the introduction are released following IL-1 α infusion and associated cell death. If occurring, then this does not necessarily take away from the findings of IL-1 α having an initiating role, but more so would put a nuance on its potency. This should be acknowledged.

The question proposed here is very interesting, as the mechanisms of OL death and the relationship between IL-1 α and other DAMPs is a great point of interest in our laboratory. Unfortunately, we do not currently have enough data to make any conclusions about these facts. The following text was added to acknowledge that further work will be needed to address this important issue (page 27, lines 640-642): “Future investigations will be necessary to know the effects of the IL-1 α /IL-1R1 pathway in later stages of SCI, along with its role in injury when other complementary DAMPs and inflammatory pathways are triggered.”

3) ** Reviewer 1, Point 3: The effect of several of the KI/KO models is limited (i.e. less than 30% increase in gene expression or less than 20% decrease). Could this not influence the extent to which these models show a change in response compared to control? Authors claim the lack of effect indicates the cell type is not directly involved in the effects but I am not convinced the data is strong enough to support that....**

Overall, it is fair for the authors to comment that some of the limitations around the use of conditional and inducible gene targeting strategies apply to many high-impact studies, yet they remain 'state-of-the-art' as no better alternatives are available. The authors do a good job in my view in acknowledging these limitations, and the combination of approaches where *Il1r1* is either selectively ablated or restored certainly is a strength of the paper.

Additional analyses have been performed to address this point, focusing specifically in recombination efficiency in relevant cell types isolated from the CNS of adult mice (as opposed to neonatal pups). The newly added data suggest that, at least for astrocytes, much higher levels (60-80%) of IL-1R1 ablation or reconstitution were achieved. This adds significantly to the strength of the observations around this particular cell type but does not necessarily address possible weaknesses and/or overreach regarding some other lines used (e.g. *Pdgfra*-CreER, see comments on author response to queries re: original Fig. 6 below).

We thank Reviewer 4 for acknowledging our efforts to characterize the recombination rate in the astrocyte ablation/reconstitution mouse lines. Regarding the limitations related to the *Pdgfra*^{CreER} transgenic mice and their derivatives, the following clarifications were made to the text to add some nuance. On page 13, lines 288-289, the text was rectified as follows: "Altogether, these results suggest that IL-1 α does not seem to cause OL cell death by a direct mechanism of action." We point out that even if the percentage of recombination was low in this line, the *in vitro* experiment investigating the direct stimulation of OLs with IL-1 α supports the idea that IL-1 α does not directly produce OL death. Still, we share the concern of the reviewer, and our text was modified and results discussed in light of this limitation, as follows (page 18, lines 425-433: "Consistent with previous data demonstrating that i.c.m. delivery of IL-1 α to *Pdgfra*^{CreER}::*Il1r1*^{r/r} mice did not result in death of mature spinal cord OLs (Fig. 5), we found that the addition of rmIL-1 α to the control medium resulted in minor, non-significant toxicity for OLs *in vitro* (Fig. 10E). However, the transfer of conditioned medium from IL-1 α -stimulated astrocytes to cultured primary OLs was sufficient in evoking their death, as assessed using the LDH cytotoxicity assay. No significant cytotoxicity was detected on primary mature OLs when incubated with astrocyte-conditioned medium from untreated astrocytes. This indicates that even if IL-1 α exposure might result in low level OL death, IL-1 α -stimulated astrocytes clearly release factors that are lethal to mature OLs."

4) ** Reviewer 1, Point 4: ...Male and female rodents have been shown to have very different neuroimmune responses. It is not clear whether sexes were analysed separately to take into account possible sex differences.

The revised manuscript convincingly shows that there is no sex difference in the key readouts (neutrophil infiltration and oligodendrocyte death) following IL-1a infusion, and hence that pooling is appropriate.

We are delighted to see that the revised manuscript further clarifies this aspect, and we thank Reviewer 4 for thoroughly reviewing our methodology.

5) ** Reviewer 1, Point 5: Lab has previous work demonstrating IL-1a release by microglia. Current goal is to determine the extent to which this release drives secondary degeneration via astrocytes and endothelial cells....did not provide enough overview of neuroinflammation and the role of cytokines/IL-1a to convince me the question is novel...similar papers in existence.

Previous work from this group in SCI indeed showed that IL-1a is released by myeloid cells, with the current paper narrowing this down to microglia being the source. Considering the temporal profile of IL-1a expression / release and the fact that robust recruitment of monocytes and macrophages more so takes place beyond the 24 hour timepoints, this finding is logical and not surprising but it remained to be unequivocally confirmed.

Otherwise, whilst I do agree with Reviewer 1 that there is a significant body of literature on IL-1a in various forms of acquired CNS injury, there are many contextual nuances and/or differences between these pathologies and a careful dissection of how the alarmin IL-1a influences the outcome from traumatic SCI remained outstanding. It is this sort of careful dissection and mechanistic understanding as to how, when and where inflammatory mediators exert their effects that will ultimately be needed to successfully design and translate new targeted therapies into the clinic.

Overall, it could be argued that the authors have addressed this satisfactory concern by rewriting aspects of their introduction, providing a contextualised overview of current insights, why alarmins are important targets for further investigation, and then also the knowledge gap, i.e.

that mechanistic insights as to how specific alarmins drive secondary damage are incomplete, as highlighted in their response and the revised text.

We truly appreciate this acknowledgment of our revised work.

6) ** Reviewer 1, Point 6: Statement...."transition from primary to secondary damage is triggered, at least in part, by the release of the intracellular content of necrotic cells....". Could give a clearer picture of the importance of IL-1a aside from basic generalisation.

This point extends from the previous one and the authors have addressed this as part of their revision (see again author response and highlighted text in the introduction, in particular lines 70-73 and 79-83.)

This point had been previously addressed, and we thank Reviewer 4 for this acknowledgement.

7) ** Reviewer 1, Point 7: ...Figure 1...Nothing has been done here to show that macrophages aren't also releasing it.

This point has been adequately addressed, with the revised Figure 1 pointing out damaged and/or activated microglia. There is also data showing that IL-1a protein is present in tdTomato-positive microglia and not GFP-positive cells, which mostly represents immune infiltrate in LysM-eGFP mice (this is well established).

We appreciate Reviewer 4's analysis of our revised figure and we fully agree with the given explanation.

8) ** Reviewer 1, Point 8: ...Figure 2...Olig2 alone is insufficient to resolutely indicate activation of OLs. Olig2 is not specific to OLs.

This has been adequately addressed by the inclusion of CC-1 staining, which is a well-established and accepted marker of OLs.

We appreciate Reviewer 4 for acknowledging the supplementary data we provided in our revision.

9) ** Reviewer 1, Point 9: ...Data not shown: OL loss increased as IL-1a concentration increased.... Figures 2 and 3 would be more impactful if combined.

The comment regarding the relationship between IL-1a concentration and OL loss was not addressed. This may be of significance as it could explain as to why olig2+cc1+ cell counts following IL-1a infusion appear quite different between experiments (compare e.g. new Fig. 2H with new Fig. 3C).

Regardless, original Figures 2 and 3 have been merged as requested.

We agree that the relationship between IL-1 α concentration and OL loss must be added. These changes were added in the previous revision of the paper, and are described on page 9 (lines 209-211) of the manuscript as well as in Supplementary Figure 4H-I: “However, the increase in neutrophil infiltration and decrease in the number of mature OLs at 24 hours post-IL-1 α injection was amplified given higher concentrations of the alarmin (Suppl. Fig. 4H-I), thus suggesting a dose-response effect.” We apologize for not including these changes in the previous rebuttal.

10) ** Reviewer 1, Point 10: BrdU experiment to examine OPC proliferation following IL-1a injection....an example where cytokine injection does not necessarily equal SCI...Olig2 alone is not specific for OLs....limitations/translatibility could be discussed further.

The authors have provided a balanced argument/overview of relevant literature in their rebuttal that supports a similarly rapid OPC response to SCI, along with the subsequent maturation of these into mature oligodendrocytes. The Discussion has been amended accordingly (lines 535-547). Coming back to my earlier comment (in relation to the response to Reviewer 1, Point 2 above), this would be another opportunity to mention that the role of IL-1a in inducing the death of mature OLs is likely restricted to the first 24 hours of injury only.

Otherwise, as per point 8, the inclusion of CC-1 as a marker for mature OLs addressed the comment regarding staining specificity.

We agree that this is a good opportunity to discuss the limitations of our model, and the differences when compared to SCI. Together with the changes already made in Point 2, we made the following

changes in lines 534-542: “This may be because the inflammatory response that develops after SCI is different than after acute central delivery of IL-1 α . Unlike in the IL-1 α injection model, SCI results in the formation of a fibrotic/glial scar that encompasses the primary damage, but simultaneously prevents spinal cord repair. Still, whether remyelination positively impacts locomotor recovery after SCI remains open for debate ⁷². A better understanding of the mechanisms driving the replacement of myelin and OL loss would be very beneficial, contributing to the understanding of various other pathological contexts. Collectively, these data suggest that SCI triggers a local response in the CNS, characterized by the release of alarmins such as IL-1 α that in turn induce the death of mature OLs in the first 24 hours.”

11) ** Reviewer 1, Point 11: Suppl. Fig 2....IL-1a and IL-1R1 KO mice recover significantly better from SCI....not sure why it's supplemental.

This data has now been included in the main manuscript and is part of the new Figure 4.

One thing that stood out but wasn't further discussed is the fact that IL-1a KO mice recovery substantially better than IL1r1 KO mice here, both in the main BMS score and also the BMS subscores. Should they not at least be similar? Differences in SCI recovery also appear present between the respective WT cohorts, most obviously so in the BMS sub-scoring. Were all strains fully backcrossed onto a C57BL/6J background? I appreciate that each experiment is individually controlled but it does bring into question the variability of the lesion model. It would be good for the authors to comment on this and also to include relevant injury parameters (actual applied force and associated displacement between experiments / groups) in the Methods section.

This is a very good observation. Although the injury parameters provided in the *Materials and Methods* section (“ [...] the vertebral column was stabilized and a contusion of 50 kdyn was performed using the Infinite Horizon SCI device (Precision Systems & Instrumentation).”) have been kept constant throughout all the experiments, several years passed between these two independent experiments. Thus, it is possible that the use of the device over time created variability between these two experiments. Even if we agree that the difference between IL-1 α -KO and IL-1R1-KO mice is interesting, the only way to confirm its significance would be to compare all these groups with a control group in the same experiment.

12) ** Reviewer 1, Point 12: Suppl. Figure 3... immunoblot analysis of IL-1R1...could this have picked up IL-1R3 instead or in addition to IL1R1?

The rebuttal addressed the confusion that may have arisen from using the term IL-1R3 (introduced into the literature for a truncated splice variant of IL1-R1 by a previous article from an independent group). The revised manuscript now refers to tIL-1R1 instead to avoid this confusion and/or concern about specificity.

12A) Otherwise, the reference in the Discussion that “Our immunoblotting findings confirm that IL-1R1 or a truncated splice variant of IL-1R1 is expressed by microglia, astrocytes and OLs ...” could perhaps be sharpened and/or be more specific, as it seems an intriguing coincidence that only those cells that express the truncated form of IL-1R1 appear to express c-Fos in response to IL-1a stimulation.

We apologize for this misunderstanding. Not all the cells that express c-Fos in response to IL-1 α stimulation seem to express the truncated IL-1R1 (τ IL-1R1). The objective of this immunoblotting was to: “confirm that IL-1R1 and/or τ IL-1R1 is expressed by microglia, astrocytes and OLs, despite our failure to detect this protein by way of immunofluorescence staining...” Furthermore, not all Fos⁺ cell-types seem to express τ IL-1R1, as microglia seems to be exempt. Of course, it is too early to confirm which cell types express which isoform of IL-1R1, and further investigation is needed. In the previous sentence in lines 491-492, we made this modification: “or a truncated splice variant of IL-1R1” for “IL-1R1 or τ IL-1R1” in order to avoid this misunderstanding.

13) ** Reviewer 1, Point 13: Figure 6...restoring Il1r1 gene expression in OPCs...claim the lack of response means that IL-1a is not directly responsible for OL death (as now shown in Fig. 2h of the revised manuscript).

Possible issues around low-ish recombination efficiency have been partially addressed through additional experiments, that is, for astrocytes but not OLs (impact of the pandemic on authors ability to do the latter is noted).

In absence of additional verification as to what the restoration efficiency was for OLs, the authors rebut by stating that "treatment of primary OLs was not toxic in cell culture experiments (see modified Fig. 10E)". This I am not convinced of, however, as there is a clear increase (more than a doubling) in LDH cytotoxicity visible in Fig. 10E where rm-IL1a is added to the DMEM (i.e.

medium not conditioned by astrocytes). This does not take away from the fact that a much greater degree of cell death is induced when OLs are exposed to conditioned medium from astrocytes stimulated with rmIL-1 α , but it would probably be significant if a direct comparison was done between both DMEM conditions (+/- IL1 α). The authors would do better therefore to err on the side of caution, remove any ambiguity and simply state in the relevant section(s) of the Results and Discussion that a low level of OL death, directly resulting from IL-1 α exposure, cannot be excluded at present because of the above limitations / observations.

We thank the reviewer for their thorough examination of our manuscript and figures. We have modified the *Results* section as such (lines 426-428: “we found that the addition of rmIL-1 α to the control medium resulted in minor, non-significant toxicity for OLs *in vitro* (Fig. 10E).” We have also tampered our conclusion to include mention of the slight toxicity, as follows (lines 431-433): “This indicates that even if IL-1 α exposure might result in low level OL death, IL-1 α -stimulated astrocytes clearly release factors that are lethal to mature OLs.”

14) ** Reviewer 1, Point 14: Figure 7 (now Fig. 6 in revised manuscript)...goal is to determine whether IL-1R1 has auto regulatory effect on microglia....don't think the possibility can be completely ruled out. For both Figure 6 and 7 (Fig. 5 and 6 in revised manuscript). Figure C appears to have a slight downward trend...

The authors partially rebut this criticism by acknowledging some of the limitations around conditional gene targeting in their Discussion. The point that was originally made here by Reviewer 1 is, however, that partial restoration (not deletion) appears to lead to a downward trend in the number of OLs (compare red bars). This is particularly obvious for what is now Figure 5C, further emphasising the previous point to tone down statements that IL-1 α does not directly induce OL death.

Otherwise, the level of recombination, spread of data points within/between groups and statistics otherwise make it reasonable to state that microglial IL-1R1 appears redundant, possibly so because of Il1r2 up regulation, as shown.

We toned down our statements discussing the direct effect of IL-1 α on OL/microglia by adding nuance to our conclusion as follows: “Altogether, these results suggest that IL-1 α does not seem

to cause OL cell death by a direct mechanism of action.” (lines 288-289) and “...suggesting that microglial IL-1R1 signaling does not seem to mediate the central effects of IL-1 α .” (line 302).

15) ** Reviewer 1, Point 15: Figure 8 (now Fig. 7 in revised manuscript)...conditional restoration of IL1-R1 in endothelial cells...small return of the baseline response? Neutrophil infiltration partially recovered and also partial attenuation of OL death. This suggests endothelial cells are involved in the response to IL-1 α . Confirmed in conditional KO of IL-1R1 in endothelial cells, but no change in recovery this time.

The authors have addressed the first part of these criticisms by repeating experiments in which rmIL1 α is injected into mice where IL-1R1 expression is selectively restored in endothelial cells. The results confirm earlier findings regarding the effect size of this manipulation regarding neutrophil recruitment and OL death. It is fair to say on the basis of these experiments that downstream consequences of IL-1 α signaling in other cell types must contribute to both phenomena. Consistent with this, neutrophil recruitment is not completely abolished when selectively knocking out IL-1R1 in endothelial cells (Fig. 7I), although it should also be noted that the loss of oligodendrocytes is pretty much prevented here (Fig. 7J).

What remains puzzling, and this was also highlighted by Reviewer 1 in their last comment, is why the functional outcomes from SCI are not in agreement with each other. Specifically, restoring IL-1R1 expression in endothelial cells partially ‘rescues’ the neutrophil response that is normally seen in response to rmIL-1 α infusion and also induces cell death, yet, there is no impact in relation to function from which one can conclude that the degree of neutrophil recruitment and/or OL death was not sufficient to attenuate the augmented recovery seen with global IL-1R1 deficiency (red vs. blue lines in Fig. 7e and f), with both strains of mice recovering significantly better compared to the WT controls (black line). On the other hand, selectively knocking out IL-1R1 in endothelial cells only reduces neutrophil recruitment and OL death, which makes sense, yet no change is seen in SCI experiments in relation to the functional outcome. This could be discussed more clearly. I am less bothered by the fact that histopathological observations in the rmIL-1 α infusion model do not correlate well here with SCI outcomes, but it does re-emphasize the point made earlier that the effects of IL-1 α may be very acute only and that other pathways are more dominant and/or significant in relation to OL death in the post-acute phase.

At a minimum, I would recommend that the authors acknowledge this and provide a nuance to their statement that "...astrocytes and CNS ECs appear to drive OL death via an IL-1R1-dependent release of ROS and other molecules that have yet to be identified" (lines 635-636), adding something along the lines that, based on the evidence at hand, we can reasonably assume that this applies to the first 24 hours post-SCI only and then less so thereafter, in addition to clarifying perhaps in light of this that "other molecules" here implies other pathways, not other molecules downstream of IL-1R1.

A similar nuance can be provided to the cited statement in the rebuttal (line 596-597 of the revised manuscript) that"secondary injury cascade after trauma SCI is complex and likely involves multiple pathways", again emphasizing the point perhaps that the latter are likely to be more dominant and/or driving OL loss in the post-acute phase.

We agree that the relation between the IL-1 α pathway and its functional outcomes remains puzzling. As suggested by Reviewer 4, we have now acknowledged this issue by adding a paragraph that discusses the limitations of our model, and restates that our model applies to the first 24 hours (see lines 634-642): "SCI generates an extensive activation of several proinflammatory pathways, many of which do not resolve over the course of the pathology⁹². The administration of IL-1 α i.c.m. allowed us to study the effects of this alarmin in the spinal cord, and to replicate histopathological markers up to 24 hours after the injection, which is when this cytokine is present. Although this model is appropriate for studying the function of IL-1 α , the administration of a single dose of this cytokine is not enough to replicate the chronicity and complexity of inflammation and the physical outcomes present in SCI. Future investigations will be necessary to know the effects of the IL-1 α /IL-1R1 pathway in later stages of SCI, along with its role in injury when other complementary DAMPs and inflammatory pathways are triggered."

Also, we modified lines 631-633 as follows: "Instead, astrocytes and CNS ECs appear to drive OL cell death via an IL-1R1-dependent release of ROS in the first 24 hours, and other pathways that have yet to be identified in the post-acute phase."

The last concern about lines 596-597 was addressed in Point #2.

16) ** Reviewer 1, Point 16: Figure 9 (relating Fig. 8 in the revised manuscript): ...using GFAP promoter to restore IL-1R1 expression in astrocytes...low efficacy of KO and KI make me wonder

if part of the lack of an observed effect is for this reason and not related to the mechanism of IL-1a action.

The revised manuscript contains new data to show that recombination efficiency was much higher (60-80% as opposed to the 20-30% that was stated in the original manuscript where cells from neonatal pups rather than adult mice were used). This indeed considerably reduces any of the initially expressed concern around interpretation / robustness of the findings.

The revised manuscript also contains data to show that only restoration of IL-1R1 in astrocytes appears to drive these cells more towards an A1 phenotype, that is, on the basis of C3 expression. This implies a direct effect of IL-1a on astrocytes as opposed to an indirect one where e.g. IL-1R1-mediated endothelial cell activation and the downstream consequences of this may (partially) drive astrocytic responses.

I would make one comment here though which is in relation to the statement that “knockout of 80% of *Ilr1* expression in astrocytes failed to reduce neutrophil infiltration but partially prevented OL death in response to IL-1a administration (Fig. 8H-J)” (line 384-385). Overall, I do not disagree with the authors that there is likely a synergistic effect of IL-1a signaling in both astrocytes and ECs that contributes to SCI outcomes. That does not take away, however, that this statement appears to be very much overreaching and is not substantiated by what is actually shown in Fig. 8J. Looking at the spread of data points within and between groups, one can only conclude one thing based on what is presented, which is that the conditional deletion of IL-1R1 from astrocytes has no bearing on the extent of OL death in these experiments, or in the very least that there is no statistical basis/evidence for this. Consistent with that, restoring IL-1R1 expression in astrocytes also does not reinstate the expected level of OL death, with the findings not being significantly different from the WT controls (Fig. 8D, rmIL-1a, grey vs. red bar). The authors should tone down these conclusions as they distract from what is otherwise an excellent manuscript, unnecessarily creating an impression perhaps that there was preconceived bias in what the experimental outcome should be.

We appreciate the reviewer’s comment and agree that some of our conclusions were erroneous. Accordingly, the text was rectified as follows (page 16, lines 376-383). “Most of these observations were confirmed via another transgenic mouse model, namely *Gfap*^{Cre}::*Il1r1*^{fl/fl} mice (Fig. 8G). In agreement with previous findings, knockout of 80% of *Il1r1* expression in astrocytes

failed to reduce neutrophil infiltration (Fig. 8H-I & Suppl. Fig. 7E-F). However, partial deletion of IL-1R1 expression in astrocytes of IL-1 α -treated *Gfap*^{Cre::Il1r1^{fl/fl} mice did not restore OL cell counts back to normal levels, i.e. those seen in PBS-treated animals (Fig. 8J & Suppl. Fig. 7G-H). Indeed, despite a slightly increased trend in OL viability in the cKO mouse line compared to control groups after i.c.m. injection of IL-1 α , these results did not reach significance.”}

However, we respectfully disagree with the reviewer’s second point that restoring IL-1R1 expression in astrocytes did not reinstate the expected level of OL death, with findings not being significantly different from the WT controls (Fig. 8D, rmIL-1 α , grey vs. red bar). In this particular graph, one would have expected OL cell counts to not be statistically different between the two groups if OL death was to be restored in IL-1 α -treated *Gfap*^{Cre::Il1r1^{r/r} mice to the extent seen in IL-1 α -treated WT mice, which is the case with a p-value of 0.2788. Importantly, both *Gfap*^{Cre::Il1r1^{r/r} and WT mice had significantly reduced numbers of OLs in their spinal cord in response to i.c.m. injection of IL-1 α compared to PBS, with respective p-values of 0.0046 and <0.0001 for these comparisons.}}

Once again, we appreciate the comments of the reviewers and are delighted that our previous responses and revisions satisfactorily addressed their points. Please do not hesitate to contact us if additional information is required.